# An Analysis of Tokenization: Transformers under Markov Data

**Nived Rajaraman**
UC Berkeley
nived.rajaraman@berkeley.edu

**Jiantao Jiao**
UC Berkeley
jiantao@berkeley.edu

**Kannan Ramchandran**
UC Berkeley
jiantao@berkeley.edu

## Abstract

While there has been a large body of research attempting to circumvent tokenization for language modeling (Clark et al., 2022; Xue et al., 2022), the current consensus is that it is a necessary initial step for designing state-of-the-art performant language models. In this paper, we investigate tokenization from a theoretical point of view by studying the behavior of transformers on simple data generating processes. When trained on data drawn from certain simple $k^{\text{th}}$-order Markov processes for $k > 1$, transformers exhibit a surprising phenomenon - in the absence of tokenization, they empirically are incredibly slow or fail to learn the right distribution and predict characters according to a unigram model (Makkuva et al., 2024). With the addition of tokenization, however, we empirically observe that transformers break through this barrier and are able to model the probabilities of sequences drawn from the source near-optimally, achieving small cross-entropy loss. With this observation as starting point, we study the end-to-end cross-entropy loss achieved by transformers with and without tokenization. With the appropriate tokenization, we show that even the simplest unigram models (over tokens) learnt by transformers are able to model the probability of sequences drawn from $k^{\text{th}}$-order Markov sources near optimally. Our analysis provides a justification for the use of tokenization in practice through studying the behavior of transformers on Markovian data.

## 1 Introduction

The training of language models is typically not an end-to-end process. Language models are often composed of a "tokenizer", which encodes a sequence of characters into a sequence of token ids, which map to substrings. The subsequent language modeling task is carried out by a neural network or transformer, which is pre-trained and fine-tuned on large datasets. The ideal goal is to jointly train the tokenizer and transformer with end-to-end accuracy as the objective. This is a challenging problem to solve efficiently, and thus, the tokenizer is generally adapted on a portion of the training dataset and frozen before the transformer is trained. In practice, byte-level/character level models such as ByT5 (Xue et al., 2022) and CANINE (Clark et al., 2022) which avoid tokenization often perform worse for the reason that semantic relationships can be harder to capture at the character level (Libovickỳ et al., 2021; Itzhak and Levy, 2021).

Though used most commonly, tokenization at the subword level often has sharp edges. Test sequences may contain rare tokens which were never seen in the training dataset. The presence of such tokens may induce undesirable behavior in the outputs of models (Rumbelow and Watkins, 2023; Kharitonov et al., 2021; Yu et al., 2021) and present an attack surface for bad actors. Moreover, tokenized models struggle on tasks that involve manipulation at the character level, such as spelling out words or reversing sentences. For similar reasons, LLMs with standard tokenizers also struggle to carry out basic arithmetic (Golkar et al., 2023). Despite this brittleness, tokenization is used in nearly all state-of-the-art LLM architectures.

38th Conference on Neural Information Processing Systems (NeurIPS 2024).

In this paper, we introduce a statistical formulation for tokenization for next-word-prediction. We study the class of models transformers are observed to express empirically under simple data generating processes, which often can have simpler descriptions. Taking a step back, rather than focusing on proxy evaluation metrics, which lead to an ever-changing goalpost, we focus on understanding the behavior of the end-to-end cross-entropy loss, $\mathcal{L}(\cdot)$. In this paper, we study a simplification of real world data generating processes and study the case where data sources are $k^{\text{th}}$-order Markov processes. Within this framework we can compare tokenizers against each other, and in the process capture several interesting phenomena. Our main results are as follows,

1. There are very simple $k^{\text{th}}$-order Markov processes such that in the absence of any tokenization, transformers trained on data drawn this source empirically predict characters according to a unigram model. This phenomenon is observed under a wide variety of hyperparameter choices. This is problematic because unigram models such as that induced by the stationary distribution are poor at modeling Markovian data and suffer from a high cross-entropy loss. This phenomenon was also recently observed in Makkuva et al. (2024).

2. When trained with tokenization, transformers are empirically observed to break through this barrier and are able to capture the probability of sequences under the Markov distribution near-optimally. In other words, in the presence of tokenization, transformers appear to achieve near-optimal cross-entropy loss. This phenomenon is observed with a multitude of tokenizers used commonly in practice.

3. We analyze a toy tokenizer which adds all length-$k$ sequences into the dictionary and show that as dictionary size grows, unigram models trained on the tokens get better at modeling the probabilities of sequences drawn from Markov sources. We then theoretically prove that tokenizers used in practice, such as the LZW tokenizer (Zouhar et al., 2023a) and a variant of the BPE tokenizer (Gage, 1994; Sennrich et al., 2016) which are learnt from data also satisfy this property but require much smaller dictionaries to achieve any target cross-entropy loss.

In our framework, the most challenging hurdle and the biggest departure from previous work such as (Zouhar et al., 2023b) is the element of generalization - understanding how a tokenizer performs on new sequences that it was not trained on. This generalization turns out to be a delicate phenomenon - we show in Appendix D that there exist tokenizers which generalize poorly in the sense that they may compress the dataset they are trained on into a short sequence of tokens, but fail to generalize to new sequences. In Appendix E we show that there exist dictionaries which generalize well (in the sense of having low cross-entropy loss) to new sequences under one encoding algorithm, but completely fail to generalize under another.

## 1.1 Related Work

Tokenization has a long history of empirical study in natural language processing. In the literature, a number of tokenizers have been developed for various domains such as math (Singh and Strouse, 2024), code (Zheng et al., 2023; Parr, 2013) and morphology-aware tokenizers for different languages like Japanese (Tolmachev et al., 2018; Den et al., 2007) and Arabic (Alyafeai et al., 2023) among many others. In modern LLMs, the most commonly used tokenizers are variants of BPE (Gage, 1994), Wordpiece (Schuster and Nakajima, 2012) and the Unigram tokenizer (Kudo, 2018) which learn a dictionary from data, rather than hard-coding language dependent rules. There has been a long line of work interpreting tokenization from various lenses (Grefenstette and Tapanainen, 1994; Palmer, 2000; Zouhar et al., 2023b).

The theoretical study of transformers has also received much attention recently. We discuss the closest relatives to our work below. Edelman et al. (2024) study the learning trajectory of transformers trained on data drawn from $1^{\text{st}}$-order Markov chains. While the authors empirically observe that the models eventually learn to predict tokens correctly according to the Markov kernel, simplicity bias slows down optimization - the models initially predict tokens according to a unigram model (in context unigrams), which delays learning the optimal solution. This phenomenon was also observed in Makkuva et al. (2024). On the positive side, Nichani et al. (2024) study an in-context causal learning task that generalizes learning in-context bigrams for $1^{\text{st}}$-order Markov processes and analyze the trajectory of gradient descent.

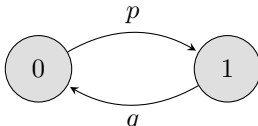

Figure 1: *2-state switching process.* The above state diagram describes the distribution of $X_n$ conditioned on $X_{n-1}$. $k^{th}$-*order extension:* the conditional probability of $X_n$ only depends on $X_{n-k}$ through the kernel, $\Pr(X_n = 1 | X_{n-k} = 0) = p$ and $\Pr(X_n = 0 | X_{n-k} = 1) = q$.

**Notation.** All logarithms are base $e$, unless specified otherwise. The Shannon entropy $H(X)$ of a categorical random variable $X$ is $-\sum_{x \in \text{supp}(X)} p(x) \log p(x)$. $H_{\text{BER}}(p)$ captures the entropy of a Bernoulli random variable with parameter $p$. The notation $O_{p,q,r}(f(n))$ (likewise $\Omega_{\{\cdot\}}$ and $\Theta_{\{\cdot\}}$) indicate that the underlying constant depends polynomially on the parameters $p$, $q$ and $r$ and $\widetilde{O}(f(n))$ (likewise, $\widetilde{\Theta}$ and $\widetilde{\Omega}$) ignores $\text{polylog}(n)$ terms. For a set $S$, $S^\star = \cup_{k=1}^{\infty} S^k$, the set of all sequences with elements drawn from $S$. For a sequence $\boldsymbol{t}$, $\boldsymbol{t}_{i:j} = (\boldsymbol{t}_i, \boldsymbol{t}_{i+1}, \cdots, \boldsymbol{t}_j)$ returns a slice.

## 2 Formulation

We consider a setting where the learner's objective is to learn a language model which models probabilities of sequences over an input alphabet $\mathcal{A}$. The data to be modeled is generated according to an unknown probability model $P : \mathcal{A}^\star \to [0, 1]$ over strings. A tokenizer is a tuple $\mathcal{T} = (\text{Dict}, \text{enc}(\cdot), \text{dec}(\cdot))$. Here Dict is a collection of tokens The encoding function $\text{enc}(\cdot) : \mathcal{A}^\star \to \text{Dict}^\star$, maps strings of characters to a sequence of tokens, and likewise, the decoding function $\text{dec}(\cdot) : \text{Dict}^\star \to \mathcal{A}^\star$ maps a sequence of tokens to a string of characters. We assume that the tokenizer is "consistent", namely, $\text{dec}(\text{enc}(\cdot))$ is the identity function.

We consider a setting where the learner has access to a training dataset which is a sequence of length $n$ sampled from a data source[1]. We study the likelihood maximization problem, where the objective of the learner is to learn an end to end model such that the cross-entropy loss is minimized. In the presence of tokenization, we have a model of the form $Q_{\text{end}} = Q \circ \text{enc}(\cdot)$ where $Q$ is a joint distribution across sequences of tokens when the tokenizer corresponding to $\text{enc}(\cdot)$ is used. The cross-entropy loss, i.e. the log-perplexity, can be written down as,

$$\mathcal{L}_m(Q_{\text{end}}) \triangleq -\mathbb{E}[\log Q(\text{enc}(\boldsymbol{s}))], \tag{1}$$

with the objective to minimize it. Here, the expectation is over $\boldsymbol{s}$, a fresh test sequence of length $m$ sampled from the data generating process. Fixing a tokenizer, let $\mathcal{Q}$ denote a family of joint distributions over tokens (i.e. likelihood models). The objective is to jointly design a tokenizer (with encoding function $\text{enc}(\cdot)$) and likelihood model $Q \in \mathcal{Q}$ with small test loss $\mathcal{L}_m(Q \circ \text{enc}(\cdot))$.

Finally, for a dictionary Dict, the unigram family of models, $\mathcal{Q}_{\text{1-gram}}$, is defined as below: $Q \in \mathcal{Q}_{\text{1-gram}}$ associates probability $Q(\boldsymbol{t}_1, \boldsymbol{t}_2, \cdots, \boldsymbol{t}_j) = Q_{\#}(j) \prod_{i=1}^{j} Q_{\text{tok}}(\boldsymbol{t}_i)$ to the sequence of tokens $\boldsymbol{t}_1, \cdots, \boldsymbol{t}_j$ for measures $Q_{\#}$ and $Q_{\text{tok}}$ supported on $\mathbb{N}$ and Dict respectively.

### 2.1 Data generating process

In this paper, we consider a simplification of real-world data generating processes by considering the case where the data generating distribution is a $k^{th}$-order Markov process over characters. Studying the behavior of transformers trained on Markov data was the subject of the works Makkuva et al. (2024) and Edelman et al. (2024), where a number of interesting phenomena were unearthed. When a transformer is trained on data from certain simple Markov processes like the one considered in Figure 1, a very peculiar phenomenon occurs - within a reasonably large number of iterations, the transformer fails to improve beyond the loss incurred by the best unigram model. This phenomenon is reproducible across a wide number of hyperparameters, including the number of feed-forward layers in the model, the embedding dimension, and the number of attention heads. In Figure 3a this is made clearer - the transformer fails to improve its test loss beyond that of the best unigram model.

---

[1]This can be thought of as the concatenation of all the individual sequences in the training dataset.

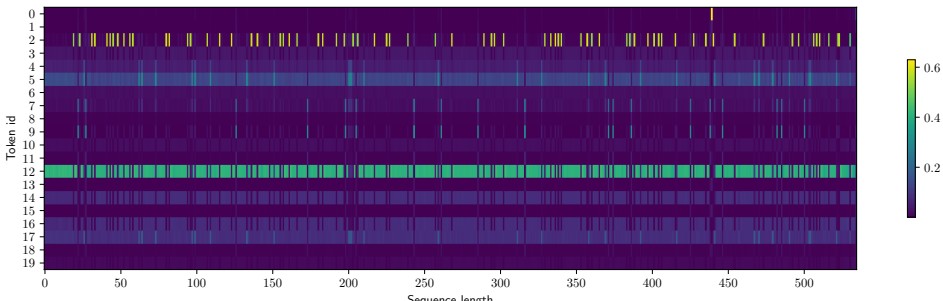

Figure 2: Token distribution returned by the transformer tokenized by a learnt BPE encoder with a dictionary size of 20. A test sequence is generated from the stochastic source and encoded into a token sequence $t$. Each narrow vertical column represents the distribution over next tokens returned by the transformer when the first $x$ tokens of $t$ are fed into the model, where $x$ is varied from 0 to the length of $t$. For most values of $x$, the model appears to predict the same distribution over the next token.

How bad can a unigram model be? It turns out that the gap between the cross-entropy of the best unigram model and that of the optimal model can be characterized precisely.

**Theorem 2.1.** *Consider any ergodic data source with stationary distribution over characters $\pi$. The unconstrained optimal likelihood model achieves cross-entropy loss, $\min_Q \mathcal{L}_m(Q) = H(P)$. In contrast, the cross-entropy loss under any unigram model $Q \in \mathcal{Q}_{1\text{-gram}}$ satisfies, $\mathcal{L}_m(Q) \geq mH(\pi)$.*

The ratio of the optimal loss $H(P)$, and the optimal unigram loss, $mH(\pi)$ can be arbitrarily large. In particular, for the switching chain in Figure 1, as $p, q \to 0$, the ratio diverges to $\infty$.

While transformers are a powerful class of models, it is concerning that they fail to learn very simple distributions such as $k^{\text{th}}$-order Markov processes. Why do they work so well in practice if they can be so slow to learn Markovian data? It turns out that there is a simple missing ingredient in all the architectures considered so far: tokenization. All the models trained in Figure 3a operate on raw character sequences drawn from the stochastic source. To understand the role of tokenization, we run another experiment and train the transformer on sequences generated from the stochastic source which are encoded into tokens by a BPE tokenizer learnt from data. The transformer now operates on sequences of tokens, rather than sequences of individual symbols. In Figure 3b we plot the results of this experiment - in the presence of tokenization, the cross-entropy loss of the end-to-end model breaks past the unigram barrier and approaches the optimal bound within a small number of iterations.

Let's peek into the model a bit more and understand its behavior. In Figure 2 we run the following experiment: we sample a random sequence of length 2000 from a Markov chain and feed it into the transformer after tokenization, resulting in $\approx 500$ tokens. We plot the next-token distribution predicted by the transformer at every single position in the input, generated by autoregressive masking. In Figure 2 we stitch together these next-token distributions, each of which is a narrow column heatmap. Visually, we observe that the plot is approximately homogeneous along the $x$-axis, implying that the next-token distribution learned does not depend strongly on the prefix at that position. Thus the transformer learns what is essentially a unigram model.

Thus, we come to a surprising conclusion: the behavior of the transformer on the $k^{\text{th}}$-order switching source in Figure 1 with and without tokenization is essentially the same. In both cases, the model learns a unigram model over the tokens - in the absence of tokenization this unigram model is in fact the stationary distribution induced by the source. If the transformer learns a unigram model in both cases, how come there is such a large gap in performance between the two? To understand this in more detail, we analyze a toy tokenizer. As a simplification, we will analyze the behavior of an arbitrary, but exact unigram model under this tokenizer.

## 3 Unigram models under tokenization

Let's consider a toy tokenizer which assigns all possible substrings of length $r$ as tokens in the dictionary and study what happens when a unigram model is trained on the tokenized sequences. The

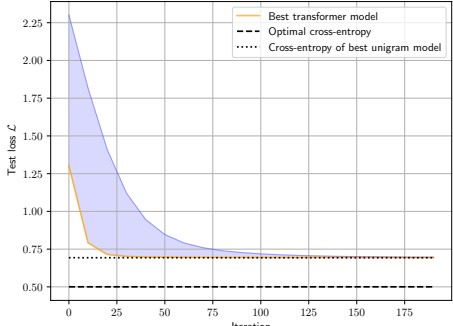 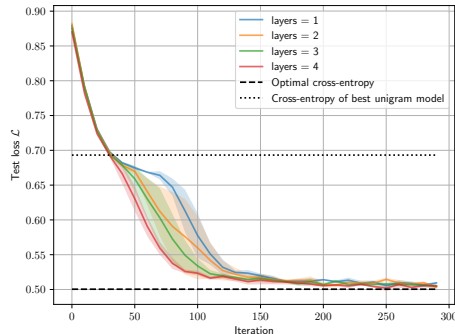

(a) The loss of the transformer fails to converge to the optimal cross-entropy loss (dashed line) and instead converges to that of the best unigram model (dotted line). The shaded blue region captures how the test loss curves vary as hyperparameters (number of layers, embedding dimension etc.) are changed.

(b) In the presence of tokenization, the test loss of the model approaches the optimal bound (dashed line). It is worth noting that the models trained here are significantly smaller than those considered in Figure 3a, having up to $70\times$ fewer parameters and yet are able to achieve the optimal cross-entropy loss.

Figure 3: Transformers trained on the order-2 switching Markov process (Figure 1) with $p = q = 0.8$. On the left we have the model trained without tokenization and on the right the model uses BPE with a dictionary of size 10 learnt from data.

total dictionary size $d = 2^r$. A sequence of characters is mapped to a sequence of tokens by simply chunking it into a sequences of $r$ characters which are replaced by the corresponding token index[2]. The resulting stochastic process on the tokens is still Markovian, but over a state space of size $2^r$. For any unigram model $Q$ on the tokens, the cross-entropy loss can be written down as,

$$\mathcal{L}_m(Q \circ \mathsf{enc}(\cdot)) = \mathbb{E}\left[\sum_{\boldsymbol{t} \in \mathsf{enc}(\boldsymbol{s})} \log(1/Q_{\text{tok}}(\boldsymbol{t}))\right] + \Theta(\log(m)),$$

where we choose $Q_{\#} = \mathrm{Unif}([m])$, which contributes an additive $\log(m)$ to the loss. Choosing $Q_{\text{tok}}(\boldsymbol{t}) = \pi(\boldsymbol{t}_1)\prod_{i=1}^{r-1} P(\boldsymbol{t}_{i+1}|\boldsymbol{t}_i)$ as the stationary probability the Markov process associates with $\boldsymbol{t}$,

$$\frac{1}{m}\mathcal{L}_m(Q \circ \mathsf{enc}(\cdot)) \approx -\frac{1}{m}\mathbb{E}\left[\log(P(\boldsymbol{s}) + \sum_{i=0}^{m/k-1} \log\left(\frac{\pi(\boldsymbol{s}_{ki+1})}{P(\boldsymbol{s}_{ki+1}|\boldsymbol{s}_{ki})}\right)\right]$$

$$\stackrel{(i)}{\approx} \frac{1}{m}H(P) + \frac{1}{mk}\left(mH(\pi) - H(P)\right)$$

$$= \frac{H(P)}{m}\left(1 - \frac{1}{\log_2(d)}\right) + \frac{H(\pi)}{\log_2(d)}. \tag{2}$$

the approximation in $(i)$ uses the fact that as $m$ grows large, $\frac{1}{m}\sum_{i=0}^{m/k} \log(P(\boldsymbol{s}_{ki+\ell+1}|\boldsymbol{s}_{ki+\ell}))$ approaches $\frac{H(P)}{k}$. With $d = 2$ (i.e., $r = 1$), we recover the performance of the character tokenizer in Theorem 2.1. An immediate implication of this simple calculation is that as $m \to \infty$, there is a unigram model which is nearly optimal as the dictionary size grows to $\infty$.

While this toy tokenizer allows us to glean this intuition behind why tokenization allows unigram models to be near-optimal, there are some obvious issues. One, the tokenizer does not adapt to the distribution of the data. Indeed, for the switching Markov source in Figure 1, as $p = q = \delta \to 0$, the source contains increasingly longer sequences of contiguous 0's and 1's. In this case, it makes since to have a dictionary containing such sequences, rather than all possible length-$r$ sequences, many of which would be seen very few times (if at all) in a test sequence. At a more technical level, in eq. (2), to get to a cross-entropy loss of $2H(P)$, the size of the dictionary required by the toy tokenizer is $e^{mH(\pi)/H(P)}$. As discussed in Example A.1 for the switching Markov process with $p = q = \delta$, this dictionary size can be extremely large and scales exponentially (in $1/\delta$) as $e^{1/\delta \log(1/\delta)}$ when $\delta$ is

---

[2]The last few characters which do not add up to $r$ in total are left untokenized. These boundary effects will not matter as the test sequences grow in length

small. In general, on stochastic sources on a much larger alphabet, such as English/ASCII, this toy tokenizer would result in a prohibitively large dictionary.

Larger dictionaries are usually correlated with the presence of rare tokens which appear infrequently at training time. This presents a problem in practice - a lot more data is often required to see enough examples of such tokens to learn good embeddings for them. More importantly, in the absence of this volume of data, rare tokens present an attack surface to elicit undesirable behavior in the model (Rumbelow and Watkins, 2023). In practice, this issue present with the toy tokenizer is, to an extent, resolved by using tokenization algorithms such as BPE or Wordpiece, which learn dictionaries from data. In the process, they are able to avoid learning extremely rare tokens, by enforcing a lower bound on the number of their occurrences in the training data to be allocated as a token. By minimizing the number of such rare tokens, the model is able to utilize its token budget in a more efficient manner.

We now introduce the main theoretical result of this paper, showing that with the appropriate tokenization algorithm with a token budget of $d$, a unigram model is not only asymptotically able to achieve the optimal cross-entropy loss, but also requires far smaller dictionaries to match the performance of the toy tokenizer considered earlier. In order to avoid dealing with the transient characteristics of the source, we consider the cross-entropy loss in eq. (1) under the assumption that the test sequences $s$ are of length $m \to \infty$. Namely, define the normalized loss,

$$\mathcal{L}(\cdot) = \lim_{m \to \infty} \frac{1}{m} \mathcal{L}_m(\cdot)$$

**Theorem 3.1.** *Consider a Markov data generating process which satisfies Assumption 3.2. Let $d$ denote a budget on the size of the dictionary. Then, there exists a tokenizer with at most $d$ tokens and encoding function* $\textsf{enc}(\cdot)$*, such that,*

$$\min_{\mathcal{Q} \in \mathcal{Q}_{1\text{-}gram}} \mathcal{L}(Q \circ \textsf{enc}(\cdot)) \leq \frac{1}{1 - \varepsilon} \min_{Q'} \mathcal{L}(Q') \tag{3}$$

*where $\varepsilon$ is $\log(1/\delta)/0.99 \log(d)^3$. Furthermore, a tokenizer satisfying eq.* (3) *with probability $\geq 1 - d^{-\Omega_\delta(\log(d))}$ can be learnt from a dataset of $\widetilde{O}_\delta(d)$ characters.*

The tokenizers considered in this theorem are far more efficient with their token budget than the toy tokenizer - to achieve a cross entropy loss within a factor 2 of optimal, the dictionary size required by these tokenizer is $d \approx 1/\delta^2$ on any source satisfying Assumption 3.2. In comparison, the toy tokenizer requires a dictionary size of $e^{1/\delta \log(1/\delta)}$ to achieve the same error. We show that the LZW tokenizer proposed in (Zouhar et al., 2023a) achieves the upper bound in eq. (3) when trained on a dataset of size $\widetilde{O}(d)$. Likewise, we also show that a sequential variant of BPE achieves the upper bound in eq. (3) up to a factor of 2 and with a worse dependency in $\varepsilon$ when trained on a dataset of size $\widetilde{O}(d^2)$. What is interesting is that neither of these algorithms explicitly learn a unigram likelihood model, $Q$, while constructing the dictionary. Yet they are able to perform as well as the tokenizers which are jointly optimized with a likelihood model, such as the Unigram tokenizer (Kudo, 2018).

**Key insight.**   While the toy tokenizer provides a high level intuition as to why tokenization might enable unigram models to model Markov sources well, here we present a different explanation which captures tokenization from an operational viewpoint. Tokenizers which do a good job at learning patterns in the data and assigning these frequent patterns as tokens in the dictionary are compatible with an i.i.d. model over tokens. A hypothetical example motivating this point: consider a tokenizer such that the distribution of tokens in the encoding of a fresh string sampled from the source is distributed i.i.d., except that whenever the token $t'$ appears, it is always followed by $t''$. An i.i.d. model on the tokens is a poor approximation since $P(t't'') \gg P(t')P(t'')$. However, by merging $t'$ and $t''$ into a new token $t$ and adding this to the dictionary, the new distribution over tokens is i.i.d. In general, this motivates why it is desirable for a tokenizer to allocate new tokens to substrings which appear next to each other frequently, i.e. a pattern in the data. As more tokens are added to the dictionary, one might expect the cross-entropy loss incurred by the best unigram model to improve.

## 3.1   Learning patterns in the source

The main result of this section is a generic reduction: dictionaries which typically encode new strings into a few long tokens (defined in a formal sense in Theorem 3.4), result in tokenizers achieving

---

[3]$\varepsilon$ is assumed to be $< 1$ in this statement. The constant 0.99 can be made arbitrarily close to 1.

near-optimal cross-entropy loss. We prove this result for Markovian sources under a regularity assumption, which is that the associated connectivity graph of the chain is complete. The analogous assumption for $k^{\text{th}}$-order sources is that the transition kernel is entry-wise bounded away from $0$. This assumption is satisfied by all the sources considered in the paper thus far, such as the $k^{\text{th}}$-order switching processes in Figure 1.

**Assumption 3.2** (Data generating process). Assume that the data source is an ergodic Markov process with transition $P(\cdot|\cdot)$ and stationary distribution $\pi$. Assume that $\min_{a,a' \in \mathcal{A}} P(a'|a) \triangleq \delta > 0$.

*Remark* 3.3. Assumption 3.2 (and its $k^{\text{th}}$-order extension) impose that there is a small but non-zero probability of observing any particular symbol after any preceding sequence. This limits the applicability of these processes in real-world scenarios where such a phenomenon may not occur. However, our motivation for this assumption is different: $\delta$ allows parameterizing the Markov process in a way which interpolates between i.i.d. ($\delta = 1/|\mathcal{A}|$) and highly non-i.i.d. ($\delta \to 0$).

For a substring $\boldsymbol{s}$ and a character $a$, define $P(\boldsymbol{s}|a) = P(\boldsymbol{s}_1|a) \prod_{i=2}^{|\boldsymbol{s}|} P(\boldsymbol{s}_i|\boldsymbol{s}_{i-1})$ denote the conditional probability of the substring $\boldsymbol{s}$. We now state the main result of this section.

**Theorem 3.4** (Bound on cross-entropy loss of dictionaries under greedy encoder). *Consider a source satisfying Assumption 3.2 and any tokenizer $\mathcal{T}$ equipped with the greedy encoder, $\textsf{enc}_{gre}(\cdot)$ with finitely long tokens. Define, $P(\boldsymbol{t}) = \mathbb{E}_{a \sim \pi}[P(\boldsymbol{t}|a)]$ and suppose $H(Q_{MLE}, P) \geq \frac{1}{\varepsilon} \log(1/\delta)$ for some $\varepsilon < 1$. Then,*

$$\min_{Q \in \mathcal{Q}_{1\text{-}gram}} \mathcal{L}(Q \circ \textsf{enc}_{gre}(\cdot)) \leq \frac{\min_Q \mathcal{L}(Q)}{1 - \varepsilon}.$$

**Interpretation.** $H(Q_{\text{MLE}}, P) = \mathbb{E}_{\boldsymbol{t} \sim Q_{\text{MLE}}}[\log(1/P(\boldsymbol{t}))]$ is large when the encoder places higher mass (i.e. larger values of $Q_{\text{MLE}}(\cdot)$) on tokens which have low probability under $P$, i.e. which correspond to longer substrings. Intuitively, this metric is higher for tokenizers which typically use long tokens (i.e. low $P(\cdot)$) to encode new strings.

## 3.2 LZW tokenizer

In this section we study the Lempel-Ziv-Welch (LZW) based tokenization scheme introduced by Zouhar et al. (2023a) and establish guarantees of the form of Theorem 3.1 for this tokenizer.

**Definition 3.5** (LZW tokenizer). Iterating from left to right, the shortest prefix of the training dataset which does not already exist as a token is assigned as the next token in the dictionary. This substring is removed and the process is iterated on the remainder of the dataset. The tokenizer uses the greedy encoding algorithm (Definition A.3) to encode new strings into tokens.

*An example of the LZW tokenizer:* For the dataset $0100111$, the dictionary created is $\{0, 1, 00, 11\}$.

The LZW tokenizer is based on the LZW algorithm for compression (Ziv and Lempel, 1978; Welch, 1984). The dictionary satisfies the property that if some substring $\boldsymbol{s}'$ exists as a token in the dictionary, then all of its prefixes must also belong to the dictionary. In the next theorem, we show that the LZW tokenizer approximately achieves the optimal cross-entropy loss.

**Theorem 3.6.** *Suppose the LZW tokenizer is trained on a dataset of length at most $d$ (thereby learning a dictionary with at most $d$ tokens). For Markov sources satisfying Assumption 3.2, with probability $\geq 1 - d^{-O_\delta(\log(d))}$, the resulting tokenizer satisfies,*

$$\min_{Q \in \mathcal{Q}_{1\text{-}gram}} \mathcal{L}(Q \cdot \textsf{enc}_{gre}(\cdot)) \leq \frac{\min_Q \mathcal{L}(Q)}{1 - \varepsilon}.$$

*where $\varepsilon = \frac{\log(1/\delta)}{0.99 \log(d)}$[4].*

The proof of this result considers all substrings $\boldsymbol{t}$ with $P(\boldsymbol{t}) \geq 1/d^{0.99}$. These substrings are reasonably high probability and observed many times in a dataset of $\widetilde{\Omega}(d)$ characters. We show that with high probability, the LZW tokenizer learns *all* of these substrings as tokens in the dictionary. Now, when processing a new string, since the greedy algorithm only emits the longest substring

---

[4]The constant $0.99$ can be made arbitrarily close to $1$.

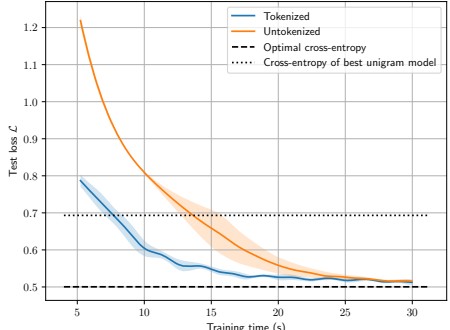 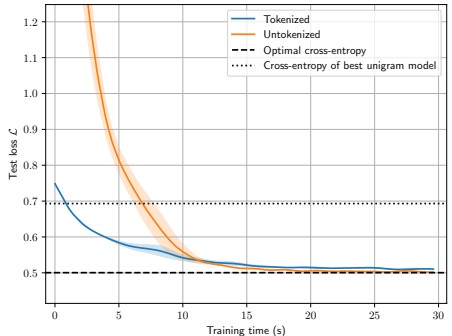

(a) Convergence rate of smallest model which is within 10% of the optimal-cross entropy by 300 epochs. The smallest untokenized model has 9010 parameters (3 layers, embedding dimension = 10). The smallest tokenized model with a dictionary size of 10 has 17880 parameters (3 layers, embedding dimension = 20). The tokenized model has more parameters but the wall-clock time taken to reach any loss value is smaller.

(b) Convergence rate of models with the same embedding dimension (20), number of heads (1) and layers (3) with and without tokenization. The model with tokenization (dictionary size of 20) appears to converge more quickly, but the error floor is subtly higher compared to the model without tokenization. Both models are trained on input sequences of length 512. The width of the tokenized model is smaller (145).

Figure 4: Test loss vs. wall-clock time for the tokenized and untokenized models when trained on the order-1 switching Markov chain (Figure 1) with $p = q = 0.8$. The tokenizer used is BPE.

which matches a token, every token allocated must fall on the "boundary" of this set, having $P(\boldsymbol{t}) \leq O(1/d^{0.99})$. By definition, this means that $H(Q_{\mathrm{MLE}}, P) = \mathbb{E}_{\boldsymbol{t} \sim Q_{\mathrm{MLE}}}[\log(1/P(\boldsymbol{t}))] = 0.99 \log(d)$. Combining this with Theorem 3.4 completes the proof. At a high level, on the infinite tree of substrings $\mathcal{A}^{\star}$ we study which nodes are populated as tokens by LZW. This structure forms a Digital Search Tree (DST) and prior work analyzes the mean and variance of the profile of the DST under various source processes (Jacquet et al., 2001; Drmota and Szpankowski, 2011; Hun and Vallée, 2014; Drmota et al., 2021). A detailed proof of Theorem 3.6 is provided in Appendix A.6.

## 4   Experimental Results

**Experiment 1 (Figures 4a and 4b)**   In this experiment we study the order-1 switching Markov chain. Transformers without tokenization empirically achieve a small cross-entropy on this learning task as seen in Figure 4a and earlier in Makkuva et al. (2024). We vary hyperparameters to find the smallest untokenized model which achieves a loss within 10% of the optimal-cross entropy within 300 epochs. Fixing a token dictionary size of 20, we also find the smallest tokenized model which achieves the same loss. Although the smallest model with tokenization is larger than the smallest model without tokenization in terms of the number of parameters, the wall-clock time taken to optimize the model to any target test loss is observed to be smaller. Thus, tokenization appears to reduce the compute time required to train the model to a target test loss in the toy example we consider. In Figure 4b we compare models with the same architecture trained with and without tokenization[5]. The model with tokenization appears to converge more quickly, although the limiting error achieved is subtly higher in comparison with the model without tokenization.

**Experiment 1 (Figure 5).**   In this experiment, we train tokenizers on the Wikitext-103-raw-v1 dataset (Merity et al., 2016) and compare the performance of unigram models trained on the GLUE dataset as the model size scales. Since the character-level tokenizer operates on a fixed vocabulary, in order to compare with the other tokenizers, we plot the number of unique $k$-grams observed in the training dataset along the $x$-axis. While this is not an apples-to-apples comparison, we use the number of unique $k$-grams in the dataset as a proxy for the complexity of the likelihood model trained.

---

[5]The model with tokenization has width equal to the typical length of sequences *after* encoding, which is smaller.

One may also use the total number of possible $k$-grams as a proxy; however a large fraction of these $k$-grams would likely never be observed in a real dataset (especially as $k$ grows).

**Experiment 2 (Table 1).** In this experiment, we compare the cross entropy loss of the best unigram model trained on pre-trained tokenizers on an array of datasets. All the considered tokenizers have dictionary sizes in the range 31K-51K. The best bigram model under the character tokenizer is consistently outperformed by the best unigram likelihood model trained under a number of pre-trained tokenizers on a variety of datasets: Rotten Tomatoes (8.5K sequences), GLUE (105K), Yelp review (650K) and Wikitext-103-v1 (1.8M).

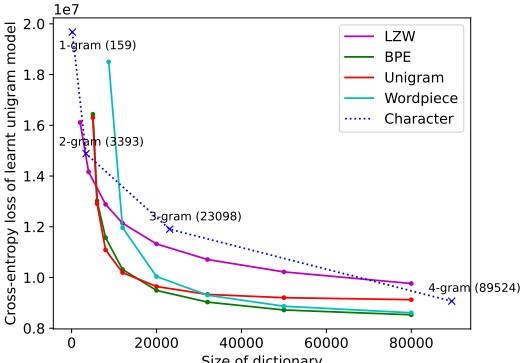

|  | RT | Wiki | Yelp | GLUE |
|---|---|---|---|---|
| BERT | 1.58 | 1.55 | 1.60 | 1.50 |
| Tinyllama | 1.75 | 1.84 | 1.82 | 1.70 |
| GPT-neox | 1.57 | 1.64 | 1.66 | 1.48 |
| Mistral | 1.69 | 1.80 | 1.75 | 1.66 |
| Phi-2 | 1.54 | 1.62 | 1.64 | 1.45 |
| Character | 2.40 | 2.45 | 2.46 | 2.38 |

Figure 5: *Performance vs. dictionary size.* Tokenizers are trained on the Wikitext-103 dataset. For all other tokenizers we train unigram models while for the the character-level tokenizer, we train $k$-gram models for $k \in \{1, 2, 3, 4\}$. Likelihood models are trained on the GLUE dataset. The parentheses indicates the number of distinct observed $k$-grams, which lower bounds the $k$-gram model complexity.

Table 1: Cross-entropy loss estimates (using eq. (55)) of unigram models trained on pre-trained tokenizers under a number of datasets. The last row (blue) is the character level tokenizer, on which a more powerful bigram model is trained. BERT is based on Wordpiece, and the remaining tokenizers are BPE based. The character-level tokenizer we use is ByT5.

## 4.1 Additional theoretical results

We present some additional theoretical results in the appendix which we discuss briefly below. In Appendix B, we do a theoretical study of the cross-entropy loss achieved by the popular BPE tokenizer. We show that a variant of BPE achieves the upper bound on the RHS of eq. (3) (Theorem 3.1) up to a factor approaching 2 as the dictionary size grows. It is an interesting question for future research to understand whether this factor of 2 can be removed, since transformers are observed to achieve the near-optimal cross-entropy loss as the dictionary size grows (cf. Figure 3b). In Appendix C, we prove finite sample bounds on the end-to-end model under the LZW tokenizer with a smoothed empirical estimator as the unigram model. This analysis reveals that there is a sweet spot for the dictionary size - too small a dictionary, and the statistical error floor is significant, too large a dictionary, and the statistical error incurred by the likelihood model dominates the overall loss. We also take a closer look into the aspect of generalization for tokenizers, which arises from the fact that the tokenizer is evaluated on data that it was not trained on. Prior work such as Zouhar et al. (2023b) show that BPE is an approximation algorithm for finding the sequence of merges which minimizes the size of the compressed dataset. This does not imply any guarantees on the end-to-end performance, or even compression power of the tokenizer on new sequences. In particular, in Appendix D we show that there exist tokenizers which compress the dataset into a short sequence of tokens, but do so in a way which fails to generalize to new sequences. Thus measuring the performance of a tokenizer necessitates understanding its behavior on data it was not trained on. In Appendix E, we show a different kind of intricacy - there exist tokenizers under which the best unigram model achieves low cross-entropy loss. However, the same dictionary under a different encoding algorithm performs nearly as poorly as the character-level tokenizer. The interaction between the dictionary and encoding algorithm is a poorly studied subject in the tokenization literature; this result emphasizes the importance of understanding this relationship.

## 5 Open questions

In this section, we discuss some limitations of our work and open questions stemming from them. We show that when transformers are trained with or without tokenization, they learn to approximately represent $k$-gram models for different values of $k$. Transformers are capable of representing far more complex behavior, which are elicited under more complex data generating processes. Extending our formulation to these settings presents an avenue to develop an even better understanding of tokenization, and would allow finer-grained comparisons between tokenizers. The behavior and role of tokenizers may be very different in these contexts. Below we discuss some concrete questions.

Our theory assumes that the underlying Markov chain has every transition occurring with non-zero probability, which is a limitation. However, the analysis for the toy tokenizer in eq. (2) shows that when the dictionary size scales as $\exp(mH(\pi)/H(P))$, even in the absence of Assumption 3.2, the tokenizer achieves the optimal cross-entropy to within a factor of 2. This leads to the following conjecture.

**Conjecture 1.** In the spirit of eliminating Assumption 3.2, is it possible to establish a version of Theorem 3.1 applicable to data drawn from any Markov chain, where $\varepsilon = \log(1/\delta)/0.99\log(d)$ is replaced by $\varepsilon = \log(mH(\pi)/H(P))/0.99\log(d)$.

In Appendix B, we analyze a variant of the BPE tokenizer, which carries out a version of sample splitting, and establish a weaker variant of Theorem 3.1 for this tokenizer. This is to simplify the statistical dependencies arising from the fact that while learning its dictionary, BPE makes a run over the entire training dataset each time a new token is added. It remains an open question to analyze and establish a variant of Theorem 3.1 for the standard BPE tokenizer.

## 6 Conclusion

We present a theoretical framework to compare and analyze different tokenization algorithms. We study the end-to-end cross-entropy loss of the tokenizer + likelihood model, and focus on the case where the data generating process is Markovian. We empirically observe that transformers with tokenization are drastically more efficient at learning $k^{\text{th}}$-order Markov processes, compared to without tokenization. We prove that algorithms such as LZW and a sequential variant of BPE learn tokenizers such that the best unigram likelihood model trained on them approaches the cross-entropy loss of the optimal likelihood model, as the vocabulary size $d$ grows.

## 7 Acknowledgements

JJ and NR were partially supported by NSF Grants IIS-1901252 and CCF-2211209. KR was partially supported by NSF Grant CCF-2211209.

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

# Appendix

## Contents

## A  Analysis of LZW: Proofs of Theorems 3.4 and 3.6

### A.1  Notation and definitions

For each character $a \in \mathcal{A}$ let $\mathcal{T}_a^\star$ denote an infinite tree, with root vertex $\emptyset$, and subsequent vertices labelled by strings $\boldsymbol{t} \in \mathcal{A}^\star$. The edge from a parent vertex $\boldsymbol{t}$ to any child $\boldsymbol{t}a'$ is labelled with the probability $P(\boldsymbol{t}a'|\boldsymbol{t})$ unless $\boldsymbol{t} = \emptyset$, in which case the edge probability is $P(a'|a)$. An infinite trajectory sampled on the tree $\mathcal{T}_a^\star$ corresponds to an infinite string sampled from the stochastic source conditioned on the first character of the string being $a$. In this paper we only consider ergodic sources (Gray and Gray, 2009) for which we can define the "entropy rate". The entropy rate fundamentally captures the compressibility of the source, and can be defined as $H_\infty \triangleq \lim_{m \to \infty} \frac{1}{m} H(P)$ where $\boldsymbol{s}$ is a length $m$ string drawn from the source. By Theorem 2.1, $H_\infty$, captures $\min_Q \mathcal{L}(Q)$.

### A.2  A basic result about the optimal achievable cross-entropy loss

The ratio of $H(P)$ and $mH(\pi)$ can be made arbitrarily large for the switching Markov chains in Figure 1 as the switching probabilities $p$ and $q$ approach 0 or 1. See Example A.1 for more details.

*Example* A.1. Consider the switching Markov process in Figure 1 on $\{0, 1\}$ with $p = q = 1 - \delta$. For this process, $\lim_{m \to \infty} \frac{1}{m} H(P) = H_{\mathsf{Ber}}(\delta) = \delta \log(1/\delta) + (1 - \delta) \log(1/(1 - \delta))$, but $\pi = \{1/2, 1/2\}$ and so $H(\pi) = H_{\mathsf{Ber}}(1/2) = \log(2)$. The ratio $\lim_{m \to \infty} \frac{mH(\pi)}{H(P)}$ goes to $\infty$ as $\delta \to 0$.

### A.3 Proof of Theorem 2.1

We first characterize the minimum achievable cross-entropy loss $\mathcal{L}_m(Q)$ without any restrictions on the likelihood model class $\mathcal{Q}$. Choosing $Q(\mathsf{enc}(s)) = Q(s) = P(s)$, the true probability of the sequence $s$, we have $\mathcal{L}_m(Q \circ \mathsf{enc}(\cdot)) = H(s)$ where $H(\cdot)$ is the entropy function. It is not that difficult to see that this is also the minimum cross-entropy loss that can be achieved. For any distribution $Q$,

$$\begin{aligned}
\mathcal{L}_m(Q) &= \mathbb{E}[\log(1/Q(s)] \\
&= \mathbb{E}[\log(P(s)/Q(s)] + \mathbb{E}[\log(1/P(s))] \\
&= H(P) + D_{\mathrm{KL}}(P\|Q).
\end{aligned}$$

On the other hand, the cross-entropy loss under any unigram model $Q \in \mathcal{Q}_{\text{1-gram}}$ satisfies,

$$\begin{aligned}
\frac{1}{m}\mathcal{L}_m(Q \circ \mathsf{enc}(\cdot)) &\stackrel{(i)}{=} -\frac{1}{m}\sum_{i=1}^{m}\mathbb{E}[\log Q_{\mathrm{tok}}(t_i)] - \frac{1}{m}\mathbb{E}[\log Q_{\#}(m)] \\
&\stackrel{(ii)}{\geq} -\sum_{a \in \mathcal{A}}\pi(a)\log Q_{\mathrm{tok}}(a) \\
&\geq H(\pi)
\end{aligned}$$

where in $(i)$, we use the definition of the unigram model $Q$, and in $(ii)$, $\pi$ is the stationary distribution over characters induced by the stochastic source, and the ergodicity of the source is used. The last equation lower bounds $H(X, Y) \geq H(X)$.

### A.4 Maximum likelihood unigram model

A number of our results (Theorems 3.4 and 3.6 to name a few) are related to bounding $\min_{Q \in \mathcal{Q}_{\text{1-gram}}} \mathcal{L}(Q \circ \mathsf{enc}(\cdot))$ for some tokenizer $\mathcal{T}$. In this section we introduce the maximum likelihood unigram model which captures the optimizer over $Q$ for any given tokenizer.

For the character level tokenizer, an examination of Theorem 2.1 shows that the optimal unigram likelihood model associates probability $Q_{\mathrm{tok}}(a) = \pi(a)$, i.e. the limiting fraction of times the character $a$ is observed in the sequence. More generally, for a non-trivial tokenizer, the corresponding optimal unigram model $Q^{\star}_{\mathrm{tok}}(t)$ ends up being the limiting expected fraction of times $t$ is observed in an encoding of a sequence. This is the maximum likelihood unigram model, which we formally define below. The unigram MLE likelihood model associates probability,

$$Q_{\mathrm{MLE}}(t) \leftarrow \lim_{m \to \infty} \mathbb{E}\left[\frac{n_t}{\sum_t n_t}\right] \tag{4}$$

to each token, where $n_t$ is the random variable capturing the number of occurrences of the token $t$ in the encoding of the length-$m$ string $s$. Restricting the class of likelihood models to the unigram models, $\mathcal{Q}_{\text{1-gram}}$, $Q_{\mathrm{MLE}}$ captures the model which minimizes eq. (1).

The unigram MLE model cannot be computed without an infinite amount of data, but can be approximated well with a finite amount of data, which forms the basis for Theorem C.1. For certain encoding algorithms, we can show that the quantity $n_t/\sum_t n_t$ asymptotically converges to its expectation (Lemma A.4). This is the reason the unigram model in eq. (4) is referred to as a "maximum likelihood" model, since $\lim_{m \to \infty} n_t/\sum_t n_t$ is the limit as $|s| = m \to \infty$ of the solution to the following likelihood maximization problem: given a sequence $s$, find the distribution over tokens, $Q$, which maximizes

$$\prod_{t \in \mathsf{enc}(s)} Q(t) \equiv \prod_{t \in \mathsf{Dict}} \left(Q(t)\right)^{n_t}.$$

As discussed previously, the unigram MLE model over tokens in eq. (4) induces a joint distribution over sequences of tokens by looking at the product of the marginal probabilities of the composed tokens; in particular,

$$Q_{\mathrm{MLE}}(t_1, \cdots, t_j) = Q_{\mathrm{MLE}}(j)\prod_{i=1}^{j} Q_{\mathrm{MLE}}(t_i),$$

where $Q_{\mathrm{MLE}}(j)$ is a distribution on the total number of tokens generated and is instantiated as Unif($[m]$).

*Remark* A.2. Note that the unigram MLE model specifies a distribution over tokens which is a function of the underlying encoding algorithm, $\mathsf{enc}(\cdot)$. Different encoders result in different population level distributions over tokens, and consequently different unigram MLE models.

**Definition A.3** (greedy encoder). Given a dictionary $\mathsf{Dict}$, the greedy encoder $\mathsf{enc}_{\mathrm{gre}}(s)$ encodes a string $s$ into tokens by greedily matching from left to right, the largest substring that exists as a token in $\mathsf{Dict}$. This substring is then removed and the process iterated on the remainder of $s$. The greedy decoder $\mathsf{dec}_{\mathrm{gre}}(\cdot)$ is a lookup table - a sequence of tokens is decoded by replacing each occurrence of a token by the corresponding substring it maps to in $\mathsf{Dict}$.

**Lemma A.4.** $\lim_{m \to \infty} \frac{n_t}{\sum_{t'} n_{t'}} \overset{a.s.}{=} \lim_{m \to \infty} \mathbb{E}\left[\frac{n_t}{\sum_{t'} n_{t'}}\right]$ *for any tokenizer having a finite vocabulary and finitely long tokens, using the greedy encoder.*

*Proof.* This result is essentially true because under the greedy encoder, the tokens in an encoding of a fresh string $t$ may be generated by an $r^{th}$-order Markov process for some $r$. For such processes, the Cesàro average of the state distributions converges to a stationary distribution of the process (i.e., the Krylov–Bogolyubov argument).

Tokens are generated as follows. Suppose the previous tokens generated were $t_1, t_2, \cdots, t_i$. The next token $t_{i+1}$ is sampled by drawing an infinite trajectory from $\mathcal{T}_a^\star$ for $a \sim P(\cdot|t_i)$ and returning the longest prefix $t$ of this trajectory which is a token in $\mathsf{Dict}$, conditional on satisfying the conditions, $t_j t_{j+1} \cdots t_i t \notin \mathsf{Dict}$ for all $j \in \{1, 2, \cdots, i\}$. This process is repeated sequentially to generate all the tokens.

Suppose the length of the longest token in the dictionary is $\ell_{\max}$. Then, the distribution from a which a token is sampled depends on at most the previous $\ell_{\max}$ tokens. The reason for this is that the dependency of the $(i+1)^{th}$ token, $t_{i+1}$, on the previously sampled tokens emerges in the constraint $t_j t_{j+1} \cdots t_i t_{i+1} \notin \mathsf{Dict}$, satisfied by any candidate $t_{i+1}$. Since each token is of length at least one, this condition is vacuously satisfied if $j < i - \ell_{\max}$.

With this view, the evolution of the state, defined as $\mathsf{state}_r = (t_{r\ell_{\max}}, t_{r\ell_{\max}-1}, \cdots, t_{(r-1)\ell_{\max}})$ evolves in a Markovian fashion. By the Krylov–Bogolyubov argument (cf. Proposition 4.2 in Chen (2018)), the time averaged visitation frequencies of a Markov chain coordinate-wise asymptotically converges to its expectation, almost surely. This expectation exists by Theorems 8.5 and 8.22 of Eisner et al. (2015) which shows that for a matrix $A$ such that $\sup_{t \in \mathbb{N}} \|A^t\|_{\mathrm{op}} < \infty$ the limit $\lim_{t \to \infty} \frac{1}{t} \sum_{i=1}^{t} A^i$ exists. For the finite-state Markov transition $A$ which captures the token generation process, condition $\sup_{t \in \mathbb{N}} \|A^t\|_{\mathrm{op}} \le |\mathsf{Dict}|^{\ell_{\max}} < \infty$. This means that the limit of the time averaged state distribution exists. Moreover, for any initial distribution $\pi_0$ over tokens, $\pi = \lim_{t \to \infty} \frac{1}{t} \sum_{i=1}^{t} \pi_0 A^i$ satisfies the condition $\pi A = \pi$, implying that the limiting time-averaged state distribution is a stationary distribution of $A$. Since the limiting time-averaged measure on the state $\mathsf{state}_r = (t_{r\ell_{\max}}, \cdots, t_{r\ell_{\max}-1}, \cdots, t_{(r-1)\ell_{\max}})$ exists, this implies that the limiting time-averaged measure of $t_{r\ell_{\max}-r'}$ for each $r' \in \{0, 1, \cdots, \ell_{\max}\}$ exists. By taking the uniform average over $r'$ and $r$, the limiting time-averaged measure of $t_i$ over $i \in \mathbb{N}$ exists. $\qquad\square$

### A.5 Proof of Theorem 3.4

Consider a string $s$ of length $m \to \infty$ which is encoded into a sequence of tokens $(t_i : i \in [|\mathsf{enc}_{\mathrm{gre}}(s)|])$. By the Asymptotic Equipartition Property (AEP) for ergodic sources, i.e. the Shannon–McMillan–Breiman theorem,

$$\Pr\left(\lim_{m \to \infty} -\frac{1}{m} \log P(s) = H_\infty\right) = 1. \tag{5}$$

Here $\lim_{m \to \infty} \frac{H(P)}{m}$ also happens to be the entropy rate of the source. We use this property to bound the length of the greedy encoding, $|\mathsf{enc}_{\mathrm{gre}}(s)|$. Indeed, the probability of $s$ may be decomposed as,

$$P(s) = P(t_1) \prod_{i=2}^{|\mathsf{enc}_{\mathrm{gre}}(s)|} P(t_i|t_{i-1}) \le \prod_{i=1}^{|\mathsf{enc}_{\mathrm{gre}}(s)|} \max_{a \in \mathcal{A}} P(t_i|a).$$

Noting that $\delta \min_a P(\boldsymbol{t}|a) \geq \max_a P(\boldsymbol{t}|a)$, up to a $\delta$ factor we may replace the max over $a$ by an expectation over $a \sim \pi$ where $\pi$ is the stationary distribution of the stochastic source. In particular,

$$P(\boldsymbol{s}) \leq \prod_{i=1}^{|\mathsf{enc}_{\mathrm{gre}}(\boldsymbol{s})|} P(\boldsymbol{t}_i)/\delta.$$

By invoking the AEP, eq. (5),

$$\lim_{m\to\infty} \frac{1}{m} \sum_{i=1}^{|\mathsf{enc}_{\mathrm{gre}}(\boldsymbol{s})|} -\log\left(P(\boldsymbol{t}_i)\right) - \log(1/\delta) \overset{\text{a.s.}}{\leq} H_\infty$$

Recall that the greedy encoder satisfies Lemma A.4 and for any $\boldsymbol{t} \in \mathsf{Dict}$, $\lim_{m\to\infty} \frac{n_{\boldsymbol{t}}}{|\mathsf{enc}_{\mathrm{gre}}(\boldsymbol{s})|} \overset{\text{a.s.}}{=} Q_{\mathrm{MLE}}(\boldsymbol{t})$. Furthermore, note that for any token $\boldsymbol{t} \in \mathsf{Dict}$, $P(\boldsymbol{t}) > \delta^{|\boldsymbol{t}|} > 0$, and $|\mathsf{enc}_{\mathrm{gre}}(\boldsymbol{s})| \leq m$ surely. By almost sure convergence,

$$\lim_{m\to\infty} \frac{|\mathsf{enc}_{\mathrm{gre}}(\boldsymbol{s})|}{m} \sum_{\boldsymbol{t}\in\mathsf{Dict}} -\frac{n_{\boldsymbol{t}}}{|\mathsf{enc}_{\mathrm{gre}}(\boldsymbol{s})|} \left(\log\left(P(\boldsymbol{t}) - \log(1/\delta)\right)\right)$$

$$\overset{\text{a.s.}}{=} \lim_{m\to\infty} \frac{|\mathsf{enc}_{\mathrm{gre}}(\boldsymbol{s})|}{m} \left(H(Q_{\mathrm{MLE}}, P) - \log(1/\delta)\right)$$

Furthermore, utilizing the assumption that $\varepsilon H(Q_{\mathrm{MLE}}, P) \geq \log(1/\delta)$ satisfied by the tokenizer,

$$\lim_{m\to\infty} \frac{(1-\varepsilon)|\mathsf{enc}_{\mathrm{gre}}(\mathbf{s})| \left(H(Q_{\mathrm{MLE}}, P)\right)}{m} \overset{\text{a.s.}}{\leq} H_\infty. \tag{6}$$

Now we are ready to bound the expected cross-entropy loss of the tokenizer. Define the unigram model $P_\pi(\boldsymbol{t}_1, \boldsymbol{t}_2, \cdots, \boldsymbol{t}_j) = P_{\mathrm{unif}}(j) \prod_{i=1}^{j} P(\boldsymbol{t}_i)$ where $P_{\mathrm{unif}}$ is the uniform measure over $[m]$. Note that we have the inequality $\min_{Q\in\mathcal{Q}_{1\text{-gram}}} \lim_{m\to\infty} \frac{1}{m}\mathcal{L}_m(Q \circ \mathsf{enc}_{\mathrm{gre}}(\cdot)) \leq \lim_{m\to\infty} \frac{1}{m}\mathcal{L}_m(P_\pi \circ \mathsf{enc}_{\mathrm{gre}}(\cdot))$ and therefore, it suffices to upper bound the RHS. In particular,

$$\mathcal{L}_m(P_\pi \circ \mathsf{enc}_{\mathrm{gre}}(\cdot)) = -\mathbb{E}[\log P_{\mathrm{unif}}(|\mathsf{enc}_{\mathrm{gre}}(\boldsymbol{s})|)] - \mathbb{E}\left[\sum_{\boldsymbol{t}\in\mathsf{enc}_{\mathrm{gre}}(\boldsymbol{s})} \log\left(P(\boldsymbol{t})\right)\right]$$

$$\leq \log(m) - \mathbb{E}\left[\sum_{\boldsymbol{t}\in\mathsf{enc}_{\mathrm{gre}}(\boldsymbol{s})} \log\left(P(\boldsymbol{t})\right)\right] \tag{7}$$

where the last inequality uses the fact that $P_{\mathrm{unif}}(|\mathsf{enc}_{\mathrm{gre}}(\boldsymbol{s})|) = 1/m$. Note that as $m \to \infty$, by assumption on the tokenizer, the fraction of times the token $\boldsymbol{t}$ appears in the encoding of $\boldsymbol{s}$ converges almost surely converges to $Q_{\mathrm{MLE}}(\boldsymbol{t})$. Since $|\mathsf{enc}_{\mathrm{gre}}(s)| \leq m$ surely and $P(\boldsymbol{t}) > \delta^{|\boldsymbol{t}|} > 0$, by an application of the Dominated Convergence Theorem,

$$-\lim_{m\to\infty} \frac{1}{m}\mathbb{E}\left[\sum_{\boldsymbol{t}\in\mathsf{enc}_{\mathrm{gre}}(\boldsymbol{s})} \log\left(P(\boldsymbol{t})\right)\right] = -\lim_{m\to\infty} \frac{1}{m}\mathbb{E}\left[|\mathsf{enc}_{\mathrm{gre}}(\boldsymbol{s})| \cdot \sum_{\boldsymbol{t}\in\mathsf{Dict}} Q_{\mathrm{MLE}}(\boldsymbol{t}) \log\left(P(\boldsymbol{t})\right)\right]$$

$$= \lim_{m\to\infty} \frac{1}{m}\mathbb{E}\left[|\mathsf{enc}_{\mathrm{gre}}(\boldsymbol{s})|H(Q_{\mathrm{MLE}}, P)\right] \tag{8}$$

Combining eq. (7) with eq. (8) and setting $\lim_{m\to\infty} \log(m)/m = 0$, and invoking eq. (6),

$$\min_{Q\in\mathcal{Q}_{1\text{-gram}}} \lim_{m\to\infty} \frac{1}{m}\mathcal{L}_m(Q_{\mathrm{MLE}} \circ \mathsf{enc}_{\mathrm{gre}}(\cdot)) = \lim_{m\to\infty} \frac{1}{m}\mathbb{E}\left[|\mathsf{enc}_{\mathrm{gre}}(\boldsymbol{s})|H(Q_{\mathrm{MLE}}, P)\right]$$

$$\leq \frac{H_\infty}{1-\varepsilon}. \tag{9}$$

By Theorem 2.1, we have that $\min_Q \lim_{m\to\infty} \frac{1}{m}\mathcal{L}_m(Q\circ\mathsf{enc}_{\mathrm{gre}}(\cdot)) = \lim_{m\to\infty} \frac{H(P)}{m} = H_\infty$, which uses the fact that the source is ergodic. Combining with eq. (9) completes the proof.

## A.6 Heavy-hitter dictionaries and a proof of Theorem 3.6

In this section we prove Theorem 3.6 and introduce the notion of a heavy-hitting dictionary. At a high level, these dictionaries contain all the substrings which have reasonably high probability of being observed many times in a dataset of size $n = \widetilde{\Omega}_\delta(d)$. We first show in Lemma A.6 that heavy hitting dictionaries generalize well in the sense of having $H(Q_{\mathrm{MLE}}, P)$ being large (in conjunction with Theorem 3.4 this implies an upper bound on the cross-entropy loss of the best unigram model). Next, we will prove that the LZW algorithm (Definition 3.5) results in a heavy hitting dictionary with high probability.

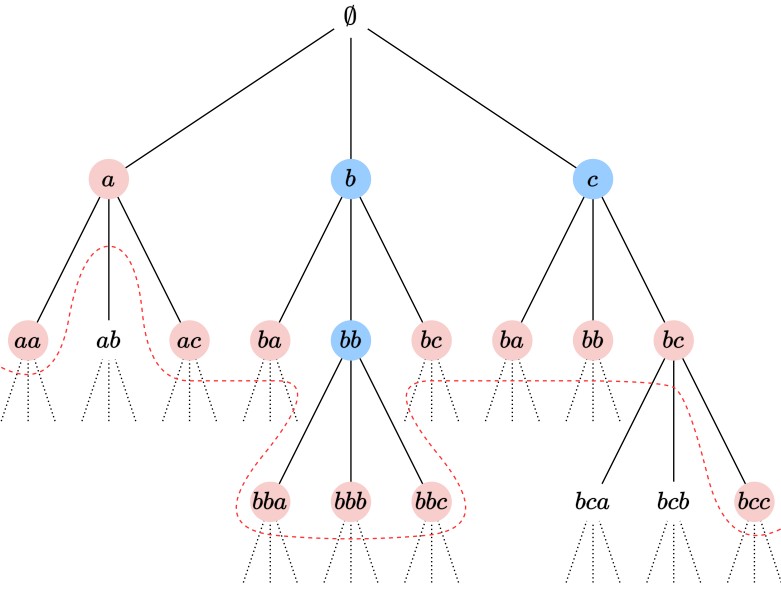

Figure 6: The circled nodes indicates substrings which are tokens in Dict. Red nodes indicate the set of "maximal tokens", which are the set of tokens which the greedy encoder assigns, leaving out those which can only be assigned as the last token of some string. Tokens like "$b$" are never assigned by the greedy encoder (save as the last token of the encoding of a string) since any sufficiently long trajectory starting with $b$ must have a longer prefix which is also a token, namely, one of $ba$, $bc$, $bba$, $bbb$ or $bbc$. The vertices of the tree which are assigned by the greedy encoder as tokens (together with all their prefixes) forms a cut of the tree, which marks the dotted red line. The heavy hitting property asserts that this cut is uniformly far away from the root node $\emptyset$, and that every vertex $s$ marked red has $P(s) \leq 1/d^{\beta}$.

**Definition A.5** ($\beta$-heavy-hitting dictionary). A token $t$ of a dictionary is said to be maximal if there exists an arbitrary substring containing $t$ as a strict prefix, and in addition, $t$ is also the largest prefix of the substring which is a token. A dictionary Dict is said to be $\beta$-heavy hitting if the set of maximal tokens is a subset of $\{s' : \max_{a \in \mathcal{A}} P(s'|a) \leq 1/d^{\beta}\}$.

A pictorial depiction of the heavy hitting property is illustrated in Figure 6.

**Lemma A.6.** *For a $\beta$-heavy-hitting dictionary, with the greedy encoder, $H(Q_{MLE}, P) \geq \beta \log(d)$.*

*Proof.* Note that the greedy encoder assigns tokens only among the set of maximal substrings (save for potentially the last token). If every maximal substring has $\max_{a \in \mathcal{A}} P(s|a) \leq 1/d^{\beta}$, by the heavy-hitting property, for any token $t$,

$$P(t) \leq \max_{a \in \mathcal{A}} P(s'|a) \leq 1/d^{\beta}.$$

Therefore,

$$H(Q_{\text{MLE}}, P) = \mathbb{E}_{t \sim Q_{\text{MLE}}}[\log(1/P(t))] \geq \beta \log(d).$$

$\square$

Define $\mathcal{M}_{\beta} = \{t : \max_{a \in \mathcal{A}} P(t|a) \geq \delta/d^{\beta}\}$. These are the set of "high-probability" substrings under the stochastic source. We will show that for $\beta$ bounded away from 1, with high probability, every substring in $\mathcal{M}_{\beta}$ is added as a token to the dictionary in a run of the LZW tokenizer (Definition 3.5). Note that if every substring in $\mathcal{M}_{\beta}$ is assigned as a token by LZW, then the algorithm must be $\beta$-heavy hitting since there always exists a maximal token on the "boundary" of the set $\mathcal{M}_{\beta}$ which is strictly contained in $\{s' : \max_{a \in \mathcal{A}} P(s'|a) \leq 1/d^{\beta}\}$.

**Lemma A.7.** *Every substring in $\mathcal{M}_\beta$ has length at most $\ell_\star \triangleq \delta^{-1}(\beta \log(d) + \log(1/\delta))$.*

*Proof.* Note that $\min_{a,a' \in \mathcal{A}} P(a|a') = \delta$, which implies that the probability of any transition must be bounded away from 1, i.e., $\max_{a,a' \in \mathcal{A}} P(a|a') \leq 1 - \delta$. This implies that,

$$\max_{a \in \mathcal{A}} P(\boldsymbol{t}|a) \leq (1 - \delta)^{|\boldsymbol{t}|} \leq e^{-\delta|\boldsymbol{t}|}. \tag{10}$$

By definition, for any substring $\boldsymbol{t} \in \mathcal{M}_\beta$, $\max_{a \in \mathcal{A}} P(\boldsymbol{t}|a) \geq \delta/d^\beta$. In conjunction with eq. (10), this implies the statement of the lemma. $\square$

In the remainder of this section, let $n$ be the size of the dataset on which LZW is run. We show that the number of tokens added to the dictionary by LZW, $d$, is $\widetilde{\Theta}_\delta(n)$. Rather than running the algorithm with early stopping (i.e., ceasing to add new tokens once the budget is hit), instead, we assume that the algorithm runs on a prefix of the dataset of length $d$. The number of tokens added this way cannot exceed $d$.

**Lemma A.8.** *With probability $\geq 1 - d^{-\Omega(\log(d/\delta)/\delta)}$, in a run of the LZW algorithm, no substring $\boldsymbol{t}$ added as a token to the dictionary satisfies $|\boldsymbol{t}| \geq \ell_{\max} \triangleq 4 \log(d|\mathcal{A}|)/\delta$.*

*Proof.* Consider any $s \in \mathbb{N}$ and any substring $\boldsymbol{t}$ of length $s$. In order for $\boldsymbol{t}$ to be assigned as a token, each of its prefixes must disjointly appear at least once in the string. Since there are at most $d$ tokens, we can upper bound the probability that $\boldsymbol{t}$ is assigned as a token as,

$$P(\boldsymbol{t} \text{ is assigned as a token}) \leq \binom{d}{s} \prod_{i=1}^{s} \max_{a \in \mathcal{A}} P(\boldsymbol{t}_{1:i}|a)$$

$$\overset{(i)}{\leq} \binom{d}{s}(1 - \delta)^{s(s-1)/2}$$

$$\leq e^{s \log(d) - \delta s(s-1)/2},$$

where $(i)$ uses the fact that $\max_{a \in \mathcal{A}} P(\boldsymbol{t}_{1:i}) \leq \prod_{j=1}^{i} \max_{a \in \mathcal{A}} P(\boldsymbol{t}_j|a) \leq (1 - \delta)^i$. By union bounding across the $|\mathcal{A}|^s$ strings of length $s$,

$$P(\text{any length } s \text{ string is assigned as a token}) \leq e^{s \log(|\mathcal{A}|) + s \log(d) - \delta s(s-1)/2}.$$

When $s = 4 \log(d|\mathcal{A}|)/\delta + 1 \triangleq \ell_{\max} + 1$, the RHS is upper bounded by $e^{-\delta \ell_{\max}^2/4} \leq d^{-\Omega(\log(d/\delta)/\delta)}$. With the same small probability, no substring of length $s' > s$ can become a token, since their length-$s$ prefixes are never assigned as tokens. $\square$

**Corollary A.9.** *With probability $\geq 1 - d^{-\Omega_\delta(\log(d))}$, learns a dictionary with at least $d^\star = d/\ell_{\max}$ tokens when run on a training sequence of length $n$ drawn from a stochastic source satisfying Assumption 3.2.*

**Lemma A.10.** *For any constant $\beta < 1$, with probability $\geq 1 - d^{-\Omega(\log(d/\delta)/\delta)} - \exp(-\widetilde{\Omega}_\delta(d^{1-\beta}))$ over the source dataset, every substring in $\mathcal{M}_\beta$ is added as a token to the dictionary in a run of the LZW algorithm. In other words, with the same probability, the LZW tokenizer results in a $\beta$-heavy hitting dictionary.*

By Corollary A.9, note that with high probability the LZW tokenizer adds at least $d^\star$ tokens to the dictionary when processing a length $d$ training sequence in entirety. In this proof, instead of generating $d$ samples, we sequentially sample $d^\star$ tokens from their joint distribution, and generate a dictionary from these samples. From Corollary A.9, with high probability this results in at most $d$ samples being generated, implying that the dictionary generated by sampling $d^\star$ tokens is a subset of the dictionary generated by a full run of the LZW tokenizer. Here, we use the fact that the LZW tokenizer adds tokens to the dictionary in a left to right fashion, and therefore a subset of the dictionary learnt by the LZW tokenizer can be generated by processing a portion of the dataset.

Next we consider a joint view for generating the dataset from the stochastic source and the dictionary learnt by LZW simultaneously. The stochastic source is sampled as a sequence of tokens. Suppose the last character of the previous token was $a'$. Sample a character $a \sim P(\cdot|a')$ and an infinite trajectory on the tree $\mathcal{T}_a^\star$. Consider the first node visited in this trajectory which does not already

exist as a token in the dictionary. The substring corresponding to this node is added as a token in the dictionary. By repeating this process, the dictionary and the source dataset are constructed sequentially and simultaneously. As alluded to before, we truncate this token sampling process to repeat at most $d^\star$ times, which results in a subset of the dictionary output by the LZW algorithm with high probability (Corollary A.9). This is simply a variant of the "Poissonization" trick to avoid statistical dependencies across tokens. Denote the set of infinite trajectories generated on the forest $\{\mathcal{T}_a^\star : a \in \mathcal{A}\}$ as $\{\mathsf{traj}_i : i \in [d^\star]\}$.

With this view of the sampling process, observe that if the substring $t$ sampled was a prefix of $\mathsf{traj}_i$ at least $|t|$ times across different values of $i$, then $t$ must be assigned as a token. In particular, in each of these $|t|$ trajectories, each of the prefixes of $t$ is assigned as a token. With this observation, the event that $t$ is not assigned as a token is contained in the event that $t$ is visited at most $|t| - 1$ times across the $d^\star$ trajectories. Observe that,

$$P(t \text{ is not assigned as a token}) \leq \sum_{i=0}^{|t|-1} \binom{d^\star}{i} \max_{a \in \mathcal{A}} (P(t|a))^i (1 - P(t|a))^{d^\star - i}.$$

Since we aim to upper bound this probability across the substrings in $t \in \mathcal{M}_\beta$, note that $(i)$ $\max_{a \in \mathcal{A}} P(t|a) \geq \delta/d^\beta$, and $(ii)$ tokens in $\mathcal{M}_\beta$ have length at most $\ell_\star = \delta^{-1}(\beta \log(d) + \log(1/\delta))$ (Lemma A.7), implying there are at most $2|\mathcal{A}|^{\ell_\star}$ substrings in this set. By union bounding,

$$P(\exists t \in \mathcal{M}_\beta \text{ not assigned as a token}) \leq 2|\mathcal{A}|^{\ell_\star} \sum_{i=0}^{\ell_\star - 1} \binom{d^\star}{i} \max_{x \geq \delta/d^\beta} x^i (1 - x)^{d^\star - i}. \quad (11)$$

**Case I.** For $i \leq \ell_\star$ and $x \geq 1/2$,

$$|\mathcal{A}|^{\ell_\star} \binom{d^\star}{i} x^i (1 - x)^{d^\star - i} \leq |\mathcal{A}|^{\ell_\star} \frac{(d^\star)^{\ell_\star}}{2^{d^\star/2}}$$

$$\leq 2^{\ell_\star \log(d^\star |\mathcal{A}|) - d^\star/2}$$

$$\leq 2^{-\Omega_{\beta,\delta}(d^\star)}, \quad (12)$$

where the last inequality uses the fact that $\ell_\star = O_{\beta,\delta}(\log(d))$.

**Case II.** For $i \leq \ell_\star$ and $\delta/d^\beta \leq x \leq 1/2$,

$$|\mathcal{A}|^{\ell_\star} \binom{d^\star}{i} x^i (1 - x)^{d^\star - i} \leq |\mathcal{A}|^{\ell_\star} \binom{d^\star}{i} (1 - x)^{d^\star}$$

$$\leq |\mathcal{A}|^{\ell_\star} (d^\star)^{\ell_\star} e^{-d^\star x}$$

$$\leq e^{\ell_\star \log(|\mathcal{A}|) + \ell_\star \log(d^\star) - d^\star x}$$

$$\leq e^{-\Omega(\delta^2 n/d^\beta / \log(d/\delta))}$$

$$\leq e^{-\Omega(\delta^2 d^{1-\beta} / \log(d/\delta))}, \quad (13)$$

where the last inequality uses the fact that $\ell_\star = O(\log(d))$, $x \geq \delta/d^\beta$, $d^\star = \Omega(d\delta/\log(d/\delta))$. By combining eq. (12) and eq. (13) with eq. (11) completes the proof, as long as $\beta$ is a constant bounded away from 1.

**Lemma A.11.** *Fix a constant $\gamma > 0$. Then, with probability $\geq 1 - d^{-\Omega_{\gamma,\delta}(\log(d))}$, none of the substrings in the set $\mathcal{N}_\gamma = \{\mathbf{s}' : \max_{a \in \mathcal{A}} P(\mathbf{s}'|a) \leq \delta/d^{1+\gamma}\}$ are assigned as tokens in a run of LZW.*

*Proof.* Define the following set of substrings,

$$S_\gamma = \left\{ \mathbf{t} : \delta/d^{1+\gamma/2} \leq \max_{a \in \mathcal{A}} P(\mathbf{t}|a) \leq 1/d^{1+\gamma/2} \right\}$$

Since the width of this band is sufficiently large, by Assumption 3.2 every substring $t$ such that $\max_{a \in \mathcal{A}} P(t|a) \leq \delta/d^{1+\gamma/2}$ has at least one prefix which falls in $S_\gamma$, and denote the longest such

prefix $\boldsymbol{t}_\gamma$. Define $T_\gamma = \{\boldsymbol{t}_\gamma : \boldsymbol{t} \in \mathcal{N}_\gamma\}$ as the set of longest prefixes in $S_\gamma$. Intuitively, if we think of the strings in $S_\gamma$ (or $T_\gamma$) as being intermediate in length, the strings in $\mathcal{N}_\gamma$ can be thought of as being particularly long: the value of $\max_{a \in \mathcal{A}} P(\boldsymbol{t}|a)$ for any $\boldsymbol{t} \in T_\gamma$ and for any $\boldsymbol{t} \in \mathcal{N}_\gamma$ are separated by a factor of at least $1/d^{\gamma/2}$. In particular, since the probability of any character is lower bounded by $\delta$, each substring in $\boldsymbol{t} \in \mathcal{N}_\gamma$ must be at least $\Delta = \frac{\gamma \log(d)}{2 \log(1/\delta)}$ symbols longer than its corresponding longest prefix in $T_\gamma$, $\boldsymbol{t}_\gamma$. An implication of this is that for $\boldsymbol{t}$ to be assigned as a token, $\boldsymbol{t}_\gamma$ must be observed at least $\Delta + 1$ times disjointly in $\boldsymbol{s}$. However, note that $\boldsymbol{t}_\gamma$ already has low marginal probability to begin with ($\ll 1/d$) so the odds of seeing this substring so many times disjointly is very small. Furthermore, note that $T_\gamma$ has at most $d^{1+\gamma/2}/\delta$ substrings, which allows the probability of this event occurring simultaneously across all substrings in $T_\gamma$ to be controlled by union bound. Under this condition, none of the substrings in $\mathcal{N}_\gamma$ are made into tokens.

In order to argue that the dictionary *does not* contain certain tokens, we may argue this property about any superset of the dictionary. In contrast, in Lemma A.10, we construct a subset of the dictionary by running LZW on the concatenation of $d^\star$ tokens sampled from their joint distribution. The superset we consider here is just to sample $d$ tokens from their joint distribution and concatenate them together to result in a string of length $\geq d$, and running LZW on this sequence (which simply would result in these $d$ tokens). As in Lemma A.10, let $\{\mathsf{traj}_i : i \in [d]\}$ denote the infinite trajectories generated from the Markov chain which are truncated to result in tokens. A sufficient condition for the event that no substring $\boldsymbol{t} \in \mathcal{N}_\gamma$ is assigned as a token by LZW is to the event that every substring $\boldsymbol{t}' \in T_\gamma$ is observed as a prefix of $\mathsf{traj}_i$ for $\Delta$ or fewer choices of $i \in [d]$. To this end define $\mathcal{E}(\boldsymbol{t}')$ as the event that $|i \in [d] : \boldsymbol{t}'$ is a prefix of $\mathsf{traj}_i| \leq \Delta$. Then,

$$
\begin{aligned}
\Pr(\mathcal{E}(\boldsymbol{t}')) &\leq \binom{n}{\Delta} (\max_{a \in \mathcal{A}} P(\boldsymbol{t}'|a))^\Delta \\
&\overset{(i)}{\leq} e^{\Delta \log(n)} \left( \frac{1}{d^{1+\gamma/2}} \right)^\Delta \\
&\leq e^{-\frac{\gamma}{2} \Delta \log(d)},
\end{aligned}
\tag{14}
$$

where $(i)$ uses the fact that $\max_{a \in \mathcal{A}} P(\boldsymbol{t}'|a) \leq 1/d^{1+\gamma/2}$ since the substring $\boldsymbol{t}'$ belongs to $T_\gamma$.

Note that the number of substrings in $S_\gamma$ (and by extension, $T_\gamma$) is at most $O_\delta(d^{1+\gamma/2})$. Recall that these substrings satisfy the condition $\max_{a \in \mathcal{A}} P(\boldsymbol{t}|a) \geq \delta/d^{1+\gamma/2}$. Observe that,

$$
\begin{aligned}
\frac{\delta |S_\gamma|}{d^{1+\gamma/2}} &\leq \sum_{\boldsymbol{t} \in S_\gamma} \max_{a \in \mathcal{A}} P(\boldsymbol{t}|a) \\
&\leq \sum_{\boldsymbol{t} \in S_\gamma} \sum_{a \in \mathcal{A}} P(\boldsymbol{t}|a) \leq |\mathcal{A}| \leq \frac{1}{\delta}.
\end{aligned}
$$

This implies that there are at most $d^{1+\gamma/2}/\delta^2$ substrings in $S_\gamma$. Finally, in conjunction with eq. (14),

$$
P(\exists \, \boldsymbol{t}' \in S_\gamma : \mathcal{E}(\boldsymbol{t}')) \leq \frac{d^{1+\gamma/2}}{\delta^2} e^{-\frac{\gamma}{2} \Delta \log(d)} = d^{-\Omega_{\gamma,\delta}(\log(d))},
$$

which implies that with high probability, no token in $S_\gamma$ is observed as a prefix of $\boldsymbol{s}^i$ for more than $\Delta$ choices of the index $i \in [d]$. Under this event, no substring in $\mathcal{N}_\gamma$ is assigned as a token. $\qquad \square$

### A.6.1 Proof of Theorem 3.6

Choosing $\beta = 0.99$ in Lemma A.10, with probability $\geq 1 - d^{-\Omega_\delta(\log(d))}$, the LZW tokenizer results in a 0.99-heavy-hitting dictionary. As a consequence of Lemma A.6, this implies that under the same event,

$$
H(Q_{\mathrm{MLE}}, P) \geq 0.99 \log(d).
$$

Finally, combining with Theorem 3.4 completes the proof.

# B    Additional Theoretical Results I: A sequential variant of BPE

While the main results in the paper focused on understanding the limits of tokenization under a bound on the dictionary size, in this section we take a more practical look and try to analyze tokenizers used commonly in practice. The Byte-Pair-Encoding (BPE) algorithm (Gage, 1994; Sennrich et al., 2016), discovered in the compression literature as REPAIR (Larsson and Moffat, 2000; Navarro and Russo, 2008) was proposed as a faster alternative to LZW. It remains as one of the most commonly implemented tokenizers in natural language processing for various downstream tasks (Radford et al., 2019; Mann et al., 2020; Touvron et al., 2023). A large proportion of open source and commercial LLMs currently use BPE as the tokenization algorithm of choice, such as GPT-2/3, Llama 1/2 and Mistral-7B to name a few.

The BPE algorithm is based on constructing the dictionary iteratively by merging pairs of tokens to result in a tokens. In each iteration, the pair of tokens which appear most frequently next to each other are merged together into a single token. Subsequently, every occurrence of the pair of tokens are replaced by the newly added token, breaking ties arbitrarily. The dictionary is thus an ordered mapping of the form $t \leftarrow (t', t'')$. To encode a new string, the BPE encoder iterates through the dictionary and for each rule $t \leftarrow (t', t'')$ replaces every consecutive occurrence of $t'$ and $t''$ by the token $t$ breaking ties arbitrarily.

To warm up our main results, it is worth understanding the behavior of the BPE tokenizer in a bit more detail. Unlike the toy tokenizer, it is a priori unclear whether unigram models trained on sequences tokenized by BPE even asymptotically (in the dictionary size) achieve the optimal cross-entropy loss. Indeed, for $\delta > 0$, consider a training sequence of length $m$ of the form,

$$s = \left( \underbrace{\underbrace{01 \cdots 01}_{2/\delta} \underbrace{10 \cdots 10}_{2/\delta}} \right) \\ {\scriptstyle \times \frac{m\delta}{4}} \tag{15}$$

The probability that this sequence is generated by the order-2 switching Markov source with $p = q = \delta$ is,

$$\approx (1 - \delta)^{\frac{m\delta}{4} \times \frac{4}{\delta} \times (1 - \delta)} (\delta)^{\frac{m\delta}{4} \times 4} = e^{-H(P)},$$

which uses the fact that $H(P) = m\delta \log(1/\delta) + m(1 - \delta) \log(1/(1 - \delta))$. This implies that even though the string has exponentially small probability, it is one of the typical sequences for this order-2 Markov source. Let's understand what happens when the BPE tokenizer is trained on this dataset. Assuming that ties are broken arbitrarily, consider the run of the BPE algorithm detailed in Table 2. Here, we assume that $1/\delta - 1$ is a power of 2 and denote $r = \log_2(1/\delta - 1)$. The algorithm first merges 0 and 1 into a single token $t_1$, which results in a long sequence of the form $t_1 \cdots \cdots t_1 1 t_1 \cdots \cdots t_1 0$ repeated $m\delta/4$ times. In subsequent rounds, the tokens $(t_1, t_1)$ is merged into $t_2$, then $(t_2, t_2)$ is merged into $t_3$ and so on, until is no longer possible. Finally, the resulting sequence is a repeating sequence of 5 tokens where within each sequence, no pair of tokens appears more than once next to each other. Eventually these 5 tokens are merged into a single token labelled $t_{r+4}$, and in subsequent rounds the tokens $(t_{r+4}, t_{r+4})$ are merged into $t_{r+5}$, $(t_{r+5}, t_{r+5})$ is merged into $t_{r+6}$ and so on, until is no longer possible.

Observe that in the initial training dataset the substrings 0000 and 1111 never appears as a contiguous sequence. However, in a test sequence of length $m$ sampled from the $2^{\text{nd}}$-order Markov source, with high probability these substrings disjointly occur $\Theta(m)$ times each. The learnt dictionary associates each such disjoint occurrence of these substrings with at least 1 token, for 0000, the $3^{\text{rd}}$ 0 must necessarily be tokenized as the token "0". Likewise, in 1111, the $3^{\text{rd}}$ 1 must necessarily be tokenized as the token "1". Therefore, when a new test string of length $m$ is tokenized, with high probability the tokens "0" and "1" form a constant fraction of the total collection of tokens.

Thus on freshly sampled test sequences, the BPE tokenizer appears to behave like the character-level tokenizer on a constant fraction of the input sequence. In particular, a simple calculation shows that the cross-entropy loss of any unigram model trained on this tokenizer must be far from the optimal

| Initial | $01\cdots\cdots 0110\cdots\cdots 10|\cdots$ |
|:---:|:---:|
| $t_1 \leftarrow (0,1)$ | $t_1\cdots\cdots t_1 1 t_1 \cdots\cdots t_1 0|\cdots$ |
| $t_2 \leftarrow (t_1, t_1)$ | $t_2\cdots t_2 1 t_2 \cdots t_2 0|\cdots$ |
| $\vdots$ | $\vdots$ |
| $t_r \leftarrow (t_{r-1}, t_{r-1})$ | $t_r t_1 1 t_r 0|\cdots$ |
| $t_{r+1} \leftarrow (t_r, t_1)$ | $t_{r+1} 1 t_r 0|\cdots$ |
| $t_{r+2} \leftarrow (t_r, 0)$ | $t_{r+1} 1 t_{r+2}|\cdots$ |
| $t_{r+3} \leftarrow (t_{r+1}, 1)$ | $t_{r+3} t_{r+2}|\cdots$ |
| $t_{r+4} \leftarrow (t_{r+3}, t_{r+2})$ | $t_{r+4}|\cdots$ |
| $t_{r+5} \leftarrow (t_{r+4}, t_{r+4})$ | $t_{r+5}|\cdots$ |
| $t_{r+6} \leftarrow (t_{r+5}, t_{r+5})$ | $t_{r+6}|\cdots.$ |
| $\vdots$ | $\vdots$ |

Table 2: A representation of the behavior of BPE when trained on the dataset in eq. (15). We assume that $1/\delta - 1$ is a power of 2 and define $r = \log_2(1/\delta - 1)$.

bound of $mH_{\text{BER}}(\delta)$ especially as $\delta$ becomes smaller,

$$\min_{Q\in\mathcal{Q}_{\text{1-gram}}} \mathcal{L}_m(Q \circ \mathsf{enc}(\cdot))$$
$$\geq \min_{Q\in\mathcal{Q}_{\text{1-gram}}} \mathbb{E}\left[n_0 \log(1/Q_{\text{tok}}(0) + n_1 \log(1/Q_{\text{tok}}(1)\right]$$
$$\overset{(i)}{\geq} \Omega(m) \cdot \min_{Q\in\mathcal{Q}_{\text{1-gram}}} \left(\log(1/P_{\text{tok}}(0) + \log(1/Q_{\text{tok}}(1)\right)$$
$$\geq \Omega(m).$$

where $(i)$ uses the fact that $\mathbb{E}[n_0], \mathbb{E}[n_1] \in \Omega(m)$ and the last inequality uses $P_{\text{tok}}(0)P_{\text{tok}}(1) \leq 1/4$ (AM-GM inequality) since they sum up to at most 1. The purpose of this example is to show that there exist pathological training datasets which appear to be drawn from a stochastic source, but on which BPE fails to learn a good dictionary for the source. Thus proving a result such as Theorem 3.1 for BPE would require arguing that training datasets such as that in eq. (15) are unlikely to be seen.

The analysis of the standard variant of BPE turns out to be complicated for other reasons too. After every token is added the training dataset becomes a mix of all the previously added tokens, and arguing about the statistics of which pair of tokens appears most frequently for the next iteration becomes involved. For instance, adding $00$ as a token may reduce the frequency of occurrence of the substring $01$, but will not affect $11$. Thus, even though $01$ may a priori have been seen more frequently, it may not be chosen by BPE as the next token after $00$.

To avoid dealing with these issues, we consider a sequential/sample-splitting variant of BPE. At a high level, the algorithm breaks down a dataset of size $\Theta(d^2)$ into $d$ chunks and learns at most 1 token from each chunk. The algorithm iterates over the chunks and finding the pair of tokens which appear most frequently next to each other in each chunk and adding it to the dictionary if it appears more than $\log(d)$ times. Every consecutive occurrence of the pair of tokens is replaced by the newly assigned token in the dataset. Thus, in each iteration $i$, at most 1 token is added, depending on the statistics of the $i^{\text{th}}$ chunk and the tokens added so far to the dictionary. Based on the final size of the dictionary a different encoder/decoder pair is used - if the algorithm adds sufficiently many tokens to the dictionary, the greedy encoder is used, and if not, a parallel implementation of BPE's encoding algorithm is used (Definition B.1). A formal description of the algorithm is in Algorithm 1.

**Definition B.1** (BPE.split encoder). The BPE.split encoder parses a new string into tokens as follows. The algorithm partitions the string into contiguous chunks of length $d$. Then, BPE's encoder is applied on each chunk, which iterates through $\mathsf{DS}$ and replaces $t't''$ by $t$ for every rule $t \leftarrow (t', t'')$ in $\mathsf{DS}$, breaking ties arbitrarily. The individual token sequences are finally spliced together and returned.

The main result of this section is that up to a small additive error, Algorithm 1 approaches a 2-approximation to the optimal cross-entropy loss.

---

**Algorithm 1** Sequential implementation of BPE

---

**Input:** $\epsilon \in (0,1)$; a dataset of size $n = \Theta(d^2)$, split into $d$ contiguous texts $\{\text{text}_1, \cdots, \text{text}_d\}$ of length $\Theta(d)$ each.
**Output:** A tokenizer $\mathcal{T}$.
// Generate Dictionary
**for** $i = 1, \cdots, d$ **do**
    **if** $\exists$ a pair of tokens/characters $(\boldsymbol{t}', \boldsymbol{t}'')$ appearing $\geq \log(d)$ times consecutively in $\text{text}_i$ **then**
        Append the rule $\boldsymbol{t} \leftarrow (\boldsymbol{t}', \boldsymbol{t}'')$ to DS
        **for** $j = i+1, \cdots, d$ **do**
            $\text{text}_j \leftarrow \text{APPLY}_{\boldsymbol{t} \leftarrow (\boldsymbol{t}', \boldsymbol{t}'')}(\text{text}_j)$;
// Can be implemented in parallel
        **end for**
    **end if**
**end for**

// Encoder and Decoder
**if** $|\text{Dict}| < d_0 \triangleq \epsilon d / 2 \log(4|\mathcal{A}|)$ **then**
    $\mathcal{T} \leftarrow (\text{Dict}, \text{DS}, \text{enc}_{\text{BPE.split}}(\cdot), \text{dec}_{\text{BPE.split}}(\cdot))$
**else**
    $\mathcal{T} \leftarrow (\text{Dict}, \emptyset, \text{enc}_{\text{gre}}(\cdot), \text{dec}_{\text{gre}}(\cdot))$
**end if**

**def** $\text{APPLY}_{\boldsymbol{t} \leftarrow (\boldsymbol{t}_1, \boldsymbol{t}_2)}(\text{text})$:
Replace every consecutive occurrence of $(\boldsymbol{t}', \boldsymbol{t}'')$ in text by $\boldsymbol{t}$, breaking ties arbitrarily.

---

**Theorem B.2.** *For any $\epsilon \in (0,1)$, run Algorithm 1 on a dataset of $n = \Theta(d^2)$ characters to learn a dictionary with at most $d$ tokens. The resulting tokenizer $\mathcal{T}$ satisfies with probability $\geq 1 - e^{-\Omega(d\epsilon^2)}$,*

$$\min_{Q \in \mathcal{Q}_{\text{1-gram}}} \mathcal{L}(Q \circ \boldsymbol{enc}(\cdot)) \leq (2 + \varepsilon) \min_Q \mathcal{L}(Q) + \epsilon$$

*where $\varepsilon = O\left(\frac{\log(1/\epsilon) \log^3(1/\delta)}{\epsilon \delta^9 \log(d)}\right)$.*

While the guarantees established for the sequential BPE tokenizer are weaker than those in Theorem 3.1, the analysis turns out to be quite involved. Theorem B.2 implies that unigram models trained on the sequential BPE tokenizer asymptotically approach the optimal cross-entropy loss up to a factor of 2.

The formal proof of this result is presented in Appendix B. What is the intuition behind using a different encoder in Algorithm 1 depending on the number of tokens in the dictionary? When the number of tokens in the dictionary is smaller than $d_0$, we know that on a $1 - d_0/d$ fraction of the iterations of Algorithm 1, a token is *not* added to the dictionary, i.e. every pair of tokens already appears at most $\log(d)$ times together. This is a datapoint of "evidence" that under the dictionary in that iteration, the BPE encoder is already good at encoding new strings (of length $\Theta(d)$) in a way where pairs of tokens do not appear consecutively with high frequency. Since future dictionaries only have more rules appended to them, dictionaries only get better at encoding new strings into tokens where pairs do not frequently appear consecutively. In other words, the BPE encoder satisfies a monotonicity property. It remains to show that dictionaries which encode new sequences in a way where no pair of tokens appear too frequently have large $H(Q_{\text{MLE}}, P)$ (to invoke Theorem 3.4). This follows from ideas introduced in (Navarro and Russo, 2008).

The case where the number of tokens is large ($\geq d_0$) turns out to present significant technical challenges for analyzing the BPE encoder. There is no longer much "evidence" that the dictionary in each iteration is good at encoding strings since in a large number of iterations a pair of tokens appear consecutively with high frequency. Analyzing the greedy encoder also presents its own challenges - although the algorithm has allocated a large number of tokens, it is possible that there are short tokens $\boldsymbol{t}$ which are maximal (i.e. they are not prefixes of other tokens). This is similar to the problem encountered by BPE when trained on the dataset in eq. (15) - although the algorithm has allocated a large number of tokens, the token 1 is maximal since every other token begins with the character 0.

However, it turns out that such tokens, although present in the dictionary, are not observed frequently while encoding a fresh string drawn from the source.

## B.1 Analysis of Algorithm 1

In this section we prove a rephrased version of Theorem B.2 which implies the statement in the main paper. Define $d_0 = \frac{\epsilon d}{2 \log(4|\mathcal{A}|)}$.

**Theorem B.3** (Rephrased Theorem B.2). *Run Algorithm 1 on a dataset of $n = \Theta(d^2)$ characters to learn a dictionary with at most $d$ tokens. The resulting tokenizer $\mathcal{T}$ satisfies one of the following 3 conditions,*

1. *Either,* $|\mathsf{Dict}| > d_0$, *and,*

$$\min_{Q \in \mathcal{Q}_{1\text{-}gram}} \mathcal{L}(Q \circ enc(\cdot)) \leq \frac{H_\infty}{1 - \varepsilon}.$$

   *Here,* $\varepsilon = O\left(\frac{\log^3(1/\delta)\log(1/\epsilon)}{\epsilon \delta^9 \log(d)}\right)$.

2. $\Pr(|\mathsf{Dict}| < d_0) = e^{-\Omega(\epsilon^2 d/\log^2(1/\delta))}$, *or,*

3. *Conditional on* $|\mathsf{Dict}| < d_0$, *with probability* $\geq 1 - e^{-\Omega(\epsilon^2 d/\log^2(1/\delta))}$,

$$\min_{Q \in \mathcal{Q}_{1\text{-}gram}} \mathcal{L}(Q \circ enc(\cdot)) \leq \left(1 - \frac{2d_0}{d}\right)\left(2H_\infty + O\left(\frac{1}{\log(d)}\right)\right) + \frac{2d_0}{d}\log(4|\mathcal{A}|).$$

   *With the choice of $d_0 = \epsilon d/2\log(4|\mathcal{A}|)$ we get the statement of Theorem B.2.*

## B.2 Analysis for the large dictionary case: $|\mathsf{Dict}| > d_0$

In the large dictionary case, Algorithm 1 uses the greedy encoder/decoder pair in conjunction with the dictionary. The proof of Theorem B.2 relies on establishing that the cross-entropy $H(Q_{\mathrm{MLE}}, P)$ of the tokenizer is large. Namely, we prove that,

**Lemma B.4.** *In Algorithm 1, assuming at least $d_0$ tokens are allocated,*

$$H(Q_{MLE}, P) = \Omega\left(\frac{\epsilon \delta^9 \log(d)}{\log(1/\epsilon)\log^3(1/\delta)}\right).$$

To show this, it suffices to argue that conditioned on any previous set of tokens, with nontrivial probability over the underlying string generated from the stochastic source, the next token is long (i.e. having conditional probability at most $O(1/\sqrt{d})$).

**Lemma B.5.** *Suppose that in a run of Algorithm 1, at least $d_0$ tokens are allocated. Suppose a set of tokens $\boldsymbol{t}_1, \cdots, \boldsymbol{t}_k$ have been sampled so far by the greedy encoder. Let $T_{i+1}$ be the random variable which denotes the next token returned by the greedy encoder, where the randomness comes from the underlying string being tokenized. Then,*

$$\Pr\left(P(T_{i+1}|\boldsymbol{t}_i) \leq 1/\sqrt{C\delta d} \,\Big|\, \boldsymbol{t}_1, \cdots, \boldsymbol{t}_i\right) \geq \frac{d_0 \delta^6 (1-\delta)^2}{8Cd\Delta|\mathcal{A}|\log(2|\mathcal{A}|)n_D} = \Omega\left(\frac{\epsilon \delta^9}{\log^3(1/\delta)\log(1/\epsilon)}\right)$$

*Proof sketch of Lemma B.5.* The proof will proceed in 2 parts. We first show in Lemma B.9 that there is a set $D_{\mathrm{valid}}$ of $\Omega(d)$ tokens in the dictionary which are neither prefixes nor suffixes of any other token in $\mathsf{Dict}$. The reason for considering this set of tokens is twofold,

1. Irrespective of what the previous set of tokens were, it is legal for a token $D_{\mathrm{valid}}$ to be sampled in the current step by the greedy encoder, since for any candidate $\boldsymbol{t} \in D_{\mathrm{valid}}$, by definition, $\boldsymbol{t}_j \cdots \boldsymbol{t}_i \boldsymbol{t} \notin \mathsf{Dict}$ for every $j \leq i$.

2. Suppose a sequence of tokens $t_1, \cdots, t_i$ have already been sampled, ending with the character $a$. Then, we may sample the next token using rejection sampling. Sample $a' \sim P(\cdot|a)$ and an infinitely long trajectory on $\mathcal{T}_{a'}^\star$. Return the last token on this trajectory which belongs to Dict, and if it so happens that $\exists j \in [i]$ such that $t_j \cdots t_i t \in$ Dict, then reject this trajectory and repeat. Since all the tokens in $D_{\text{valid}}$ are not prefixes of another token, any trajectory which reaches a token in $D_{\text{valid}}$ must terminate the rejection sampling process.

Next, in Lemma B.10, we show that since the number of tokens in $D_{\text{valid}}$ is sufficiently large, $\Omega(d)$, with constant (in $d$) probability, a trajectory rolled out in the first round of the rejection sampling process will reach a token $t \in D_{\text{valid}}$ which has small probability, i.e. $\max_{a \in \mathcal{A}} P(t|a) \leq 1/\text{poly}(d)$. By the previous arguments, this must mean that the rejection sampling process terminates on this "low probability" token, resulting in the statement of the lemma. $\qquad \square$

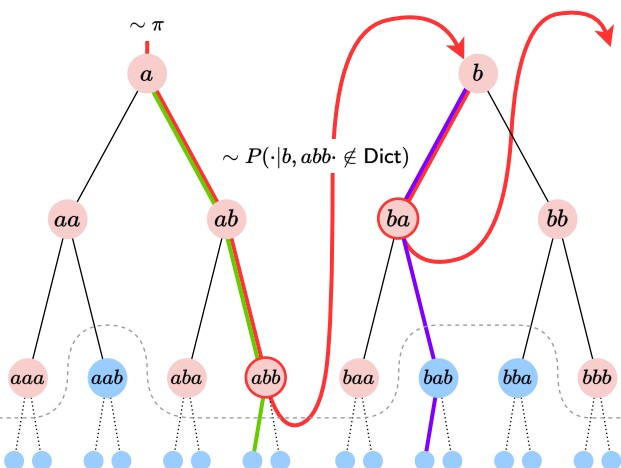

Figure 7: *Jointly generating a sequence and its greedy encoding:* In this example we use the greedy encoder under the dictionary composed of all the substrings shadowed red. The first character ($a$) is sampled from the stationary distribution. Then an infinite string is sampled on the tree with $a$ as root (green path). The last substring on this path which is a token ($t_1 = abb$) is returned by the greedy encoder. Then the next character $x = b$ is sampled from the source conditioned on the previous character ($b$) and further conditioned on $t_1 x \notin$ Dict. Finally, another infinite string is sampled on the tree with $x = b$ as root (purple path) and the last substring on this path which is a token ($t_2 = ba$) is returned by the greedy encoder. Repeating this process, we can generate a string, here, $abbba \cdots$, as well as its greedy encoding, $(abb, ba, \cdots)$.

**Proof of Theorem B.3.1 and Lemma B.4** It is easy to see why Lemma B.5 implies a lower bound on the cross entropy $H(Q_{\text{MLE}}, P)$ of the tokenizer. By Lemma A.4 for the greedy encoder,

$$Q_{\text{MLE}}(t) = \lim_{m \to \infty} \mathbb{E}\left[\frac{n_t}{\sum_{t'} n_{t'}}\right] \overset{\text{a.s.}}{=} \lim_{m \to \infty} \frac{n_t}{\sum_{t'} n_{t'}}. \qquad (16)$$

Since the limit $m \to \infty$ of the RHS exists by Lemma A.4, we may let $m \to \infty$ in any way we like, and in particular we may simply sample $i^\star$ tokens, $t_1, \cdots, t_{i^\star}$ sequentially according to the process in Figure 7. Here, the first token sampled is returned by generating an infinitely long string on $\mathcal{T}_a^\star$ where $a \sim \pi$ and then truncating this trajectory to the longest token which belongs to Dict. Subsequently for every $i > 1$, $t_i$ is generated by sampling a fresh infinitely long string from $\mathcal{T}_a^\star$ where $a$ is sampled from the $P(\cdot|a')$ where $a'$ is the last character $t_{i-1}$ and then returning the largest prefix of this string which is a token in Dict, conditioned on $t_j \cdots t_{i-1} t_i \notin$ Dict for any $j < i$.

and concatenate the corresponding substrings to get an $m = \sum_{i=1}^{i^\star} |\boldsymbol{t}_i|$ length character string. Letting $i^\star \to \infty$, we must have $m \to \infty$ surely since $m \geq i^\star$. In this view, eq. (16) can be rewritten as,

$$Q_{\mathrm{MLE}}(\boldsymbol{t}) = \lim_{i^\star \to \infty} \frac{n_{\boldsymbol{t}}}{i^\star} = \lim_{i^\star \to \infty} \frac{1}{i^\star} \sum_{i=1}^{i^\star} \mathbb{I}(\boldsymbol{t}_i = \boldsymbol{t}) \stackrel{\text{a.s.}}{=} \lim_{i^\star \to \infty} \frac{1}{i^\star} \sum_{i=1}^{i^\star} \mathbb{E}[\mathbb{I}(\boldsymbol{t}_i = \boldsymbol{t})|\boldsymbol{t}_1, \cdots, \boldsymbol{t}_{i-1}] \quad (17)$$

where the last inequality follows by the sequential nature of the token sampling process and a martingale argument. Consider the set of tokens $T$ such that $\boldsymbol{t} \in T$ satisfies $\max_{a \in \mathcal{A}} P(\boldsymbol{t}|a) \leq \sqrt{1/C\delta^3 d}$. From eq. (17), summing across $\boldsymbol{t} \in T$, we have that,

$$\sum_{\boldsymbol{t} \in T} Q_{\mathrm{MLE}}(\boldsymbol{t}) \stackrel{\text{a.s.}}{=} \lim_{i^\star \to \infty} \frac{1}{i^\star} \sum_{i=1}^{i^\star} \Pr\left(\boldsymbol{t}_i \in T|\boldsymbol{t}_1, \cdots, \boldsymbol{t}_{i-1}\right) = \Omega\left(\frac{\epsilon \delta^9}{\log^3(1/\delta)\log(1/\epsilon)}\right) \quad (18)$$

where in the last inequality, we use Lemma B.5 and the fact that $\delta \max_{a \in \mathcal{A}} P(\boldsymbol{t}|a) \geq \min_{a \in \mathcal{A}} P(\boldsymbol{t}|a)$. Therefore,

$$H(Q_{\mathrm{MLE}}, P) \geq \sum_{\boldsymbol{t} \in T} Q_{\mathrm{MLE}}(\boldsymbol{t}) \log(1/P(\boldsymbol{t})) \geq \sum_{\boldsymbol{t} \in T} Q_{\mathrm{MLE}}(\boldsymbol{t}) \log(\sqrt{C\delta^3 d})$$

where in $(i)$ we use the fact that for $\boldsymbol{t} \in T$, $\max_{a \in \mathcal{A}} P(\boldsymbol{t}|a) \leq 1/\sqrt{C\delta^3 d}$, which implies that $P(\boldsymbol{t}) \leq 1/\sqrt{C\delta^3 d}$. Finally, combining with eq. (18) completes the proof of Lemma B.4. Furthermore, since the cross-entropy $H(Q_{\mathrm{MLE}}, P)$ was established to be large, by invoking the reduction in Theorem 3.4, we complete the proof of Theorem B.3.1.

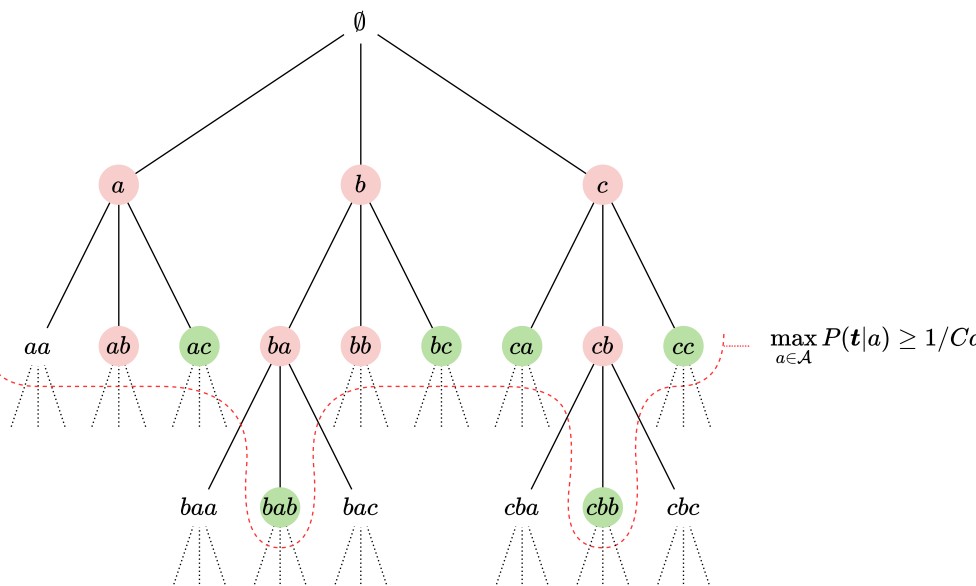

Figure 8: The circled nodes indicate substrings which are tokens in $\mathsf{Dict}$. The red boundary is the set of substrings $\boldsymbol{t}$ such that $\max_{a \in \mathcal{A}} P(\boldsymbol{t}|a) \geq 1/Cd$. By Lemma B.8, none of the nodes which fall outside this boundary are assigned as tokens in a run of Algorithm 1. The set of circled substrings are the set of tokens in $\mathsf{Dict}$. Among them, the ones circled green are the tokens in $D_{\mathrm{valid}}$, which are not prefixes or suffixes of any other tokens in $\mathsf{Dict}$. Substrings such as $cb$ or $ba$ which are tokens in $\mathsf{Dict}$ do not belong to $D_{\mathrm{valid}}$ because they are prefixes of longer tokens (in this case, $cbb$ and $bab$ respectively). On the other hand, substrings like $ab$ do not belong to $D_{\mathrm{valid}}$ since they are suffixes of tokens in $\mathsf{Dict}$, in this case, $bab$. Lemma B.9 asserts that the number of tokens in $D_{\mathrm{valid}}$ are $\Omega(d)$ in number, assuming that $\mathsf{Dict}$ has $\Omega(d)$ tokens to begin with.

**Notation.** For each $a \in \mathcal{A}$ and $j \in \mathbb{N} \cup \{0\}$, define a *level set* of substrings,

$$S_j^a = \left\{(1-\delta)^{j+1} < P(\boldsymbol{t}|\boldsymbol{t}_1 = a) \leq (1-\delta)^j\right\}$$

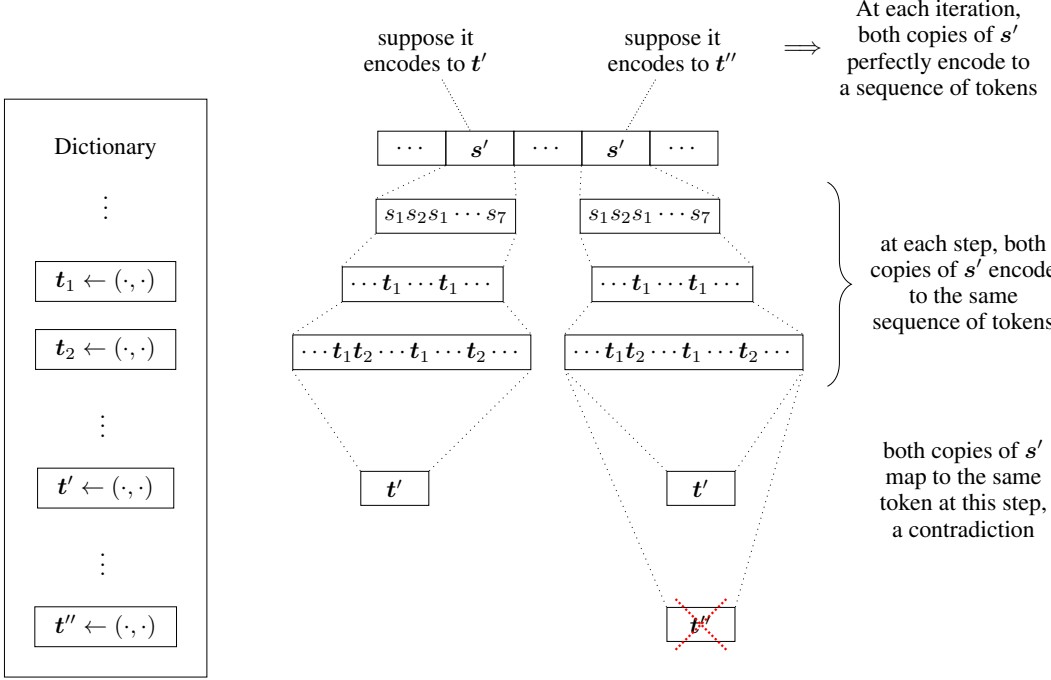

Figure 9: A pictorial representation of the proof of Lemma B.6

where $t_1$ denotes the first character of $t$. And likewise, define the sets $S_j = \cup_{a \in \mathcal{A}} S_j^a$, $S_{\leq j}^a$ and $S_{\geq j}^a$ as the union of $S_{j'}^a$ over $j' \geq j$, $j' \leq j$ and $S_{\leq j}$ and $S_{\geq j}$ as the union of $S_{\leq j}^a$ and $S_{\geq j}^a$ over $a \in \mathcal{A}$. Furthermore for a large universal constant $C > 0$, define parameters,

$$\Delta = \frac{\log(\delta)}{\log(1-\delta)} \asymp \Theta\left(\frac{\log(1/\delta)}{\delta}\right); \quad n_D = 1 - \frac{2\log(4Cd/\delta d_0)}{\log(1-\delta)} \asymp \Theta\left(\frac{\log(1/\epsilon\delta)}{\delta}\right). \quad (19)$$

We first begin by stating a folklore result: every pair of tokens assigned by a merging-based dictionary generation algorithm have distinct character representations.

**Lemma B.6.** *If Algorithm 1 assigns a new token in some round, it's character representation must be distinct from that of all previously assigned tokens.*

*Proof.* A pictorial proof is in Figure 9. We will prove this result by contradiction. Suppose $t$ and $t'$ are tokens which decode to the same character substring, $s'$. Consider all occurrences of $s'$ in the dataset which in some iteration encode into $t'$ or $t''$, and denote these disjoint locations $\mathcal{S}$. Recall that at these locations, $s'$ eventually is assumed to map to a singular token $t'$ or $t''$. Therefore, at every step in the merging process these occurrences of $s'$ must perfectly map to a sequence of tokens.

Now consider the merging process at the first time before any of the rules corresponding to tokens in $t'$ or $t''$ are implemented. Prior to this time, all the occurrences of $s'$ corresponding to the locations in $\mathcal{S}$ have not been tokenized yet. When the first rule corresponding to one of the tokens in $\{t', t''\}$ is implemented, all the strings in $\mathcal{S}$ must be modified identically. This uses the fact that we can isolate each of these occurrences of $s'$ while carrying out the merging process, since each location must be distinct. At every step, the encodings of these copies of $s'$ must be the same, and therefore $t'$ and $t''$ cannot be two distinct tokens. □

**Lemma B.7.** *The size of the level set $S_j^a$ is bounded by $(1-\delta)^{-(j+1)}$.*

*Proof.* Since the probability of any transition is at most $1 - \delta$, this implies that any infinite trajectory on the tree $\mathcal{T}_a^\star$ can intersect at most one vertex in $S_j^a$. Therefore, $\sum_{t \in S_j^a} P(t|t_1 = a) \leq 1$. By the lower bound on $P(t|t_1 = a)$ for $t \in S_j^a$, this implies the statement of the lemma. □

Next we show that with high probability none of the substrings $t$ having probability mass (under $P$) of at most $\delta/Cd$ conditioned on the first character, are assigned as tokens by Algorithm 1.

**Lemma B.8.** *In a run of Algorithm 1, for a sufficiently large constant $C > 0$, with probability $d^{-\Omega(1)}\mathsf{poly}(1/\delta)$ all assigned tokens $t \in \mathsf{Dict}$ satisfy $\max_{a \in \mathcal{A}} P(t|a) \geq 1/Cd$. In other words, none of the substrings in $S_{\geq j^\star}$ are added as tokens to the dictionary in a run of Algorithm 1, where,*

$$j^\star \triangleq \log(\delta/Cd)/\log(1-\delta)$$

*Proof.* Consider some $j \geq j^\star$ and $a \in \mathcal{A}$ and substring $t \in S_j^a$. In the $i^{th}$ stage of the algorithm where $\mathsf{text}_i$ is being processed, for $t$ to be assigned as a token, at the very least, $t$ must appear at least $\log(d)$ times disjointly in $\mathsf{text}_i$. Therefore,

$$P(t \in S_j^a \text{ is assigned as a token in } \mathsf{text}_i) \leq \binom{d}{\log(d)}\left(\max_{a \in \mathcal{A}} P(t|a)\right)^{\log(d)}$$

$$\leq d^{\log(d)}\left(\frac{1}{Cd}(1-\delta)^{j-j^\star}\right)^{\log(d)}$$

$$\leq d^{-\log(C)}(1-\delta)^{(j-j^\star)\log(d)}$$

Union bounding over $S_j^a$ over $j \geq j^\star$ using the bound on $|S_j^a|$ in Lemma B.7, and over $a \in \mathcal{A}$ and $i \in [d]$ results in the bound,

$$P(t \in S_{\geq j^\star} \text{ is assigned as a token in step } i \text{ for some } i \in [d]) \leq d^{-\Omega(1)}\sum_{j \geq j^\star}\frac{(1-\delta)^{(j-j^\star)\log(d)}}{(1-\delta)^{j+1}} \leq \frac{d^{-\Omega(1)}}{\delta(1-\delta)}$$

$\square$

**Lemma B.9.** *Consider the set of tokens $D_{valid}$ which are not a prefix or a suffix of any other token in $\mathsf{Dict}$. That is, $D_{valid} = \{t \in \mathsf{Dict} : \nexists s : st \in \mathsf{Dict}\} \cap \{t \in \mathsf{Dict} : \nexists s : ts \in \mathsf{Dict}\}$. If $|\mathsf{Dict}| \geq d_0$, then,*

$$|D_{valid}| \geq \frac{d_0}{4n_D}.$$

*where $n_D$ is defined in eq. (19).*

*Proof.* For any token $t \in D_{valid}$, there may be at most $2|t|$ tokens which are suffixes or prefixes of it and belong to $\mathsf{Dict}$. More importantly, every token in $\mathsf{Dict}$ not belonging to $D_{valid}$ must either be a prefix or a suffix of some token in $D_{valid}$. Split the suffixes and prefixes of the tokens in $D_{valid}$ into four sets,

1. $S_{\mathrm{suff,min}} = \bigcup_{t \in D_{valid}}\{t' \in \mathsf{Dict} : t' \in \mathsf{suff}(t), |t'| \leq |t| - n_D\}$,

2. $S_{\mathrm{suff,max}} = \bigcup_{t \in D_{valid}}\{t' \in \mathsf{Dict} : t' \in \mathsf{suff}(t), |t'| > |t| - n_D\}$,

3. $S_{\mathrm{pre,min}} = \bigcup_{t \in D_{valid}}\{t' \in \mathsf{Dict} : t' \in \mathsf{pre}(t), |t'| \leq |t| - n_D\}$,

4. $S_{\mathrm{pre,max}} = \bigcup_{t \in D_{valid}}\{t' \in \mathsf{Dict} : t' \in \mathsf{pre}(t), |t'| > |t| - n_D\}$.

where $n_D$ is defined in eq. (19). Note from Lemma B.8 that all the tokens $t \in \mathsf{Dict}$ all satisfy $\max_{a \in \mathcal{A}} P(t|a) \geq 1/Cd$. Therefore, the tokens in $S_{\mathrm{pre,min}}$ and $S_{\mathrm{suff,min}}$ all satisfy, $\max_{a \in \mathcal{A}} P(t|a) \geq d/C(1-\delta)^{n_D}$. By summing Lemma B.7 over appropriate $j$, we get that $|S_{\mathrm{pre,min}}| + |S_{\mathrm{suff,min}}| \leq 2Cd(1-\delta)^{n_D-1}/\delta$.

On the other hand, corresponding to any $t \in D_{valid}$, there are at most $n_D$ tokens in $S_{\mathrm{pre,max}}$ or $S_{\mathrm{suff,max}}$ and and therefore $|S_{\mathrm{pre,max}}|, |S_{\mathrm{suff,max}}| \leq n_D \cdot |D_{valid}|$. Since every token in $\mathsf{Dict}$ either belongs to $D_{valid}$ or is a suffix of some token in $D_{valid}$, $S_{\mathrm{pre,min}} \cup S_{\mathrm{pre,max}} \cup S_{\mathrm{suff,min}} \cup S_{\mathrm{suff,max}} = |\mathsf{Dict}|$ and,

$$2n_D \cdot |D_{valid}| + \frac{2C(1-\delta)^{n_D-1}d}{\delta} \geq d_0$$

Recalling the choice of $n_D = 1 - \frac{2\log(4Cd/\delta d_0)}{\log(1-\delta)}$, we get that,

$$|D_{\text{valid}}| \geq \frac{d_0}{4n_D}.$$

$\square$

**Lemma B.10.** *Suppose Algorithm 1 assigns at least $d_0$ tokens. For any character $a \in \mathcal{A}$, sample an $a' \sim P(\cdot|a)$ and an infinite trajectory on the tree $\mathcal{T}_{a'}^{\star}$, denoted traj. Then,*

$$\mathbb{E}_{a' \sim P(\cdot|a)} \left[ \Pr_{\text{traj} \sim \mathcal{T}_{a'}^{\star}} \left( \min_{t \in \text{traj} \cap D_{\text{valid}}} P(t|a) \leq \sqrt{\delta/Cd} \Big| a' \right) \right] \geq \frac{d_0 \delta^6 (1-\delta)^2}{8Cd\Delta|\mathcal{A}|n_D}.$$

*where the notation $\mathcal{T}_{a'}^{\star}$ is used to overload the distribution over infinite trajectories on $\mathcal{T}_{a'}^{\star}$. The parameters $n_D$ and $\Delta$ are defined in eq. (19).*

*Proof.* By Lemma B.8, recall that the $\geq d_0$ tokens assigned in a run of Algorithm 1, with high probability, are substrings in $S_{\leq j^{\star}}$. For any $a \in \mathcal{A}$, the total number of substrings in $S_{\leq j^{\star}}$ can be bounded as,

$$|S_{\leq j^{\star}}| \leq \sum_{a \in \mathcal{A}} \sum_{j=0}^{j^{\star}} |S_j^a| \leq \sum_{a \in \mathcal{A}} \sum_{j=0}^{j^{\star}} \frac{1}{(1-\delta)^{j+1}} \leq \frac{C|\mathcal{A}|d}{\delta(1-\delta)}. \tag{20}$$

In order to prove this result, we use a counting argument and the fact that no tokens in $S_{>j^{\star}}$ are assigned. Consider some character $a$ and all the leaves in the forest $S_{\leq j^{\star}}$. Since every transition has $\geq \delta$ probability of occurring, across all leaf nodes $t \in S_{\leq j^{\star}}$, $P(t|a')$ are within a $\delta^2(1-\delta)$ factor of each other across different $a' \in \mathcal{A}$. In particular, by counting the number of paths in $\mathcal{T}^{\star}$ (i.e. paths in $\mathcal{T}_a^{\star}$ from $\emptyset$ to leaf nodes in $S_{\leq j^{\star}}^a$ across $a \in \mathcal{A}$) along which a token in Dict exists in $S_{\geq j^{\star}/2}$, we can also compute the probability mass across such trajectories up to a factor of $\delta^2(1-\delta)$.

Taking the union across $a \in \mathcal{A}$, consider the paths in $\mathcal{T}_a^{\star}$ from $\emptyset$ to leaf nodes in $S_{\leq j^{\star}}^a$. From Lemma B.9, $\sum_{j \leq j^{\star}} |D_{\text{valid}} \cap S_j| \geq d_0/4n_D$, where $n_D = 1 - 2\log(4Cd/\delta d_0)/\log(1-\delta)$. Note that for sufficiently large $d = \Omega(\log(1/\epsilon\delta)/\delta^5)$, by Lemma B.7, $\sum_{j \leq j^{\star}/2} |S_j| = \sqrt{Cd/\delta}/\delta(1-\delta) \leq d_0/8n_D$. Therefore,

$$\sum_{j^{\star}/2 < j \leq j^{\star}} |D_{\text{valid}} \cap S_j| \geq \frac{d_0}{8n_D}. \tag{21}$$

Define $\Delta = \log(\delta)/\log(1-\delta)$. Combining eq. (21) with eq. (20) and applying the probabilistic method, there exists an $i^{\star} \geq j^{\star}/2$ such that,

$$\frac{|D_{\text{valid}} \cap (S_{i^{\star}+1} \cup \cdots \cup S_{i^{\star}+\Delta})|}{|S_{i^{\star}+1} \cup \cdots \cup S_{i^{\star}+\Delta}|} \geq \frac{\delta(1-\delta)d_0}{8Cd|\mathcal{A}|n_D}. \tag{22}$$

Note that $\Delta$ is chosen to be sufficiently large, so that every infinite trajectory on $\mathcal{T}_{a'}^{\star}$ must intersect at least once with the band of vertices $S_{i^{\star}+\Delta+1}^{a'} \cup \cdots \cup S_{i+2\Delta}^{a'}$. Note that this band is different from the one considered in eq. (22). Define $L_{a'}$ as the set of longest prefixes across infinite trajectories in $\mathcal{T}_{a'}^{\star}$ which belong to $S_{i^{\star}+\Delta+1}^{a'} \cup \cdots \cup S_{i+2\Delta}^{a'}$.

Note that our objective is to show that an infinite trajectory sampled on $\mathcal{T}_{a'}^{\star}$ where $a' \sim P(\cdot|a)$, has a long prefix in Dict. We can truncate this trajectory to lower bound this probability, and therefore, we assume that the infinite trajectories on $\mathcal{T}_{a'}^{\star}$ terminate once they reach a substring in $L_{a'}$. Furthermore, note that although $\Delta$ is large, it is still a constant depending on $\delta$. Therefore, the band of states $S_{i^{\star}+\Delta+1}^{a'} \cup \cdots \cup S_{i^{\star}+2\Delta}^{a'}$ is not too wide, and all the substrings in $L_{a'}$ have approximately similar probabilities to each other. In particular, for any character $a \in \mathcal{A}$, and for any $a' \in \mathcal{A}$ and $t \in L_{a'}$, decomposing $P(t|a)$ as $P(t|t_1 = a')P(a'|a)$,

$$\delta^2(1-\delta) \cdot (1-\delta)^{i+\Delta} \overset{(i)}{\leq} P(t|a) \overset{(ii)}{\leq} (1-\delta)^{i+\Delta}. \tag{23}$$

Inequality $(i)$ follows from the fact that all transition probabilities are at least $\delta$, so every leaf node in $L_{a'}$ must have $P(t|t_1 = a') \geq (1-\delta)^{i+2\Delta+1}$, and the fact that $P(a'|a) \geq \delta$. Inequality $(ii)$

follows similarly from the fact that $t$ is a leaf node of $L_{a'}$ and therefore $P(t|t_1 = a') \leq (1-\delta)^{i+\Delta}$. Therefore, instead of bounding the probability of any event under the distribution over substrings in $L_{a'}$ induced by truncating the infinite strings sampled on $\mathcal{T}_{a'}^\star$, it suffices to count the fraction of substrings in $L_{a'}$ satisfying the event (which are equivalent up to a $\delta(1-\delta)$ factor). Define,

$$\mathsf{pre}(t) = (t_1, t_{1:2}, t_{1:3}, \cdots, t_{1:|t|})$$

As the set of prefixes of $t$ (including $t$). Note that at most $\Delta$ of the prefixes of any substring $t$ can intersect with $S_{i^\star+1}^a \cup \cdots \cup S_{i^\star+\Delta}^a$. Therefore,

$$\sum_{a' \in \mathcal{A}} \sum_{t \in L_{a'}} \mathbf{1}(\mathsf{pre}(t) \cap D_{\text{valid}} \cap (S_{i^\star+1}^{a'} \cup \cdots \cup S_{i^\star+\Delta}^{a'}) \neq \emptyset)$$

$$\geq \sum_{a' \in \mathcal{A}} \sum_{t \in L_{a'}} \frac{|\mathsf{pre}(t) \cap D_{\text{valid}} \cap (S_{i^\star+1}^{a'} \cup \cdots \cup S_{i^\star+\Delta}^{a'})|}{\Delta}$$

$$\overset{(i)}{\geq} \sum_{a' \in \mathcal{A}} \frac{|D_{\text{valid}} \cap (S_{i^\star+1}^{a'} \cup \cdots \cup S_{i^\star+\Delta}^{a'})|}{\Delta}$$

$$\overset{(ii)}{\geq} \frac{\delta d_0(1-\delta)}{8Cd\Delta|\mathcal{A}|n_D} \sum_{a' \in \mathcal{A}} |S_{i^\star+1}^{a'} \cup \cdots \cup S_{i^\star+\Delta}^{a'}|$$

$$\overset{(iii)}{\geq} \frac{\delta^3 d_0(1-\delta)}{8Cd\Delta|\mathcal{A}|n_D} \sum_{a' \in \mathcal{A}} |L_{a'}|,$$

where $(i)$ uses the fact that the prefixes of $t \in L_{a'}$ cover all the substrings in $S_{\leq i^\star+\Delta}^{a'}$, and therefore $\cup_{t \in L_{a'}} \mathsf{pre}(t) \supset S_{i^\star+1}^{a'} \cup \cdots \cup S_{i^\star+\Delta}^{a'}$, and $(ii)$ uses eq. (22). Finally, $(iii)$ uses the fact that $\Delta$ is not too large, and therefore, for any substring $t' \in S_{i^\star+1}^{a'} \cup \cdots \cup S_{i^\star+\Delta}^{a'}$, there are at most $1/(1-\delta)^{2\Delta} = 1/\delta^2$ substrings $t \in L_{a'}$ which contain it as a prefix. This means, $|L_{a'}| \leq |S_{i^\star+1}^{a'} \cup \cdots \cup S_{i^\star+\Delta}^{a'}|/\delta^2$. After dividing by $\sum_{a' \in \mathcal{A}} |L_{a'}|$ on both sides, this implies,

$$\mathbb{E}_{a' \sim \text{Unif}(\mathcal{A})} \left[ \Pr_{t \sim \text{Unif}(L_{a'})} \left( \mathsf{pre}(t) \cap D_{\text{valid}} \cap (S_{i^\star+1}^{a'} \cup \cdots \cup S_{i^\star+\Delta}^{a'}) \neq \emptyset \Big| a' \right) \right] \geq \frac{\delta^3 d_0(1-\delta)}{8Cd\Delta|\mathcal{A}|n_D}.$$

$$(24)$$

The event inside the inner probability term is the event that an infinitely long string (truncated at $L_{a'}$) has a prefix which lies in $D_{\text{valid}}$ and which intersects with $S_{i^\star+1}^{a'} \cup \cdots \cup S_{i^\star+\Delta}^{a'}$, which implies that it has probability $P(t|a) \leq \sqrt{\delta/Cd}$. Therefore, we have that for any $a \in \mathcal{A}$, sampling an $a' \sim P(\cdot|a)$ and an infinite trajectory $\mathsf{traj} \sim \mathcal{T}_{a'}^\star$,

$$\mathbb{E}_{a' \sim P(\cdot|a)} \left[ \Pr_{\mathsf{traj} \sim \mathcal{T}_{a'}^\star} \left( \min_{t \in \mathsf{traj} \cap D_{\text{valid}}} P(t|a) \leq \sqrt{\delta/Cd} \Big| a' \right) \right]$$

$$\overset{(i)}{\geq} \delta^2(1-\delta) \cdot \mathbb{E}_{a' \sim P(\cdot|a)} \left[ \Pr_{t' \sim \text{Unif}(L_{a'})} \left( \min_{t \in \mathsf{pre}(t') \cap D_{\text{valid}}} P(t|a) \leq \sqrt{\delta/Cd} \Big| a' \right) \right]$$

$$\overset{(ii)}{\geq} \delta^2(1-\delta) \cdot \mathbb{E}_{a' \sim P(\cdot|a)} \left[ \Pr_{t' \sim \text{Unif}(L_{a'})} \left( \mathsf{pre}(t') \cap D_{\text{valid}} \cap (S_{i^\star+1}^{a'} \cup \cdots \cup S_{i^\star+\Delta}^{a'}) \neq \emptyset \Big| a' \right) \right]$$

$$\overset{(iii)}{\geq} \delta^3(1-\delta) \cdot \mathbb{E}_{a' \sim \text{Unif}(\mathcal{A})} \left[ \Pr_{t' \sim \text{Unif}(L_{a'})} \left( \mathsf{pre}(t') \cap D_{\text{valid}} \cap (S_{i^\star+1}^a \cup \cdots \cup S_{i^\star+\Delta}^a) \neq \emptyset \Big| a' \right) \right]$$

$$\geq \delta^3(1-\delta) \cdot \frac{\delta^3 d_0(1-\delta)}{8Cd\Delta|\mathcal{A}|n_D}.$$

Here $(i)$ follows by truncating the trajectory $\mathsf{traj}$ to terminate at a node in $\cup_{a' \in \mathcal{A}} L_{a'}$ and from eq. (23), $(ii)$ follows by arguing that $i^\star \leq j^\star/2$ and therefore if a prefix of $t'$ lies in $S_{i^\star+1}^{a'} \cup \cdots \cup S_{i^\star+\Delta}^{a'}$, then it must have $P(t|a) \leq \sqrt{\delta/Cd}$. Inequality $(iii)$ follows by noting that all the transitions $P(a'|a)$ have probability $\geq \delta$, and the last inequality follows from eq. (24). $\qquad \square$

**Proof of Lemma B.5**

Lemma B.10 concludes that given any previous sequence of tokens terminating in a character $a$, with constant probability, an infinite trajectory sampled from $\mathcal{T}_{a'}^{\star}$ with $a' \sim P(\cdot|a)$ has as prefix, a substring $\boldsymbol{t}$, which not only has low probability, with $P(\boldsymbol{t}|a) \leq \sqrt{\delta/Cd}$, but also belongs to the subset of tokens $D_{\text{valid}}$. Note that regardless of the previously sampled tokens, it is legal to sample any token in $D_{\text{valid}}$ as the current token, since by definition, these tokens are not the suffixes of any other tokens in Dict. Moreover, if any trajectory on $\mathcal{T}_{a'}^{\star}$ reaches a token in $D_{\text{valid}}$, then it must be largest token along that trajectory, since none of the tokens in $D_{\text{valid}}$ are prefixes of another token in Dict.

Consider generating a new token by rejection sampling. Suppose the set of previous tokens $\boldsymbol{t}_1, \cdots, \boldsymbol{t}_i$ end in some character $a$. Sample the next character $a' \sim P(\cdot|a)$ and an infinite trajectory on $\mathcal{T}_{a'}^{\star}$. If it reaches an illegal token $\boldsymbol{t}$ such that $\boldsymbol{t}_j \boldsymbol{t}_{j+1} \cdots \boldsymbol{t}_i \boldsymbol{t}$ already exists in Dict, this token is rejected and the trajectory is resampled. By the prefix-free property of these tokens, if this trajectory visits a token in $D_{\text{valid}}$, it must immediately be output as the next token. Note that this probability is lower bounded by,

$$\mathbb{E}_{a' \sim P(\cdot|a)} \left[ \Pr_{\text{traj} \sim \mathcal{T}_{a'}^{\star}} \left( \min_{\boldsymbol{t} \in \text{traj} \cap D_{\text{valid}}} P(\boldsymbol{t}|a) \leq \sqrt{\delta/Cd} \middle| a' \right) \right]$$

which is lower bounded by $\text{poly}(\epsilon, \delta)$, the subject of Lemma B.10. Therefore with this probability, the process terminates in the first step with a token in $D_{\text{valid}}$ being sampled.

## B.3 Analysis in the small dictionary case

In this section, we will prove Theorem B.3.2 and Theorem B.3.3. In particular we show that, either,

1. The dictionary is small with low probability. i.e., $\Pr(|\text{Dict}| < d_0) = e^{-\Omega(\epsilon^2 d/\log^2(1/\delta))}$, or,

2. Or conditioned on the dictionary being small, $|\text{Dict}| < d_0$, with high probability $\geq 1 - e^{-\Omega(\epsilon^2 d/\log^2(1/\delta))}$,

$$\min_{Q \in \mathcal{Q}_{1\text{-gram}}} \mathcal{L}(Q \circ \text{enc}(\cdot)) \leq 4 \left( 1 - \frac{2d_0}{d} + O\left( \frac{1}{\log(d)} \right) \right) H_\infty + \frac{2d_0}{d} \cdot \log(2|\mathcal{A}|).$$

For $i \in [d]$, define the indicator random variable,

$$X(\boldsymbol{s}', \text{Dict}) = \mathbf{1}(\exists \text{a pair of tokens in } \text{enc}_{\text{BPE}}(\boldsymbol{s}') \text{ under Dict appears at least } \log(d) \text{ times}).$$

which captures the event that the string $\boldsymbol{s}'$ is compressed well by the dictionary Dict under the sequential encoder.

Let $\text{Dict}_i$ denote the dictionary stored by Algorithm 1 right after $\text{text}_i$ is processed. The key insight behind this lemma is the following statement, asserting that the sequential encoder satisfies a "monotonicity" property: for any $j$ and string $\boldsymbol{s}'$, if there exists a pair of tokens appearing more than $\log(d)$ times consecutively in the sequential encoding of $\boldsymbol{s}'$ under $\text{Dict}_j$, then there must exist a pair of tokens appearing more than $\log(d)$ times consecutively in the greedy encoding of $\boldsymbol{s}'$ under $\text{Dict}_i$ for any $i < j$. This implies that $X(\boldsymbol{s}', \text{Dict}_j) \leq X(\boldsymbol{s}', \text{Dict}_i)$ if $i < j$ for any string $\boldsymbol{s}'$. This monotonicity property implies that the last dictionary output by the learner, $\text{Dict}_d$ sequentially encodes a $1 - \epsilon$ fraction of the previously seen texts, $\text{text}_i$ in a way where every pair of tokens appears at most $\log(d)$ times. While $\text{Dict}_d$ is correlated with these texts, we can circumvent this correlation by using a martingale argument to prove the statement of the lemma.

**Lemma B.11.** *Let* Dict *be the dictionary returned by Algorithm 1. Then,*

$$\min \left\{ \Pr \left( \mathbb{E}\left[ X(\boldsymbol{s}', Dict) \middle| Dict \right] \geq 2d_0/d \middle| |Dict| < d_0 \right), \Pr \left( |Dict| < d_0 \right) \right\} \leq e^{-\epsilon^2 d/8\log^2(1/\delta)}.$$

*where $\boldsymbol{s}'$ is a fresh substring of length $d$ sampled from the stochastic source.*

*Proof.* Let $\text{Dict}_i$ denote the state of dictionary returned by Algorithm 1 right after $\text{text}_i$ is processed. Then, $\text{Dict}_d$ is the final dictionary returned by Algorithm 1. Suppose $\mathbb{E}\left[ X(\boldsymbol{s}', \text{Dict}_d) \middle| \text{Dict}_d \right] \geq 2d_0/d$,

where $s'$ is a fresh substring of length $d$ sampled from the stochastic source. Using monotonicity of the sequential encoder, almost surely for any string $s'$, $X(s', \mathsf{Dict}_i) \leq X(s', \mathsf{Dict}_j)$ for any $j > i$. Therefore,

$$\mathbb{E}\big[X\big(s', \mathsf{Dict}_d\big)\big|\mathsf{Dict}_d\big] \geq 2d_0/d \implies \sum_{i=1}^{d-1} \mathbb{E}\big[X\big(s', \mathsf{Dict}_i\big)\big|\mathsf{Dict}_i\big] \geq 2d_0 \cdot \frac{d-1}{d} \qquad (25)$$

Note in this expectation, $s'$ is an independent string of length $d$ sampled from the stochastic source. Since $\mathsf{Dict}_i$ and $\mathsf{text}_{i+1}$ are independent, we may instead write,

$$\sum_{i=1}^{d-1} \mathbb{E}\big[X\big(\mathsf{text}_{i+1}, \mathsf{Dict}_i\big)\big|\mathsf{Dict}_i, \mathsf{text}_i, \mathsf{Dict}_{i-1}, \cdots, \mathsf{Dict}_1, \mathsf{text}_1\big] \geq 2d_0 \cdot \frac{d-1}{d}.$$

For brevity, denote $X_i = X(\mathsf{text}_{i+1}, \mathsf{Dict}_i)$ and define the filtration $\mathcal{F}_i = \sigma(\{\mathsf{text}_1, \mathsf{Dict}_1, \cdots, \mathsf{text}_i, \mathsf{Dict}_i\})$. Note that $\sum_{j=1}^{i} X_j - \mathbb{E}[X_j|\mathcal{F}_i]$ forms a martingale sequence under the filtration $\{\mathcal{F}_i : i \in [d]\}$. Therefore, by the Azuma-Hoeffding inequality, for any $\eta > 0$,

$$\Pr\left(\sum_{i=1}^{d-1} \mathbb{E}[X_i|\mathcal{F}_i] - X_i \leq -\eta\right) \leq e^{-\eta^2}. \qquad (26)$$

Under Case I, we have that $\sum_{i=1}^{d} X_i \leq d_0$. Therefore, from eq. (25) and eq. (26),

$$\Pr\left(|\mathsf{Dict}| < d_0; \ \mathbb{E}\big[X(s', \mathsf{Dict})\big|\mathsf{Dict}\big] \geq 2d_0/d\right) \leq \Pr\left(\sum_{i=1}^{d-1} X_i < d_0; \ \sum_{i=1}^{d-1} \mathbb{E}[X_i|\mathcal{F}_i] \geq 2d_0 \cdot \frac{d-1}{d}\right)$$

$$\leq \Pr\left(\sum_{i=1}^{d-1} \mathbb{E}[X_i|\mathcal{F}_i] - X_i \geq d_0 \cdot \frac{d-2}{d}\right)$$

$$\leq e^{-d_0^2(1-2/d)^2}$$

$$\leq e^{-d_0^2/2} = e^{-\epsilon^2 d/8 \log^2(1/\delta)}.$$

Finally, using the inequality $\Pr(A, B) = \Pr(A|B)\Pr(B) \geq (\min\{\Pr(A), \Pr(B)\})^2$ completes the proof. $\qquad\square$

**Proofs of Theorem B.3.2 and Theorem B.3.3** If $\Pr(|\mathsf{Dict}| < d_0) \leq \epsilon^{-\epsilon^2 d/8 \log^2(1/\delta)}$ the proof of Theorem B.3.2 concludes. Otherwise, consider the case $\Pr(|\mathsf{Dict}| < d_0) > \epsilon^{-\epsilon^2 d/8 \log^2(1/\delta)}$, whereby, $\mathbb{E}[X(s', \mathsf{Dict})|\mathsf{Dict}] \leq 2d_0/d$ with probability $\geq 1 - e^{-\epsilon^2 d/8 \log^2(1/\delta)}$ conditioned on $|\mathsf{Dict}| < d_0$ by Lemma B.11. Recall that when $|\mathsf{Dict}| < d_0$, Algorithm 1 uses a parallel implementation of the sequential encoder which chunks a new string into pieces of length $d$, denoted $\{\mathsf{chunk}_i : i \in [d]\}$ and uses the sequential encoder under $\mathsf{Dict}_d$ to tokenize each chunk. Note that since the source is Markovian, the chunked process $\{\mathsf{chunk}_i = (X_{id+1}, X_{id+2}, \cdots, X_{(i+1)d}) : i = 1, 2, \cdots\}$ is also Markovian and ergodic. Therefore, by a similar limiting argument as in Lemma A.4, using the Krylov–Bogolyubov argument (cf. Proposition 4.2 in Chen (2018)) for Markov processes, we have that,

$$\lim_{\ell \to \infty} \frac{\sum_{i=1}^{\ell} X(\mathsf{chunk}_i, \mathsf{Dict})}{\ell} = \mathbb{E}[X(s', \mathsf{Dict})] \leq \frac{2d_0}{d}.$$

where $s'$ is a fresh string of length $d$ sampled with initial state distribution as the stationary measure of the stochastic source. On the remaining (limiting) $1 - 2d_0/d$ fraction of the chunks, their sequential encodings have every pair of tokens appearing at most $\log(d)$ times consecutively. Using Theorem 1 of Navarro and Russo (2008), the number of tokens in the encoding of each of these chunks cannot be too large, and satisfies,

$$|\mathsf{enc}_{\mathrm{BPE}}(\mathsf{chunk}_i)| \cdot \log |\mathsf{enc}_{\mathrm{BPE}}(\mathsf{chunk}_i)| \leq 2dH_\infty + O(d/\log(d))$$
$$\implies |\mathsf{enc}_{\mathrm{BPE}}(\mathsf{chunk}_i)| \cdot \log d \leq 2dH_\infty + O(d/\log(d)) \qquad (27)$$

For the (limiting) $2d_0/d$ fraction of the "bad" chunks, their sequential encodings may have one or more pairs of tokens which appear more than $\log(d)$ times consecutively.

Define $\mathcal{E}_i = \{X(\mathsf{chunk}_i, \mathsf{Dict}) = 1\}$ where $\mathsf{Dict} = \mathsf{Dict}_d$ is the dictionary returned by Algorithm 1 and consider the unigram model $Q_{\mathrm{uni}}(\boldsymbol{t}) = \frac{1}{2}Q_1(\boldsymbol{t}) + \frac{1}{2}Q_2(\boldsymbol{t})$, which is the uniform mixture of two models,

$$Q_1(\boldsymbol{t}) \propto \frac{1}{(2|\mathcal{A}|)^{|\boldsymbol{t}|}}, \quad \text{and} \quad Q_2(\boldsymbol{t}) = \mathbb{E}\left[\frac{n_{\boldsymbol{t}}^1}{|\mathsf{enc}_{\mathrm{BPE.split}}(\mathsf{chunk}_1)|}\,\middle|\,\mathcal{E}_1^c\right],$$

and let $Q_{\mathrm{uni}}(\boldsymbol{t}_1, \cdots, \boldsymbol{t}_i) = Q_{\#}(j)\prod_{j=1}^i Q_{\mathrm{uni}}(\boldsymbol{t}_i)$ for some distribution $Q_{\#}(i)$ over the number of tokens to be chosen later. We will analyze the case where the total number of chunks $\ell$ is finite and take the limit $m \to \infty$ later. Then, the overall loss of the algorithm is,

$$\begin{aligned}
&\mathcal{L}_m(Q_{\mathrm{uni}} \circ \mathsf{enc}(\cdot))\\
&= -\mathbb{E}[\log Q_{\mathrm{uni}}(\mathsf{enc}_{\mathrm{BPE.split}}(\boldsymbol{s}))]\\
&= -\sum_{\boldsymbol{t} \in \mathsf{Dict}} \mathbb{E}[n_{\boldsymbol{t}} \log Q_{\mathrm{uni}}(\boldsymbol{t}) + \log Q_{\mathrm{uni}}(|\mathsf{enc}_{\mathrm{BPE.split}}(\boldsymbol{s})|)]\\
&\overset{(i)}{=} -\sum_{i=1}^{\ell} \mathbb{E}\left[\sum_{\boldsymbol{t} \in \mathsf{Dict}} n_{\boldsymbol{t}}^i \log Q_{\mathrm{uni}}(\boldsymbol{t})\right] + \log(m)\\
&= -\sum_{i=1}^{\ell} \mathbb{E}\left[\sum_{\boldsymbol{t} \in \mathsf{Dict}} n_{\boldsymbol{t}}^i \log Q_{\mathrm{uni}}(\boldsymbol{t})\,\middle|\,\mathcal{E}_i\right]\Pr(\mathcal{E}_i) + \mathbb{E}\left[\sum_{\boldsymbol{t} \in \mathsf{Dict}} n_{\boldsymbol{t}}^i \log Q_{\mathrm{uni}}(\boldsymbol{t})\,\middle|\,\mathcal{E}_i^c\right]\Pr(\mathcal{E}_i^c) + \log(m).
\end{aligned}$$
(28)

where $n_{\boldsymbol{t}}^i$ is the number of times $\boldsymbol{t}$ is observed in the BPE encoding of $\mathsf{chunk}_i$ and $(i)$ uses the fact that $|\mathsf{enc}_{\mathrm{BPE.split}}(\boldsymbol{s})|$ follows some distribution supported on $[m]$, which implies its entropy is upper bounded by $\log(m)$. First observe that,

$$\begin{aligned}
\sum_{i=1}^{\ell} \mathbb{E}\left[\sum_{\boldsymbol{t} \in \mathsf{Dict}} n_{\boldsymbol{t}}^i \log Q_{\mathrm{uni}}(\boldsymbol{t})\,\middle|\,\mathcal{E}_i^c\right] &\leq \sum_{i=1}^{\ell} \mathbb{E}\left[|\mathsf{enc}_{\mathrm{BPE}}(\mathsf{chunk}_i)| \cdot \sum_{\boldsymbol{t} \in \mathsf{Dict}} \frac{n_{\boldsymbol{t}}^i}{|\mathsf{enc}_{\mathrm{BPE}}(\mathsf{chunk}_i)|} \log Q_{\mathrm{uni}}(\boldsymbol{t})\,\middle|\,\mathcal{E}_i^c\right]\\
&\overset{(i)}{\leq} \ell\left(\frac{2dH_{\infty} + O(d/\log(d))}{\log(d)}\right)\sum_{\boldsymbol{t} \in \mathsf{Dict}} Q_2(\boldsymbol{t})\log Q_{\mathrm{uni}}(\boldsymbol{t})
\end{aligned}$$

where $(i)$ uses the upper bound on $|\mathsf{enc}_{\mathrm{BPE.split}}(\mathsf{chunk}_i)|$ under the event $\mathcal{E}_i^c$ (eq. (27)). Since $Q_{\mathrm{uni}}(\boldsymbol{t}) = \frac{1}{2}Q_1(\boldsymbol{t}) + \frac{1}{2}Q_2(\boldsymbol{t}) \geq \frac{1}{2}Q_2(\boldsymbol{t})$ and $Q_2$ is a distribution supported on at most $d$ tokens, this term results in the upper bound,

$$\sum_{i=1}^{\ell} \mathbb{E}\left[\sum_{\boldsymbol{t} \in \mathsf{Dict}} n_{\boldsymbol{t}}^i \log Q_{\mathrm{uni}}(\boldsymbol{t})\,\middle|\,\mathcal{E}_i^c\right] \leq \ell\left(\frac{2dH_{\infty} + O(d/\log(d))}{\log(d)}\right)\log(2d).$$
(29)

On the other hand, since $Q_{\mathrm{uni}}(\boldsymbol{t}) \geq \frac{1}{2}Q_1(\boldsymbol{t})$,

$$\begin{aligned}
\sum_{i=1}^{\ell} \mathbb{E}\left[\sum_{\boldsymbol{t} \in \mathsf{Dict}} n_{\boldsymbol{t}}^i \log(1/Q_{\mathrm{uni}}(\boldsymbol{t}))\,\middle|\,\mathcal{E}_i\right] &\leq \sum_{i=1}^{\ell} \mathbb{E}\left[\sum_{\boldsymbol{t} \in \mathsf{Dict}} n_{\boldsymbol{t}}^i \log(2/Q_1(\boldsymbol{t}))\,\middle|\,\mathcal{E}_i\right]\\
&\leq \sum_{i=1}^{\ell} \mathbb{E}\left[\sum_{\boldsymbol{t} \in \mathsf{Dict}} n_{\boldsymbol{t}}^i \left(\log(2) + |\boldsymbol{t}|\log(2|\mathcal{A}|)\right)\,\middle|\,\mathcal{E}_i\right]\\
&\leq \ell d\log(2) + \ell d\log(2|\mathcal{A}|)
\end{aligned}$$
(30)

where the last inequality uses the fact that $\sum_{\boldsymbol{t} \in \mathsf{Dict}} n_{\boldsymbol{t}}^i \leq d$ and $\sum_{\boldsymbol{t} \in \mathsf{Dict}} |\boldsymbol{t}|n_{\boldsymbol{t}}^i = d$ computes the length of $\mathsf{chunk}_i$.

Overall, since $\sum_{i=1}^{\ell} \Pr(\mathcal{E}_i) \leq 2d_0/d$ by eq. (27), combining this with eqs. (28) to (30),

$$\mathcal{L}_m(Q_{\mathrm{uni}} \circ \mathsf{enc}(\cdot)) \leq \left(1 - \frac{2d_0}{d}\right)\ell\left(\frac{2dH_{\infty} + O(d/\log(d))}{\log(d)}\right)\log(2d) + \frac{2d_0}{d}\ell d\log(4|\mathcal{A}|).$$

Dividing throughout by the length of the character sequence $m \in [d(\ell - 1), d\ell]$ and letting $\ell \to \infty$,

$$\min_{Q \in \mathcal{Q}_{1\text{-gram}}} \mathcal{L}(Q \circ \mathsf{enc}(\cdot)) \leq \mathcal{L}(Q_{\text{uni}} \circ \mathsf{enc}(\cdot)) \leq \left(1 - \frac{2d_0}{d}\right)\left(2H_\infty + O\left(\frac{1}{\log(d)}\right)\right) + \frac{2d_0}{d}\log(4|\mathcal{A}|).$$

## C  Additional Theoretical Results II: Learning the likelihood model

The guarantees we prove in Theorems 3.1, 3.6 and B.2 on various tokenizers assume that the downstream model is trained optimally. In practice, these models are trained from a finite dataset and the sample complexity of learning this likelihood model scales with the number of tokens in the dictionary. In this section, we step away from the transformer architecture and focus on analyzing the performance of a simple estimator for the unigram model based on Laplace smoothing. We leave the problem of analyzing the finite-sample statistical error of simple transformer models trained with gradient descent as an interesting open direction for future research.

The result of Theorem 3.1 establishes that under appropriate assumptions on the Markov source, there exists a tokenizer $\mathcal{T}$ and a unigram model over tokens $Q^\star \in \mathcal{Q}_{1\text{-gram}}$ such that,

$$\lim_{m \to \infty} \frac{1}{m} \mathbb{E}\left[\log(1/Q^\star(\mathsf{enc}(s))\right]$$

$$\leq (1 + \varepsilon) \cdot \lim_{m \to \infty} \frac{1}{m}\mathbb{E}\left[\log(1/P(s))\right]$$

Or in other words,

$$\lim_{m \to \infty} \frac{1}{m}\mathsf{KL}(P, Q^\star(\mathsf{enc}(\cdot))) \leq \varepsilon \cdot \lim_{m \to \infty} \frac{1}{m}\mathbb{E}\left[\log(1/P(s))\right].$$

This implies that with the appropriate tokenization, the measure associated to the string by the best unigram model over tokens is close to that induced by the true Markov distribution over characters in KL divergence. In this section, we establish finite-sample guarantees on learning $Q^\star$ specifically for the LZW tokenizer. The approach we consider for distribution learning is a smoothed Laplace estimator described in more detail in Algorithm 2.

For any constant $\theta \in (0, 1)$, define $\mathcal{E}_\theta$ as the event that every maximal token $t$ (Definition A.5) in the LZW dictionary satisfies $1/d^{1-\theta} \geq \max_a P(t|a) \geq \delta/d^{1+\theta}$. By Lemmas A.10 and A.11 when the LZW tokenizer is trained on a dataset of size $\widetilde{\Omega}_\delta(d)$ drawn from a stochastic source satisfying Assumption 3.2, $\mathcal{E}_\theta$ occurs with probability $\geq 1 - d^{-\Omega_{\theta,\delta}(\log(d))}$.

**Theorem C.1.** *Consider any constant $\theta \in (0, 1)$, failure probability $\eta \in (0, 1)$ and approximation error $\xi \in (0, 1)$. Assume that the learnt LZW tokenizer $\mathcal{T}_{LZW}$ satisfies the event $\mathcal{E}_\theta$, which occurs with probability $\geq 1 - d^{-\Omega_{\theta,\delta}(\log(d))}$. Assume that $d^{1-3\theta} \geq 1 + \delta^{-2}$ and that the stochastic source satisfies Assumption 3.2. For an absolute constant $C > 0$, assume that the size of the training dataset is at least $n_{lm}^\star(\xi)$, where,*

$$n_{lm}^\star \triangleq \frac{Cd^{1+\theta}\log^3(d/\eta\delta)\log\log(d/\eta)}{\delta\xi^2}$$

*Then, Algorithm 2 learns a unigram model $\widehat{Q}$ such that,*

$$\mathcal{L}(\widehat{Q} \circ \mathbf{enc}_{gre}(\cdot)) \leq (1 + \xi)\min_{Q \in \mathcal{Q}_{1\text{-gram}}} \mathcal{L}(Q \circ \mathbf{enc}_{gre}(\cdot))$$

*with probability $\geq 1 - \eta$.*

In conjunction with Theorem 3.6, this gives end-to-end guarantees on the cross-entropy loss of the LZW tokenizer (with vocabulary size $\leq d$) with the Laplace estimator as the downstream unigram model. We instantiate this result choosing $\theta = 0.01$ in Theorem C.1.

**Corollary C.2.** *Choose any $\xi \in (0, 1)$. Suppose the data source satisfies Assumption 3.2. On a dataset of size $\widetilde{\Omega}_\delta(d)$ drawn from the source, train an LZW tokenizer $\mathcal{T}_{LZW}$ with $d$ tokens. Subsequently, using Algorithm 2, learn a unigram model $\widehat{Q}$ using a dataset of size at least $\widetilde{\Omega}(d^{1.01}/\delta\xi^2)$ drawn from the source. Then, with probability $\geq 1 - d^{-\Omega_\delta(\log(d))}$,*

$$\mathcal{L}(\widehat{Q} \circ \mathbf{enc}_{gre}(\cdot)) \leq \frac{1 + \xi}{1 - \varepsilon}\min_Q \mathcal{L}(Q),$$

*where $\varepsilon = \frac{\log(1/\delta)}{0.99\log(d)}$.*

The analysis of Theorem C.1 relies on showing that the distribution over tokens induced when a string sampled from the data source is encoded into tokens by the greedy encoder and the LZW dictionary is a Markov process. In general, given a set of previously sampled tokens $t_1, \cdots, t_i$, the next token $t_{i+1}$ is sampled from the distribution $P(t_{i+1}|t_i; \forall j \in [i], t_{i-j+1} \cdots t_i t_{i+1} \notin \mathsf{Dict})$. The conditioning is to simply guarantee that the previous tokens which were sampled were indeed maximal, since if $t_i t_{i+1} \in \mathsf{Dict}$, then the previous token returned would in fact have been this longer token and not $t_i$ (and likewise for $t_{i-1} t_i t_{i+1}$ and so on). While in general, this process is complicated and depends on all the previous tokens sampled, for the LZW dictionary, we show that the conditioning $\{\forall j \in [i], t_{i-j+1} \cdots t_i t_{i+1} \notin \mathsf{Dict}\}$ can be removed, thereby resulting in a simple Markov process over tokens.

Furthermore, we establish that this Markov process has a relatively large spectral gap. The optimal unigram model ends up being the stationary distribution over tokens induced by greedy encoder. Given the large spectral gap of the Markov process over tokens, estimating the stationary distribution of this process in KL divergence ends up being closely related to estimating a distribution from i.i.d. samples in the same metric. For this problem, the de-facto choice of estimator is the Laplace estimator, and several existing results provide finite-sample bounds on the KL divergence (Braess and Sauer, 2004; Han et al., 2021; Mourtada and Gaïffas, 2022). The Laplace estimator (Line 6 of Algorithm 2) is simply a smoothed empirical estimate to account for the degeneracy of the KL divergence in its second argument as any coordinate approaches 0. The non-i.i.d.ness of the Markov process is circumvented by using concentration inequalities which are a function of the spectral gap (Naor et al., 2020).

---

**Algorithm 2** Training likelihood model on tokens

---

    **Input:** A training dataset of size $n_{\mathrm{lm}}$, likelihood model class $\mathcal{Q}$, likelihood model training algorithm $\mathsf{TrainLM}$
    **Output:** Likelihood model $Q \in \mathcal{Q}$.
1: Tokenize the training dataset into a sequence of tokens $\mathcal{T} = (t_1, \cdots, t_i)$.
2: Train a likelihood model $Q$ on the tokenized dataset $\mathcal{T}$ using the $\mathsf{TrainLM}(\mathcal{T}, \mathcal{Q})$ subroutine.

    // In the case of $\mathcal{Q} = \mathcal{Q}_{\text{1-gram}}$ use the Laplace estimator
    **def** $\mathsf{TrainLM}(\mathcal{T}, \mathcal{Q}_{\text{1-gram}})$:
3: Truncate the dataset to the first $n' = \lfloor n_{\mathrm{lm}}/\ell_{\max} \rfloor$ tokens where $\ell_{\max} = 4\log(d|\mathcal{A}|)/\delta$. Let the truncated dataset be $\mathcal{T}_{\text{trunc}}$
4: Construct the unigram model $\widehat{Q}$ with $\widehat{Q}_{\#} = \mathrm{Unif}([m])$ and $\widehat{Q}_{\text{tok}}(t) = \frac{n_t+1}{n_t+|\mathsf{Dict}|}$.
    // $n_t$ is the number of times $t$ appears in $\mathcal{T}_{\text{trunc}}$.
    // Test sequences are assumed to be of length $m$.

---

### C.1 Proof of Theorem C.1

Since $\mathcal{T}_{\text{LZW}}$ uses the greedy encoder, the cross-entropy loss of the unigram model learnt by Algorithm 2 is,

$$\mathcal{L}(\widehat{Q} \circ \mathsf{enc}_{\text{gre}}(\cdot)) - \min_{Q \in \mathcal{Q}_{\text{1-gram}}} \mathcal{L}(Q \circ \mathsf{enc}_{\text{gre}}(\cdot))$$

$$= \max_{Q \in \mathcal{Q}_{\text{1-gram}}} \lim_{m \to \infty} \frac{1}{m} \mathbb{E}[\log(Q(\mathsf{enc}_{\text{gre}}(s))/\widehat{Q}(\mathsf{enc}_{\text{gre}}(s)))]$$

$$\overset{(i)}{=} \max_{Q \in \mathcal{Q}_{\text{1-gram}}} \lim_{m \to \infty} \frac{1}{m} \mathbb{E}\left[|\mathsf{enc}_{\text{gre}}(s)| \sum_{t \in \mathsf{Dict}} \frac{n_t}{|\mathsf{enc}_{\text{gre}}(s)|} \log(Q_{\text{tok}}(t)/\widehat{Q}_{\text{tok}}(t))\right] + \frac{\log(m)}{m}$$

$$\overset{(ii)}{\leq} \lim_{m \to \infty} \frac{1}{m} \mathbb{E}\left[|\mathsf{enc}_{\text{gre}}(s)| \sum_{t \in \mathsf{Dict}} \frac{n_t}{|\mathsf{enc}_{\text{gre}}(s)|} \log\left(\frac{n_t/|\mathsf{enc}_{\text{gre}}(s)|}{\widehat{Q}_{\text{tok}}(t)}\right)\right] + \frac{\log(m)}{m}$$

where in $(i)$ we use the fact that $\widehat{Q}_{\#} = \mathrm{Unif}([m])$ and in $(ii)$ we take the $\max\{\cdot\}$ inside the limit and the expectation (Fatou's lemma and Jensen's inequality) and plug in the maximizer of the negative cross-entropy, $Q_{\text{tok}}(t) = \frac{n_t}{|\mathsf{enc}_{\text{gre}}(s)|}$. Note that $\lim_{m \to \infty} \frac{n_t}{|\mathsf{enc}_{\text{gre}}(s)|} \overset{\text{a.s.}}{=} Q_{\text{MLE}}(t)$ by Lemma A.4.

Moreover, since $|\mathsf{enc}(s)|/m \leq 1$ and $\widehat{Q}_{\mathrm{tok}}(t) > 0$ surely, by the Dominated Convergence Theorem,

$$\mathcal{L}(\widehat{Q} \circ \mathsf{enc}_{\mathrm{gre}}(\cdot)) - \min_{Q \in \mathcal{Q}_{\text{1-gram}}} \mathcal{L}(Q \circ \mathsf{enc}_{\mathrm{gre}}(\cdot)) \leq \lim_{m \to \infty} \frac{1}{m} \mathbb{E}[|\mathsf{enc}_{\mathrm{gre}}(s)|] \cdot \mathsf{KL}(Q_{\mathrm{MLE}}, \widehat{Q}_{\mathrm{tok}}) \quad (31)$$

By eq. (6), we have that for any tokenizer using the greedy encoder,

$$\lim_{m \to \infty} \frac{|\mathsf{enc}_{\mathrm{gre}}(\mathbf{s})| \left( H(Q_{\mathrm{MLE}}, P) - \log(1/\delta) \right)}{m} \overset{\text{a.s.}}{\leq} H_\infty.$$

Furthermore under the event $\mathcal{E}_\theta$ which implies that the learnt dictionary is $(1-\theta)$-heavy hitting (cf. Definition A.5), which implies that,

$$H(Q_{\mathrm{MLE}}, P) \geq (1 - \theta) \log(d).$$

Therefore, by almost sure boundedness, we have that,

$$\lim_{m \to \infty} \frac{1}{m} \mathbb{E}\left[ |\mathsf{enc}_{\mathrm{gre}}(\mathbf{s})| \right] \leq \frac{H_\infty}{(1-\theta)\log(d) - \log(1/\delta)} \leq \frac{\min_{Q \in \mathcal{Q}_{\text{1-gram}}} \mathcal{L}(Q \circ \mathsf{enc}(\cdot))}{(1-\theta)\log(d) - \log(1/\delta)}$$

Putting this together with eq. (31), we have that,

$$\mathcal{L}(\widehat{Q} \circ \mathsf{enc}_{\mathrm{gre}}(\cdot)) \leq \left( 1 + \mathsf{KL}(Q_{\mathrm{MLE}}, \widehat{Q}_{\mathrm{tok}}) \right) \min_{Q \in \mathcal{Q}_{\text{1-gram}}} \mathcal{L}(Q \circ \mathsf{enc}_{\mathrm{gre}}(\cdot)), \quad (32)$$

which uses the assumption $(1-\theta)\log(d) \geq 1 + \log(1/\delta)$. In the remainder of the proof we upper bound the KL term.

By the law of large numbers established in eq. (34) and the fact that $\frac{n_t}{\sum_{t'} n_{t'}} \in [0, 1]$, we have that,

$$Q_{\mathrm{MLE}}(t) = \lim_{m \to \infty} \mathbb{E}\left[ \frac{n_t}{\sum_{t'} n_{t'}} \right] = \lim_{m \to \infty} \frac{\mathbb{E}[n_t]}{\mathbb{E}\left[ \sum_{t'} n_{t'} \right]} = \pi(t),$$

where $\pi(t)$ denote the stationary distribution over tokens induced by the greedy encoding process, which exists for the LZW tokenizer. This distribution is in fact an ergodic Markov process, as we discuss next.

By Lemmas A.10 and A.11, for any constant $\theta \in (0, 1)$, with probability $\geq 1 - d^{-\Omega_{\theta,\delta}(\log(d))}$, every maximal token in the the LZW dictionary satisfies $1/d^{1-\theta} \geq \max_a P(t|a) \geq \delta/d^{1+\theta}$. Let $S_{\mathrm{gre}}$ denote the set of tokens which have a non-zero probability (over a string drawn from the Markov source) of being chosen by the greedy encoder while encoding the string. More importantly, note that for any sequence of tokens $t_1, \cdots, t_i$, the next token is necessarily in $S_{\mathrm{gre}}$ and can be any token in this set. The reason for this is that for any $t_i, t \in S_{\mathrm{gre}}$, the concatenation $t_i t \notin S_{\mathrm{gre}}$ since $\max_{a \in \mathcal{A}} P(t_i t|a) \leq 1/\delta d^{2(1-\theta)}$, which is smaller than the $\max_{a \in \mathcal{A}} P(t'|a) \geq \delta/d^{1+\theta}$ for any token $t' \in S_{\mathrm{gre}}$ as long as $d^{1-3\theta} \geq 1/\delta^2$. This constraint implies that in the sampling procedure in Figure 7, it suffices to drop the conditioning on the event $t_j t_{j+1} \cdots t_i t \notin \mathsf{Dict}$ while sampling the next token $t$. This condition automatically implies that the sequence of tokens conditionally follows a Markov process with $\Pr(t_{i+1} = t | t_1, \cdots, t_i) = P(t | \mathsf{last}(t_i))$. Since the probability of every transition is lower bounded, this means that the Markov chain is ergodic. Moreover, the pseudo-spectral gap (Naor et al., 2020), $1 - \lambda$ can be lower bounded by the Dobrushin contraction coefficient, $\kappa$,

$$\begin{aligned}
1 - \lambda \leq \kappa &\triangleq \max_{(t,t') \in \mathsf{Dict}^2} \| \Pr(\cdot|t) - \Pr(\cdot|t') \|_{\mathrm{TV}} \\
&= \max_{(t,t') \in \mathsf{Dict}^2} 1 - \sum_{t'' \in \mathsf{Dict}} \min\{\Pr(t''|t), \Pr(t''|t')\} \\
&\leq 1 - \delta d/d^{1+\theta} \\
&= 1 - \delta d^{-\theta}. \quad (33)
\end{aligned}$$

Recall that the learner is given a training dataset of $n_{\mathrm{lm}}$ characters to train the likelihood model. By Lemma A.8, with probability $\geq 1 - d^{-\Omega(\log(d/\delta)/\delta)}$, in the run of the LZW tokenization algorithm, every token in the dictionary has length at most $\ell_{\max} = 4\log(d|\mathcal{A}|)/\delta$. Therefore, suppose the learner

always truncates the dataset to the first $n' = \lfloor n_{\mathrm{lm}}/\ell_{\max}\rfloor$ tokens and runs the Laplace estimator on this truncated dataset. With this, we move onto upper bounding,

$$\mathsf{KL}(Q_{\mathrm{MLE}}, \widehat{Q}_{\mathrm{tok}}) = \sum_{\boldsymbol{t}\in\mathsf{Dict}} \pi(\boldsymbol{t}) \log\left(\pi(\boldsymbol{t})/\widehat{Q}_{\mathrm{tok}}(\boldsymbol{t})\right)$$

which necessitates lower bounding $\widehat{Q}_{\mathrm{tok}}(\boldsymbol{t})$ for every $\boldsymbol{t}$. Recall that the learner's estimate $\widehat{Q}(\boldsymbol{t})$ in Algorithm 2 is the Laplace estimator, $\frac{n_{\boldsymbol{t}}+1}{\sum_{\boldsymbol{t}'} n_{\boldsymbol{t}'} + |\mathsf{Dict}|}$, where $\{n_{\boldsymbol{t}} : \boldsymbol{t}\in\mathsf{Dict}\}$ is computed by truncating the dataset to the first $n'$ tokens. Firstly, by invoking Corollary 1.3 of Naor et al. (2020) for the function $n_{\boldsymbol{t}} = \sum_{i=1}^{n'} \mathbb{I}(\boldsymbol{t}_i = \boldsymbol{t})$,

$$\Pr\left(|n_{\boldsymbol{t}} - \mathbb{E}[n_{\boldsymbol{t}}]| \geq c\sqrt{\frac{\mathbb{E}[n_{\boldsymbol{t}}]}{1-\lambda}\cdot\log(1/\eta)}\right) \leq \eta \tag{34}$$

for a universal constant $c > 0$. In particular, this implies that with probability $\geq 1 - \eta$, simultaneously for all $\boldsymbol{t}$,

$$|n_{\boldsymbol{t}} - \mathbb{E}[n_{\boldsymbol{t}}]| \leq \Delta_{\boldsymbol{t}} \triangleq \sqrt{\frac{d^{\theta}}{\delta}\mathbb{E}[n_{\boldsymbol{t}}]\cdot\log(|\mathsf{Dict}|/\eta)}, \text{ and, } \mathbb{E}[n_{\boldsymbol{t}}] - n_{\boldsymbol{t}} \geq \mathbb{E}[n_{\boldsymbol{t}}].$$

Under this event, for any $\boldsymbol{t}$, the estimate is lower bounded by,

$$\widehat{Q}_{\mathrm{tok}}(\boldsymbol{t}) = \frac{n_{\boldsymbol{t}}+1}{n'+|\mathsf{Dict}|} \geq \frac{\mathbb{E}[n_{\boldsymbol{t}}]+1-\min\{\mathbb{E}[n_{\boldsymbol{t}}], \Delta_{\boldsymbol{t}}\}}{n'+|\mathsf{Dict}|}$$

$$\geq \max\left\{\pi(\boldsymbol{t}) - \frac{(\Delta_{\boldsymbol{t}}-1)\,n' + |\mathsf{Dict}|\mathbb{E}[n_{\boldsymbol{t}}]}{(n')^2 + n'|\mathsf{Dict}|}, \frac{1}{n'+|\mathsf{Dict}|}\right\}$$

$$\geq \max\left\{\pi(\boldsymbol{t}) - \frac{\Delta_{\boldsymbol{t}}n' + |\mathsf{Dict}|\mathbb{E}[n_{\boldsymbol{t}}]}{(n')^2}, \frac{1}{n'+|\mathsf{Dict}|}\right\}$$

Suppose the following condition is satisfied,

$$n' = \frac{4rd^{\theta}|\mathsf{Dict}|\log(|\mathsf{Dict}|/\eta)}{\delta} \tag{C1}$$

for some $r \geq 4$. Under this condition, we have that $n' \geq 2\sqrt{r}\Delta$ and $n' \geq 4r|\mathsf{Dict}|$.

**Case I.** $\Delta_{\boldsymbol{t}}n' \geq |\mathsf{Dict}|\mathbb{E}[n_{\boldsymbol{t}}]$.

In this case, we have the upper bound,

$$\widehat{Q}_{\mathrm{tok}}(\boldsymbol{t}) \geq \max\left\{\pi(\boldsymbol{t}) - \frac{2\Delta_{\boldsymbol{t}}}{n'}, \frac{1}{n'+|\mathsf{Dict}|}\right\}$$

$$= \max\left\{\pi(\boldsymbol{t}) - 2\frac{\sqrt{\frac{d^{\theta}}{\delta}\mathbb{E}[n_{\boldsymbol{t}}]\cdot\log(|\mathsf{Dict}|/\eta)}}{n'}, \frac{1}{n'+|\mathsf{Dict}|}\right\}$$

$$\geq \max\left\{\pi(\boldsymbol{t}) - \sqrt{\frac{\pi(\boldsymbol{t})}{r|\mathsf{Dict}|}}, \frac{1}{2n'}\right\}.$$

where the last inequality uses eq. (C1).

Consider two sub-cases,

**Sub-case I.** $\pi(\boldsymbol{t}) \geq 2/r|\mathsf{Dict}|$. Define this event $\mathcal{C}_{\mathsf{I}}$.

Here,

$$\pi(\boldsymbol{t})\log(\pi(\boldsymbol{t})/\widehat{Q}_{\mathrm{tok}}(\boldsymbol{t})) \leq -\pi(\boldsymbol{t})\log\left(1 - \sqrt{\frac{1}{\pi(\boldsymbol{t})r|\mathsf{Dict}|}}\right) \leq \frac{3}{2}\sqrt{\frac{\pi(\boldsymbol{t})}{r|\mathsf{Dict}|}}. \tag{35}$$

**Sub-case II.** $\pi(\boldsymbol{t}) \le 2/r|\mathsf{Dict}|$. Define this event $\mathcal{C}_{\mathsf{II}}$.

Here,

$$\pi(\boldsymbol{t}) \log(\pi(\boldsymbol{t})/\widehat{Q}_{\mathrm{tok}}(\boldsymbol{t})) \le \pi(\boldsymbol{t}) \log\left(2n'\pi(\boldsymbol{t})\right) \le \max\left\{0, \frac{2}{r|\mathsf{Dict}|} \log\left(\frac{4n'}{r|\mathsf{Dict}|}\right)\right\}$$

$$\le \frac{2}{r|\mathsf{Dict}|} \log\left(16 d^\theta \log(|\mathsf{Dict}|/\eta)\right) \qquad (36)$$

**Case II.** $\Delta_{\boldsymbol{t}} n' < |\mathsf{Dict}|\mathbb{E}[n_{\boldsymbol{t}}]$. Define this event $\mathcal{C}_{\mathsf{III}}$.

In this case we have the upper bound,

$$\widehat{Q}_{\mathrm{tok}}(\boldsymbol{t}) \ge \pi(\boldsymbol{t}) - \frac{2|\mathsf{Dict}|\mathbb{E}[n_{\boldsymbol{t}}]}{(n')^2} \ge \pi(\boldsymbol{t}) - \frac{\pi(\boldsymbol{t})}{2r}$$

where the last inequality follows from eq. (C1). This implies that,

$$\pi(\boldsymbol{t}) \log(\pi(\boldsymbol{t})/\widehat{Q}_{\mathrm{tok}}(\boldsymbol{t})) \le -\pi(\boldsymbol{t}) \log(1 - 1/2r) \le \frac{\pi(\boldsymbol{t})}{r}. \qquad (37)$$

By using the geometric ergodicity of this Markov process (eq. (33)), when $n'$ tokens are sampled from an arbitrary initial distribution,

$$\left(1 - \kappa^{n'}\right) \pi(\boldsymbol{t}) \le \frac{\mathbb{E}[n_{\boldsymbol{t}}]}{n'} \le \kappa^{n'} + \left(1 - \kappa^{n'}\right)\pi(\boldsymbol{t}) \implies \pi(\boldsymbol{t}) \le \frac{\widehat{Q}_{\mathrm{tok}}(\boldsymbol{t})}{1 - e^{-4r|\mathsf{Dict}|\log(|\mathsf{Dict}|/\eta)}} = \frac{\widehat{Q}_{\mathrm{tok}}(\boldsymbol{t})}{1 - d^{-r}}$$

where in the implication, we use the condition on $n'$ in eq. (C1) and the bound on the contraction coefficient $\kappa$ in eq. (33).

$\mathsf{KL}(Q_{\mathrm{MLE}}, \widehat{Q}_{\mathrm{tok}})$

$$= \sum_{\boldsymbol{t} \in \mathsf{Dict}} \pi(\boldsymbol{t}) \log(\pi(\boldsymbol{t})/\widehat{Q}_{\mathrm{tok}}(\boldsymbol{t}))$$

$$\le \sum_{\boldsymbol{t} \in \mathsf{Dict}} \frac{\pi(\boldsymbol{t})}{1 - d^{-r}} \log(\pi(\boldsymbol{t})/\widehat{Q}_{\mathrm{tok}}(\boldsymbol{t})) - \log(1 - d^{-r})$$

$$\le \frac{1}{1 - d^{-r}} \sum_{\boldsymbol{t} \in \mathsf{Dict}} \mathbb{I}(\mathcal{C}_{\mathsf{I}})\pi(\boldsymbol{t})\log(\pi(\boldsymbol{t})/\widehat{Q}_{\mathrm{tok}}(\boldsymbol{t})) + \mathbb{I}(\mathcal{C}_{\mathsf{II}})\pi(\boldsymbol{t})\log(\pi(\boldsymbol{t})/\widehat{Q}_{\mathrm{tok}}(\boldsymbol{t})) + \mathbb{I}(\mathcal{C}_{\mathsf{III}})\pi(\boldsymbol{t})\log(\pi(\boldsymbol{t})/\widehat{Q}_{\mathrm{tok}}(\boldsymbol{t})) + 2d^{-r}$$

$$\le \frac{1}{1 - d^{-r}} \sum_{\boldsymbol{t} \in \mathsf{Dict}} \underbrace{\mathbb{I}(\mathcal{C}_{\mathsf{I}})\frac{3}{2}\sqrt{\frac{\pi(\boldsymbol{t})}{r|\mathsf{Dict}|}}}_{eq.\ (35)} + \underbrace{\mathbb{I}(\mathcal{C}_{\mathsf{II}})\frac{2}{r|\mathsf{Dict}|}\log\left(16d^\theta \log(|\mathsf{Dict}|/\eta)\right)}_{eq.\ (36)} + \underbrace{\mathbb{I}(\mathcal{C}_{\mathsf{III}})\frac{\pi(\boldsymbol{t})}{r}}_{eq.\ (37)} + 2d^{-r}$$

$$\le \frac{1}{1 - d^{-r}}\left(\frac{3}{2}\sqrt{\frac{|\mathsf{Dict}|}{r|\mathsf{Dict}|}} + \frac{2}{r}\log(16d^\theta \log(|\mathsf{Dict}|/\eta)) + \frac{1}{r}\right) + 2d^{-r}$$

$$\le \frac{5}{\sqrt{r}}\log(16d^\theta \log(|\mathsf{Dict}|/\eta))$$

Combining with eq. (32), we get the bound,

$$\mathcal{L}(\widehat{Q} \circ \mathsf{enc}_{\mathrm{gre}}(\cdot)) \le \left(1 + \mathsf{KL}(Q_{\mathrm{MLE}}, \widehat{Q})\right) \min_{Q \in \mathcal{Q}_{\text{1-gram}}} \mathcal{L}(Q \circ \mathsf{enc}_{\mathrm{gre}}(\cdot))$$

$$\le \left(1 + \frac{5}{\sqrt{r}}\log(16d^\theta \log(d/\eta))\right) \min_{Q \in \mathcal{Q}_{\text{1-gram}}} \mathcal{L}(Q \circ \mathsf{enc}_{\mathrm{gre}}(\cdot)).$$

Rescaling $r$ to be $r(\log(16d^\theta \log(d/\eta)))^2$ completes the proof.

# D  Additional Theoretical Results III: The generalization ability of tokenizers

The proofs of the upper bounds in the paper (Theorems 3.6 and B.2) relied on showing that the entropy $H(Q_{\text{MLE}}, P)$ is large, or in other words, the algorithm typically encodes new strings into long length (i.e. low probability under $P$) tokens. This statement about generalization to new strings is fundamentally different from having a tokenizer which compresses the training dataset well. In other words, consider the following modification: the measure $Q_{\text{MLE}}$ is defined as the expected empirical distribution over tokens when a new string is encoded into tokens, and not on the source dataset used to construct the dictionary. Suppose the definition of $Q_{\text{MLE}}$ is changed to the empirical distribution over tokens in the source dataset. Under this new definition of the MLE unigram model, the largeness of the $H(Q_{\text{MLE}}, P)$ metric, in a sense, captures compressing the source dataset well. However, we show that in general, this does not result in good tokenizers that minimize the population cross-entropy loss, suffering from $\min_{Q \in \mathcal{Q}_{\text{1-gram}}} \mathcal{L}(Q \circ \text{enc}(\cdot)) \approx H(\pi) \gg H_\infty$.

**Theorem D.1.** *Consider the stochastic source in example A.1 having entropy rate $H_\infty = \delta \log(1/\delta) + (1-\delta)\log(1/(1-\delta))$. Consider a training dataset of size $n$. For a dictionary* Dict *and* $t \in$ Dict, *define* $\widehat{Q}_{MLE}(t) = \frac{n_t(s_{src})}{|\text{enc}(s_{src})|}$ *as the empirical distribution over tokens induced by the greedy encoder when encoding the training dataset,* $s_{src}$. *There exists a dictionary* Dict *such that with probability* $\geq 1 - e^{-\Omega(\sqrt{n})}$ *over the training dataset,*

$$H(\widehat{Q}_{MLE}, P_\gamma) \geq n H_\infty (1 - O(n^{-1/4}))$$

*is large. However, for this dictionary, for any encoding algorithm (including the greedy encoder), the resulting tokenizer* $\mathcal{T} = (\text{Dict}, \emptyset, \text{enc}(\cdot), \text{dec}(\cdot))$ *satisfies,*

$$\min_{Q \in \mathcal{Q}_{\text{1-gram}}} \mathcal{L}(Q \circ \text{enc}(\cdot)) \geq (1 - \varepsilon)H(\pi)$$

*where* $\varepsilon = 2n e^{-n H_\infty (1 - O(n^{-1/4}))}$.

*Proof.* Suppose the entire training dataset was compressed into a single token, $t_{\text{src}}$. The dictionary is $\mathcal{A} \cup t_{\text{src}}$. In the following argument, we show that the number of occurrences, $n_{t_{\text{src}}}$, of the entire training dataset $t_{\text{src}}$ in a new string of length $m$ generated from the stochastic source, $s$, converges to its expectation as $m \to \infty$. Let $\pi_n^{(i)}$ denote the stationary distribution of the Markov process induced by the stochastic source over length-$n$ strings with a shift of $i$ from the starting position, and let $n_t^{(i)}$ denote the number of times $t$ appears in the training dataset starting at the position $i + rn$ for some $r > 0$. Then,

$$\lim_{m \to \infty} \frac{n_{t_{\text{src}}}}{m} = \frac{1}{n} \lim_{m \to \infty} \sum_{i=0}^{n-1} \frac{n_{t_{\text{src}}}^{(i)}}{m/n} \overset{\text{a.s.}}{=} \frac{1}{n} \sum_{i=0}^{n-1} \mathbb{E}_{t' \sim \pi_n^{(i)}}[P(t_{\text{src}}|t')] \leq \max_{a \in \mathcal{A}} P(t_{\text{src}}|a). \tag{38}$$

The second equation follows by considering the Markov process induced over length $n$ strings and applying the Krylov–Bogolyubov argument for ergodic and homogeneous Markov processes.

In Lemma D.2, we show that with probability $\geq 1 - e^{-\Omega(\sqrt{n})}$, the token $t_{\text{src}}$ constructed from the source dataset satisfies, $\max_{a \in \mathcal{A}} P(t|a) \leq e^{-n H_\infty (1 - O(n^{-1/4}))}$. In other words, the source string has exponentially small probability. Combining this with eq. (38), with probability $\geq 1 - e^{-\Omega(\sqrt{n})}$ over the source dataset, the number of occurrences of the substring $t_{\text{src}}$ in a new string $s$ is upper bounded by,

$$\lim_{m \to \infty} \frac{n_{t_{\text{src}}}}{m} \overset{\text{a.s.}}{\leq} e^{-n H_\infty (1 - O(n^{-1/4}))} \triangleq \varepsilon/2n.$$

By the Krylov–Bogolyubov argument, for each $a \in \mathcal{A} = \{0, 1\}$, $\lim_{m \to \infty} \frac{n_a}{m} \overset{\text{a.s.}}{=} \pi(a)$. More importantly, the number of times $a$ is made as a token is upper bounded by $n_a$ and lower bounded by $n_a - n n_{t_{\text{src}}}$. Therefore,

$$(1 - \varepsilon)\pi(a) = \pi(a) - \frac{\varepsilon}{2} \overset{\text{a.s.}}{\leq} \lim_{m \to \infty} \frac{n_a}{m} \overset{\text{a.s.}}{\leq} \pi(a) = \frac{1}{2} \tag{39}$$

Finally, putting everything together,

$$\min_{Q \in \mathcal{Q}_{\text{1-gram}}} \lim_{m \to \infty} \frac{1}{m} \mathcal{L}_m(Q \circ \mathsf{enc}(\cdot)) = \min_{Q \in \mathcal{Q}_{\text{1-gram}}} \lim_{m \to \infty} -\frac{1}{m} \mathbb{E}\left[\log(Q_\#(|\mathsf{enc}(s)|)) + \sum_{t \in \text{Dict}} n_t \log Q_{\text{tok}}(t)\right]$$

$$\geq \min_{Q \in \mathcal{Q}_{\text{1-gram}}} \lim_{m \to \infty} -\frac{1}{m} \mathbb{E}\left[\sum_{a \in \mathcal{A}} n_a \log Q_{\text{tok}}(a)\right]$$

$$\overset{(i)}{\geq} \min_{Q \in \mathcal{Q}_{\text{1-gram}}} -(1 - \varepsilon) \sum_{a \in \mathcal{A}} \pi(a) \log Q_{\text{tok}}(a)$$

$$\geq (1 - \varepsilon) H(\pi).$$

where $(i)$ follows from the lower bound on $n_a/m$ in eq. (39). This completes the proof. □

**Lemma D.2.** *With probability $\geq 1 - e^{-\Omega(\sqrt{n})}$ over the source dataset,*

$$\max_{a \in \mathcal{A}} P(\boldsymbol{t}_{src}|a) \leq e^{-nH(\delta)(1 - O(n^{-1/4}))}.$$

*Proof.* Let $X$ denote the number of $i \in [n-1]$ such that $\boldsymbol{s}_i \neq \boldsymbol{s}_{i+1}$ in $\boldsymbol{s}$, the stochastic source. Since the transition of the Markov process only depends on whether the next character is the same as the previous character, we can write down,

$$\max_{a \in \mathcal{A}} \log P(\boldsymbol{t}_{\text{src}}|a) = -(X + 1) \log(\delta) - (n - 1 - X) \log(1 - \delta).$$

Note that $X$ is a sum of $n - 1$ i.i.d. random variables, since $\mathbb{I}(\boldsymbol{s}_i \neq \boldsymbol{s}_{i+1}) \sim \mathsf{Ber}(\delta)$ does not depend on whether $\boldsymbol{s}_i = 0$ or $= 1$. In particular, by Hoeffding's inequality, we have that with probability $\geq 1 - e^{-\Omega(\sqrt{n})}$,

$$\left|\frac{1}{n} \max_{a \in \mathcal{A}} \log P(\boldsymbol{t}_{\text{src}}|a) - H(\delta)\right| \leq O\left(n^{-1/4}\right),$$

which uses the fact that $\mathbb{E}[X] = \delta(n - 1)$ and $H_\infty = \delta \log(1/\delta) + (1 - \delta) \log(1/(1 - \delta))$. Taking an exponential on both sides proves the statement of the lemma. □

# E  Additional Theoretical Results IV: Interaction between the dictionary and encoding algorithm

In this section, we show another kind of barrier to generalization, which brings out the relationship between the encoding algorithm and the dictionary. We show that there exist dictionaries which generalize under the minimal encoder, i.e. the encoding algorithm which encodes a string into the shortest number of possible tokens, but at the same time, completely fail to generalize under the greedy encoder. This means that in the process of constructing good tokenizers, it does not suffice to think about the dictionary in isolation. Its interaction with the encoding algorithm is pertinent.

**Definition E.1** (minimal encoder). The minimal encoder parses a new string into the fewest possible number of tokens from the dictionary as possible. Ties are broken arbitrarily.

**Theorem E.2.** *There exists a stochastic source parameterized by $\delta \in (0, 0.5)$ and a dictionary* $\mathsf{Dict}$ *such that under the minimal encoder/decoder pair, the resulting tokenizer,* $\mathcal{T} = (\mathsf{Dict}, \emptyset, \mathsf{enc}_{min}(\cdot), \mathsf{dec}_{min}(\cdot))$ *generalizes near-optimally,*

$$\min_{Q \in \mathcal{Q}_{\text{1-gram}}} \mathcal{L}(Q \circ \mathsf{enc}_{min}(\cdot)) \leq 1.273 H_\infty. \tag{40}$$

*Here the entropy rate of the source, $H_\infty$, is $\delta \log(\sqrt{2}/\delta) + (1 - \delta) \log(1/(1 - \delta))$. However, the same dictionary* $\mathsf{Dict}$ *under the greedy encoder/decoder pair, i.e.* $\mathcal{T}' = (\mathsf{Dict}, \emptyset, \mathsf{enc}_{gre}(\cdot), \mathsf{dec}_{gre}(\cdot))$, *generalizes poorly, suffering from cross-entropy scaling as,*

$$\min_{Q \in \mathcal{Q}_{\text{1-gram}}} \mathcal{L}(Q \circ \mathsf{enc}_{gre}(\cdot)) \geq \frac{1 - o_\delta(1)}{3} H(\pi). \tag{41}$$

*where the entropy of the stationary distribution of the source is $H(\pi) = \frac{1}{2} \log(8)$ and the $1 - o_\delta(1)$ term is $(1 - \delta)^2 (1 + \delta)^{-1}$.*

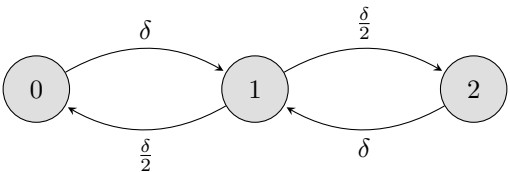

Figure 10: order-1 Markov source used in the proof of Theorem E.2

This means that the greedy encoder is not really compatible with the dictionary in the sense that the cross-entropy loss of the tokenizer is a constant multiple away from that achieved by the character-level tokenizer. The separation between eq. (40), and eq. (41) only manifests as $\delta$ becomes smaller and smaller.

In this section, we prove that generalization of a dictionary is a function of the underlying tokenization algorithm used. In particular, the greedy encoder is not universal, and there exists dictionaries under the minimum-length encoder/decoder which achieve small cross-entropy loss, which do not generalize under the greedy encoder/decoder.

We split the proof of Theorem E.2 into two parts. We first define the stochastic source and dictionary we consider. Then we show that under the minimum-length encoder, the asymptotic cross-entropy loss is upper bounded by $H_\infty$ up to a constant. Finally, we show that under the greedy-encoder, the same dictionary suffers from high cross-entropy loss, which is a constant factor away from that of the character encoder.

### E.1 Stochastic source and dictionary.

Consider an extension of the switching Markov source in example A.1 to $\mathcal{A} = \{0, 1, 2\}$. The Markov chain is described in Figure 10. The transition of the Markov chain is $P(0|0) = P(1|1) = P(2|2) = 1 - \delta$, and $P(1|0) = P(2|1) = \delta$ and $P(2|1) = P(0|1) = \delta/2$, with the remaining transitions being 0-probability. For a parameter $\ell > 0$ to be instantiated later, define $S_1$ (resp. $S_0$, $S_2$) as the set of all-1 (resp. all-0, all-2) strings of length $\leq \ell - 1$, including the empty string. Consider a dictionary composed of the following set of tokens, $\{1s : s \in S_0 \cup S_1 \cup S_2\}$. Therefore, the tokens follow the template $10 \cdots 0$, $11 \cdots 1$ or $12 \cdots 2$ and are of length at most $\ell$. $\ell$ is chosen to be $1 + 2\log(1/\delta)/\delta$.

Although we use the minimal encoder in the statement of Theorem E.2, for the purpose of analysis, define the following encoding algorithm: if the new string is prefixed by $10 \cdots 0$ or $12 \cdots 2$, select the largest prefix which exists in dictionary and assign it as a token. If the new string starts with a sequence $11 \cdots 1$ of length $x$, consider the first $\max\{\ell, x - 1\}$ length prefix and assign it as a token. Finally, if the string starts with 0 or 2, assign that character as token. Once the first token has been assigned, remove it and repeat.

### E.2 Minimal encoder achieves the optimal cross-entropy loss up to a constant.

First consider a simplification of the overall cross-entropy loss,

$$\min_{Q \in \mathcal{Q}_{\text{1-gram}}} \lim_{m \to \infty} \frac{1}{m} \mathcal{L}_m(Q \circ \text{enc}(\cdot))$$

$$= \min_{Q \in \mathcal{Q}_{\text{1-gram}}} \lim_{m \to \infty} -\frac{1}{m} \mathbb{E}\left[\log Q_\#(|\text{enc}_{\min}(s)|) + \sum_{t \in \text{Dict}} n_t \log Q_{\text{tok}}(t)\right] \tag{42}$$

$$\leq \lim_{m \to \infty} \frac{1}{m} \mathbb{E}\left[\log(m) + |\text{enc}_{\min}(s)| \log |\text{Dict}|\right], \tag{43}$$

where in the last inequality we upper bound by choosing $Q_\# = \text{Unif}([m])$ and $Q_{\text{tok}}(t) = 1/|\text{Dict}|$. Note that $|\text{Dict}| \leq 2\ell + 1$ and letting $\lim_{m \to \infty} \log(m)/m = 0$,

$$\min_{Q \in \mathcal{Q}_{\text{1-gram}}} \lim_{m \to \infty} \frac{1}{m} \mathcal{L}_m(Q \circ \text{enc}(\cdot)) \leq \lim_{m \to \infty} \frac{1}{m} \mathbb{E}[|\text{enc}_{\min}(s)| \log(2\ell + 1)]$$

$$\leq \lim_{m \to \infty} \frac{1}{m} \mathbb{E}\left[|\text{enc}(s)| \log(2\ell + 1)\right], \tag{44}$$

where in $(i)$, we replace $|\mathsf{enc}_{\min}(\boldsymbol{s})|$ by $|\mathsf{enc}(\boldsymbol{s})|$, which is the encoder we define in Appendix E.1. By definition of the minimal encoder, $|\mathsf{enc}_{\min}(\boldsymbol{s})| \leq |\mathsf{enc}(\boldsymbol{s})|$ surely. Recall that the encoder $\mathsf{enc}(\cdot)$ processes strings in a sequential (left-to-right) manner. In particular, by a similar argument as Lemma A.4, we can show that under this encoder, the limit $n_{\boldsymbol{t}}/\sum_{\boldsymbol{t}'} n_{\boldsymbol{t}'}$ almost surely converges to its expectation. More importantly, since, $\sum_{\boldsymbol{t} \in \mathsf{Dict}} |\boldsymbol{t}| n_{\boldsymbol{t}} = m$, we have that,

$$\lim_{m\to\infty} \frac{|\mathsf{enc}(\boldsymbol{s})|}{m} \overset{\text{a.s.}}{=} \frac{1}{\mathbb{E}_{\boldsymbol{t}\sim Q_{\mathrm{MLE}}}[|\boldsymbol{t}|]}.$$

converges to some limit almost surely. Therefore, from eq. (44),

$$\min_{Q\in\mathcal{Q}_{\text{1-gram}}} \lim_{m\to\infty} \frac{1}{m}\mathcal{L}_m(Q\circ\mathsf{enc}(\cdot)) \leq \operatorname*{ess\,limsup}_{m\to\infty} \frac{|\mathsf{enc}(\boldsymbol{s})|}{m}\log(2\ell+1). \tag{45}$$

where the essential lim-sup captures the almost sure limit $1/\mathbb{E}_{\boldsymbol{t}\sim Q_{\mathrm{MLE}}}[|\boldsymbol{t}|]$. The almost sure convergence of $|\mathsf{enc}(s)|/m$ also implies that we can let the limit $m$ go to $\infty$ in any manner, and the limit will remain the same. In particular, consider a process parameterized by $i^\star$ for generating the source string, such that surely $m \geq i^\star$, where the total number of characters, $m$, is a random variable. As $i^\star \to \infty$, we will also have $m \to \infty$ surely, and so the limit of $|\mathsf{enc}(\boldsymbol{s})|/m$ under this modified stochastic process should also converge to the same limit.

Rather than sampling a string of a fixed length $m$ from the source, consider the following sampling model: for $i^\star \to \infty$, sample $i^\star$ geometric random variables $X_1,\cdots,X_{i^\star} \overset{\text{i.i.d.}}{\sim} \mathsf{Geo}(\delta)$ and construct the source string as the concatenation of $i^\star$ strings alternating between successive 1's and successive 0's or 2's (with the choice between the two made uniformly at random), with the $i^{th}$ string of length $X_i + 1$. The overall number of characters sampled, $m$, is surely at least $i^\star$.

Under this stochastic process, the size of the encoding of the string is upper bounded by,

$$|\mathsf{enc}(\boldsymbol{s})| \leq |X_1 + 1| + \sum_{i=2}^{i^\star} \left(1 + (X_i + 1 - \ell)_+\right)$$

This bound follows from the fact that in any substring $\boldsymbol{s}'$ of successive 1's followed by a substring $\boldsymbol{s}''$ of successive 0's or 2's, the encoder tokenizes the first $\max\{\ell, |\boldsymbol{s}'|-1\}$ length prefix of $\boldsymbol{s}'$ as a token, and the remaining characters in $\boldsymbol{s}'$ into individual tokens except the last. Then, the last character of $\boldsymbol{s}'$ and the first $\max\{\ell-1, |\boldsymbol{s}''|\}$ characters of $\boldsymbol{s}''$ are assigned as token. The remainder of $\boldsymbol{s}''$ is assigned as individual tokens. Each of $\boldsymbol{s}'$ or $\boldsymbol{s}''$ of length $x$, is allocated into at most $1 + (x + 1 - \ell)_+$ tokens.

For any $i$, $\Pr(X_i \geq u) = (1-\delta)^u$, and therefore, summing over $u \geq \ell$, we get that $\mathbb{E}[(X_i+1-\ell)_+] = \frac{(1-\delta)^{\ell-1}}{\delta}$. With $\ell = 1 + 2\log(1/\delta)/\delta$, this expectation is upper bounded by $\delta$. Therefore,

$$\lim_{i^\star\to\infty} \frac{\mathbb{E}[|\mathsf{enc}(\boldsymbol{s})|]}{i^\star} \leq \lim_{i^\star\to\infty} \frac{1}{i^\star}\mathbb{E}\left[|X_1| + \sum_{i=2}^{i^\star}\left(1 + (X_i+1-\ell)_+\right)\right] \leq 1 + \delta$$

More importantly, by the strong law of large numbers for a sum of independent random variables, $(|X_1+1| + \sum_{i=2}^{i^\star}(1+(X_i+1-\ell)_+))/i^\star$, and therefore $|\mathsf{enc}(\boldsymbol{s})|/i^\star$ is asymptotically almost surely upper bounded as,

$$\lim_{i^\star\to\infty} \frac{|\mathsf{enc}(\boldsymbol{s})|}{i^\star} \overset{\text{a.s.}}{\leq} 1 + \delta, \tag{46}$$

On the other hand, the number of characters generated, $m$, equals $\sum_{i=1}^{i^\star}(X_i+1)$, and satisfies, $\lim_{i^\star\to\infty}\mathbb{E}[m]/i^\star = 1 + \delta^{-1}$. By another application of the strong law of large numbers for a sum of independent random variables,

$$\lim_{i^\star\to\infty} \frac{m}{i^\star} \overset{\text{a.s.}}{=} 1 + \delta^{-1}. \tag{47}$$

By combining eqs. (46) and (47), we have that,

$$\lim_{i^\star\to\infty} \frac{|\mathsf{enc}(\boldsymbol{s})|}{m} \overset{\text{a.s.}}{\leq} \frac{1+\delta}{1+\delta^{-1}} = \frac{1}{\delta}.$$

Finally, combining with eq. (45) and the ensuing discussion, we may upper bound the limiting cross-entropy loss by,

$$\min_{Q \in \mathcal{Q}_{\text{1-gram}}} \lim_{m \to \infty} \frac{1}{m} \mathcal{L}_m(Q \circ \mathsf{enc}(\cdot)) \leq \delta \log(2\ell + 1) = \delta \log(3 + 4\log(1/\delta)/\delta).$$

Note for this Markovian source, it is a short calculation to see that,

$$H_\infty = \mathbb{E}_{x \sim \pi}[H(P(\cdot|x))] = \delta \log(\sqrt{2}/\delta) + (1 - \delta) \log(1/(1 - \delta))$$

Note that for any $\delta \leq 1/2$, numerical evaluation gives the inequality,

$$1 \leq \frac{\delta \log(3 + 4\log(1/\delta)/\delta)}{H_\infty} \leq 1.273$$

with the approximation factor improving as $\delta$ becomes smaller. Therefore, this tokenizer achieves a normalized cross-entropy loss which asymptotically scales as a constant multiple of the entropy rate of the source.

### E.3 Greedy-encoder achieves poor cross-entropy loss

Note that the greedy encoder picks the largest prefix of the string which is a token, assigns and removes it, and iterates on the rest of the string. The greedy encoder's behavior is easy to analyze - every string of consecutive 1's in the new string is broken into chunks of length $\ell$ (save potentially the last chunk) and each chunk is assigned as a token in $\{1s : s \in S_1\} \subset \mathsf{Dict}$. If the length of this substring of successive 1's is not $1, \ell + 1, 2\ell + 1, \cdots$, or in general, $\equiv 1 \mod \ell$, every character in the next sequence, composed of 0's or 2's is tokenized into individual characters.

Similar to eq. (42) to eq. (43), consider a simplification of the overall cross-entropy loss,

$$\min_{Q \in \mathcal{Q}_{\text{1-gram}}} \lim_{m \to \infty} \frac{1}{m} \mathcal{L}_m(Q \circ \mathsf{enc}_{\text{gre}}(\cdot))$$

$$= \min_{Q \in \mathcal{Q}_{\text{1-gram}}} \lim_{m \to \infty} -\frac{1}{m} \mathbb{E}\left[\log Q_{\#}(|\mathsf{enc}_{\text{gre}}(s)|) + |\mathsf{enc}_{\text{gre}}(s)| \sum_{t \in \mathsf{Dict}} \frac{n_t}{|\mathsf{enc}_{\text{gre}}(s)|} \log Q_{\text{tok}}(t)\right]$$

$$\geq \min_{Q \in \mathcal{Q}_{\text{1-gram}}} \lim_{m \to \infty} -\frac{1}{m} \mathbb{E}\left[|\mathsf{enc}_{\text{gre}}(s)| \sum_{\substack{t \in \mathsf{Dict} \\ Q_{\text{MLE}}(t) > 0}} Q_{\text{MLE}}(t) \log Q_{\text{tok}}(t)\right],$$

where the last equation uses the fact that by Lemma A.4, for the greedy encoder, $\lim_{m \to \infty} \frac{n_t}{|\mathsf{enc}_{\min}(s)|} \overset{\text{a.s.}}{=} Q_{\text{MLE}}(t)$. The minimizer of this objective subject to $\sum_{t \in \mathsf{Dict}: Q_{\text{MLE}}(t) > 0} Q_{\text{tok}}(t) \leq 1$ is $Q_{\text{tok}}(t) = Q_{\text{MLE}}(t)$ resulting in the inequality,

$$\min_{Q \in \mathcal{Q}_{\text{1-gram}}} \lim_{m \to \infty} \frac{1}{m} \mathcal{L}_m(Q \circ \mathsf{enc}_{\text{gre}}(\cdot)) \geq \lim_{m \to \infty} \frac{1}{m} \mathbb{E}\left[|\mathsf{enc}_{\text{gre}}(s)| H(Q_{\text{MLE}})\right], \tag{48}$$

where we use the convention $0 \log(1/0) \triangleq \lim_{P \to 0} P \log(1/P) = 0$ and therefore we may sum over tokens such that $Q_{\text{MLE}}(t) = 0$ for free.

Considering the same geometric sampling model as in Appendix E.2, and Lemma A.4, we may study the almost sure limit $Q_{\text{MLE}}(t) = \lim_{m \to \infty} n_t/|\mathsf{enc}_{\text{gre}}(s)|$ by computing $\lim_{i^\star \to \infty} n_t/|\mathsf{enc}_{\text{gre}}(s)|$ under the geometric sampling model since the almost sure limit exists. Recall that in the geometric sampling model, we generate the overall source string by concatenating $i^\star$ strings of length $X_1 + 1, \cdots, X_{i^\star} + 1$ where $X_i \sim \mathsf{Geo}(\delta)$, with the strings alternating between successive 1's and successive 0's or 2's (with the choice between the two made by the flip of a fair coin). For $x \in \{0, 1, 2\}$, let $\mathcal{E}_i(x)$ denote the event that $X_i$ is a string composed only of all $x$'s. The length of the greedy encoding of $s$ is lower bounded by,

$$|\mathsf{enc}_{\text{gre}}(s)| \geq \sum_{i=1}^{i^\star} X_i \cdot \mathbb{I}(X_{i-1} \not\equiv 1 \mod \ell)\mathbb{I}(\mathcal{E}_i(0) \cup \mathcal{E}_i(2)). \tag{49}$$

Which captures for the fact that all 0's and 2's are encoded into singular tokens unless the previous string of 1's was of length $\equiv 1 \mod \ell$. By the law of large numbers of the RHS of eq. (49), the following a.a.s. lower bound is satisfied,

$$\lim_{i^\star \to \infty} \frac{|\mathsf{enc}_{\mathrm{gre}}(s)|}{i^\star} \overset{\text{a.s.}}{\geq} \frac{1}{2\delta} \left( 1 - \sum_{u=0}^{\infty} \delta(1-\delta)^{\ell u+1} \right) = \frac{1}{2\delta} \left( 1 - \frac{\delta(1-\delta)}{1-(1-\delta)^\ell} \right) \geq \frac{1-\delta}{2\delta}, \quad (50)$$

where the last inequality uses the fact that $\ell = 1 + 2\log(1/\delta)/\delta$. Likewise, observe that, $|\mathsf{enc}_{\mathrm{gre}}(s)| \leq m$ surely, and following the analysis in Appendix E.2 of eq. (47), we have that,

$$\lim_{i^\star \to \infty} \frac{|\mathsf{enc}_{\mathrm{gre}}(s)|}{i^\star} \leq \lim_{i^\star \to \infty} \frac{m}{i^\star} \overset{\text{a.s.}}{=} 1 + \delta^{-1}. \quad (51)$$

For $x \in \{0, 2\}$, observe that the expected number of times the token $x$ is observed in the encoding of $s$, $n_x$ can be written as,

$$n_x \geq \sum_{i=1}^{i^\star} \left( (X_i + 1) \cdot \mathbb{I}(X_{i-1} \not\equiv 1 \mod \ell) \right) \mathbb{I}(\mathcal{E}_i(x)). \quad (52)$$

In particular, taking the expectation of eq. (52),

$$\mathbb{E}[n_x | \mathcal{E}_1(0) \cup \mathcal{E}_1(2)], \ \mathbb{E}[n_x | \mathcal{E}_1(1)] \geq \frac{i^\star - 1}{4}(1 + \delta^{-1}) \left( 1 - \sum_{u=0}^{\infty} \delta(1-\delta)^{\ell u+1} \right) \geq \frac{i^\star - 1}{4} \cdot \frac{1 - \delta^2}{\delta}. \quad (53)$$

Note that in any realization of the geometric sampling process, in eq. (52), either the odd indexed substrings are all-1's or the even indexed substrings are all-1's. Therefore, surely, all the non-zero terms in the above summation are of the same parity. Moreover, since the $i^{th}$ term in the sum only depends on $X_i$ and $X_{i-1}$, conditioned on whether the non-zero parities are even or odd, $n_x$ can be written as a sum of $\approx i^\star/2$ mutually independent terms. By the strong law of large numbers on each of the conditional processes, eqs. (52) and (53) implies that for $x \in \{0, 2\}$,

$$\lim_{i^\star \to \infty} \frac{n_x}{i^\star} \overset{\text{a.s.}}{\geq} \frac{1 - \delta^2}{4\delta}.$$

To upper bound $n_x$, note that it is upper bounded by the number of times the character $x$ appears in the source string, which by the strong law of large numbers a.a.s (after normalizing by $i^\star$), scales as $1/4\delta$. Finally, to bound $Q_{\mathrm{MLE}}(t)$ which is the sequential nature of the encoder, using a similar proof as Lemma A.4, we can show that $n_t / \sum_{t'} n_{t'}$ converges to the unigram MLE model for this tokenizer. For the token $x \in \{0, 2\}$,

$$\lim_{i^\star \to \infty} \frac{n_x}{|\mathsf{enc}(s)|} = Q_{\mathrm{MLE}}(x) \leq \mathbb{E} \left[ \lim_{i^\star \to \infty} \frac{n_x}{n_2 + n_0} \right] \quad (54)$$

Using the a.a.s. upper and lower bounds on $|\mathsf{enc}(s)|$, $n_0$ and $n_2$ derived in eqs. (51) and (54), we arrive at lower and upper bounds on $Q_{\mathrm{MLE}}(x)$ for $x \in \{0, 2\}$,

$$\frac{1}{4} \approx \frac{1-\delta}{4} = \frac{(1-\delta^2)}{4\delta(1+\delta^{-1})} \leq Q_{\mathrm{MLE}}(x) \leq \frac{1}{2(1-\delta^2)} \approx \frac{1}{2}.$$

Since there are at least two tokens having probability bounded away from 0 and 1 by a constant under the MLE unigram model, the entropy of $Q_{\mathrm{MLE}}$ must also be lower bounded by a constant. Indeed,

$$H(Q_{\mathrm{MLE}}) \geq 2 \min_{\frac{1-\delta}{4} \leq y \leq \frac{1}{2(1-\delta^2)}} y \log(1/y).$$

It is easy to verify that for $\delta \leq 0.5$, the minimizer is achieved at $y = \frac{1-\delta}{4}$, which leads to the lower bound,

$$H(Q_{\mathrm{MLE}}) \geq \left( \frac{1-\delta}{2} \right) \log \left( \frac{4}{1-\delta} \right).$$

| | |
|---|---|
| Architecture | GPT-2 |
| Batch size | Grid-searched in $\{8, 16, 32\}$ |
| Gradient acc. steps | 1 |
| Tokenizer dictionary size | $\{10, 20\}$ |
| Tokenizer dataset size | $10,000$ |
| Optimizer | AdamW ($\beta_1 = 0.9, \beta_2 = 0.95$) |
| Learning rate | 0.002 |
| Scheduler | Cosine |
| # Iterations | 8000 |
| Weight decay | $1 \times 10^{-3}$ |
| Dropout | 0 |
| Sequence length | 512 |
| Embedding dimension | Grid-searched in $\{10, 20, 30, 40\}$ |
| # layers | Grid-searched in $\{1, 2, 4, 8\}$ |
| # heads | Grid-searched in $\{1, 2, 4, 8, 16\}$ |
| Repetitions | 5 |

Table 3: Hyperparameter choices

Finally, combining this lower bound on $H(Q_{\mathrm{MLE}})$ with eq. (48), we have that,

$$
\begin{aligned}
\min_{Q \in \mathcal{Q}_{\text{1-gram}}} \lim_{m \to \infty} \frac{1}{m} \mathcal{L}_m(Q \circ \mathsf{enc}(\cdot)) &= \lim_{i^\star \to \infty} \mathbb{E}\left[ \frac{|\mathsf{enc}_{\mathrm{gre}}(\boldsymbol{s})|}{m} H(Q_{\mathrm{MLE}}) \right] \\
&\geq \lim_{i^\star \to \infty} \mathbb{E}\left[ \frac{|\mathsf{enc}_{\mathrm{gre}}(\boldsymbol{s})|}{m} \right] \cdot \left( \frac{1-\delta}{2} \right) \log\left( \frac{4}{1-\delta} \right) \\
&\overset{(i)}{\geq} \frac{1-\delta}{2\delta(1+\delta^{-1})} \cdot \left( \frac{1-\delta}{2} \right) \log\left( \frac{4}{1-\delta} \right) \\
&\geq \frac{(1-\delta)^2}{3(1+\delta)} H(\pi)
\end{aligned}
$$

where $(i)$ follows from the lower bound on $|\mathsf{enc}_{\mathrm{gre}}(\boldsymbol{s})|$ in eq. (50) with the almost sure limit of $m$ in eq. (47) and noting that $|\mathsf{enc}_{\mathrm{gre}}(\boldsymbol{s})|/m \leq 1$ surely. The last inequality follows by simplifying using $\pi = (1/4, 1/2, 1/4)$ and $H(\pi) = \frac{1}{2}\log(8)$.

## F   Experiment details

**Experiment 1 (Figures 4a and 4b).**   In this and previous experiments (Figures 2, 3a and 3b), we train the transformers on a single GPU on an $8\times$ A100 node. The wall-clock time measured does not count time spent in validation loss evaluations. The hyperparameter choices are listed in Table 3.

**Experiment 2 (Table 1).**   We evaluate pre-trained tokenizers on various datasets. In this experiment, we do not evaluate the likelihood model on test sequences, rather, we estimate the cross-entropy of the best unigram model by using the approximation,

$$
-\mathbb{E}\left[ \sum_{\boldsymbol{t} \in \mathsf{Dict}} n_{\boldsymbol{t}} \log Q_{\mathrm{MLE}}(\boldsymbol{t}) \right] \approx -\sum_{\boldsymbol{t} \in \mathsf{Dict}} \widehat{n}_{\boldsymbol{t}} \log(\widehat{Q}(\boldsymbol{t})) \tag{55}
$$

where $\widehat{Q}(\boldsymbol{t}) = \frac{\widehat{n}_{\boldsymbol{t}}}{\sum_{\boldsymbol{t}} \widehat{n}_{\boldsymbol{t}}}$ is the MLE unigram model learnt from a finite dataset, which we choose here as GLUE (Wang et al., 2019), and $\widehat{n}_{\boldsymbol{t}}$ is the number of times the token $\boldsymbol{t}$ is observed in the encoding of the dataset. This approximation allows us to separate the error stemming from learning a suboptimal likelihood model which tends to have higher sample complexity requirements and focus on the asymptotic error of the tokenizer.

We use Monte-carlo sampling to approximate the cross-entropy loss estimator in eq. (55). These approximations tends to underestimate the true cross-entropy loss due to the concavity of $x \log(1/x)$ close to $0$. In general, the gap between the approximation and the true error is expected to grow with $k$. Therefore, the true difference between the estimate of the best unigram model on a tokenizer and the best $k$-gram model for $k \geq 2$ on the character level tokenizer is likely to be larger than the reported figures.

**Experiment 3 (Figure 5).** We train the LZW, BPE, Unigram and Wordpiece tokenizers with dictionary sizes $\{5000, 6000, 8000, 12000, 20000, 32000, 50000, 80000\}$. The cross-entropy loss incurred by the best 1-gram model is estimated using eq. (55) while for $k$-gram models for $k \geq 2$, we use Monte-carlo sampling to estimate the cross-entropy of the empirical $k$-gram model computed using the GLUE dataset. For the $k$-gram models trained on the character level tokenizer, since the vocabulary size is fixed, we instead plot the number of distinct $k$-grams on the $x$-axis. While this is not a true measure of the number of parameters in the underlying $k$-gram model, we use this as a proxy for the same.

