# OpenReview forum: "An Analysis of Tokenization: Transformers under Markov Data"
_NeurIPS.cc/2024/Conference — NeurIPS 2024 spotlight_

### Official Review · Reviewer_RvTD · 2024-07-13

**Soundness:** 3
**Presentation:** 3
**Contribution:** 3
**Rating:** 7
**Confidence:** 3

**Summary:**

This paper presents a study on tokenization by investigating the behavior of transformers on simple data generating processes . It shows that, in the absence of any tokenization, transformers trained on $k$th-order Markov processes predict characters according to a unigram model, which is quite problematic given how poor unigram models are at modeling Markovian data. Paradoxically, they observe that, even the simplest unigram model learnt by transformers *with the appropriate tokenization* is able to model the probability of sequences sampled from a $k$th-order Markov process.

**Strengths:**

- The paper is well written, with empirical observation intermingled with theory, which I quite liked. The theory is also accompanied by a lot if intuition, insight and interpretation, which really helps drive the point home.

**Weaknesses:**

- In section 3.2, the authors chose to focus on developing guarantees for a newly developed tokenizer, which, to my knowledge, is seldom used. It would've been maybe of greater use to the community to also, or instead, establish these guarantees for the more commonly-used tokenizers, such as BPE.

- I appreciate that this is mostly a theoretical study of tokenizers, and while the observations put forward are valuable, I found myself wondering what practical takeaways this paper presents to improve current tokenizers. That is something I would love the authors to comment on.

**Questions:**

Please see the Weaknesses section above

**Limitations:**

yes

---

> ### Author Rebuttal · Authors · 2024-08-07
>
> We thank the reviewer for the feedback and questions. Below we addressed the key points mentioned.
>
> ### **[W1] Analysis for commonly-used tokenizers such as BPE**
>
> The guarantees in section 3.2 study guarantees for the LZW tokenizer, which is arguably not used much in practice. However, in Section B of the Appendix (Additional Theoretical Results I: A sequential variant of BPE) we study a variant of BPE which is analytically tractable, and present a result similar to Theorem 3.1 for this tokenizer (cf. Theorem B.2). The discussion of this result in the main paper was deferred to a short excerpt in Section 4.1 due to a lack of space, but in the subsequent version of the paper, we shall include a longer discussion. This was indeed our motivation to analyze (a tractable variant of) BPE as well - it remains one of the most commonly employed tokenizers in practice across a number of domains.
>
> ### **[W2] Practical takeaways**
>
> We point the reviewer to the response to [W3] of Reviewer veC3 where we have summarized the key empirical takeways of the paper.

---

> > ### Comment · Reviewer_RvTD · 2024-08-12
> >
> > Thank you for addressing my questions as well as the questions and concerns raised by the other reviewers. I think this is a good paper, and will raise my score accordingly.

---

### Official Review · Reviewer_zrKY · 2024-07-13

**Soundness:** 3
**Presentation:** 2
**Contribution:** 3
**Rating:** 6
**Confidence:** 2

**Summary:**

The authors show that tokenization is a fundamental property of transformer-based models, in the sense that without it, it is hard (if not impossible) to achieve low cross-entropy loss on next-word prediction. They show that tokenization helps breaking the unigram barrier (i.e., the best loss a unigram model can achieve) and give a theoretical characterization of the information tokenizers provide in terms of statistics on token distribution.

In particular:

Section 2.1 shows how models without tokenization cannot achieve the optimal cross-entropy loss, while when equipped with a tokenizer they break the unigram barrier.

Section 3 studies tokenizers that assign all possible substrings (up to length r) as tokens in the dictionary and shows their theoretical optimality in learning processes ruled by k-Markov chains. A consequence is that unigram models can also do that, in the limit.

Of course, this comes at the expense of the model's efficiency and potential attacks that one can run on an exponential number of tokens (i.e., the surface attack grows very large).

Finally, the authors show that tokenizers can trade off the vocabulary size while maintaining low cross-entropy (i.e., they can behave like an optimal model).

Finally, they extend the theoretical framework to LZW tokenizers.

Experiments are conducted on tokenized vs. non-tokenized models on {k=1}-Markov models and then on some real datasets to show that tokenizers trade-off complexity and efficiency in learning an optimal representation of the characters (and their frequency) in the training distribution.

**Strengths:**

The article studies an important problem, and I think there is value in the paper.
To the best of my knowledge, comparing BPE to non-tokenized models is new, and the figures give some interesting insights (e.g., Figure 2).
Your paper contains much theoretical work, contributing to its quality.
The results in the limit for unigram and BPE/LZW models are noticeable (Section 3 and Eq. 2).

In general, the results seem solid and are also interesting for linguists and NLP researchers. BPE and other tokenization methods find a trade-off between unigram models, as per Eq. 2, and the complexity of the resulting vocabulary (and model).

**Weaknesses:**

One of the main weaknesses of this work is how it is presented.
Maybe it's me, but I found it quite hard to read. See questions.

Another concern is how theoretical results apply to real-world datasets. See questions, but Fig. 5 seems to mitigate the impact of your theoretical results.
In fact, for the vocabulary that grows larger, all the models have a similar value of cross entropy (for around ~50K tokens).

The article seems rushed, as there are many typos (I just listed some).
- Line 150 “the a”
- Line 173, “it make since” --> “sense”
- Line 175, eq. and many others --> Eq. (it’s not wrong per-se, but you capitalize Fig, Example, etc.)
- The Notation paragraph shouldn’t go with related works but should be in the next section.
- Notation in 2 is a bit sloppy (this is a personal suggestion): you can use D() and E() for the decoder/encoder (and enclose them with \mathcal).

**Questions:**

You say that Transformers without tokenization fail at modelling simple k-order Markov models, while with BPE (and other techniques), they succeed. I would say that is simply because BPE injects statistical information in the data and splits it accordingly. BPE is "trained" to aggregate "bytes" according to their frequency in the training data, so it somehow informs a model with the stationary distribution of most-frequent contiguous characters.
Am I missing any non-trivial observation here?

There is a reference in the Appendix to the model used (GPT-2), but nothing on the main paper. For example, I asked myself multiple times what models you used for the tokenized and non-tokenized networks.

By unigram models, do you mean those where the probability of each token approximates the inverse frequency of a character/token in the training data?

If I understand correctly, in Fig. 2 (a), models without tokenization fail at breaking the unigram barrier (so the best they can do is model the inverse character frequency). How does that connect to the relative success of character-level tokenization? There are plenty of methods that use character-level tokenization, and they probably work much better than unigrams. Is there anything I am missing here?

In Figure 2 you mention that plot (2b) has 70x less parameters, but you do not specify why (Is it to prove tokenization helps?).
Do you use GPT-2, as mentioned in the Appendix? If so, do you use a smaller model for the figure on the left and a larger one for the one on the right?

Fig. 3 is hard to understand. I read it many times, but I still do not fully understand what it conveys. The heatmap goes from 0. to ~0.6, though it is unclear what the measure is (I guess it is the probability?).

**Limitations:**

Please see previous sections.

---

> ### Author Rebuttal · Authors · 2024-08-07
>
> We thank the reviewer for the feedback. Below we address the weaknesses/questions pointed out:
>
> ### **[W1] Fig. 5 mitigates the impact of the theory**
>
> On datasets like Wikitext-103, transformers trained with these tokenizers (BPE, Unigram, Wordpiece) indeed perform similarly in ablations. The purpose of this figure was to observe that the test-loss appears to decay as roughly $\propto 1/\log(|\text{Dict}|)$ agreeing with our theory (Theorem 3.1). When we generalize beyond these settings to more complex languages and more specific downstream tasks, the differences start becoming noticeable. There are many possible reasons and counterfactuals to control for in these kinds of experiments. For instance, Wordpiece appears to be worse compared to Unigram and BPE in languages which are not space-delimited like Japanese [3]. Unfortunately, these questions are at a granularity which can't be addressed by studying Markovian data models. We believe that the extensions of our work to more specific losses / different data-distributions can play a role, and help in revealing fine-grained differences between tokenizers.
>
> ___
>
> ### **[Q1] BPE succeeds because of aggregating data according to frequencies**
>
> There are plenty of seemingly “reasonable” ways in which a tokenizer may aggregate data according to their frequencies while constructing the dictionary, but which ultimately end up performing poorly. Consider the following modification of the Wordpiece tokenizer where we merge the pair of tokens which maximize $n_{t_1, t_2} / n_{t_1}^2$ (the original Wordpiece prescribes merging the maximizer of $n_{t_1,t_2} / n_{t_1} n_{t_2}$). While this tokenizer indeed seems reasonable per the above definition, it turns out that it performs quite poorly. Training on randomly generated letters from the english alphabet, the results are tabulated in Table 1 of the attached pdf.
>
> Notice the clear a pattern in how modified Wordpiece behaves: it appears to merge the longest token in the dictionary thus far with a single character in each round. While the tokenizer “objective” itself does not prescribe this behavior, the tokenizer exhibits this trend because of the $n_{t_1}^2$ term in the denominator. This incentivizes the merge to pick a starting token ($t_1$) which has very low frequency (because of the $1/n_1^2$ term in the denominator).
>
> Thus the behavior of tokenization / design of appropriate merge rules is a nuanced phenomenon. While there appear to be natural choices under which the intuition “tokenizer is trained to aggregate bytes according to their frequencies $\implies$learning unigram models are sufficient” is true, there are also cases where this intuition fails altogether.
>
> On a separate note, it is not apparent how to design tokenizers which learn dictionaries with the *minimal number of tokens* required to achieve low loss (under unigram likelihood models). Our proofs show this is true for LZW and approximately so for a variant of BPE. Thus, these tokenizers are hard to improve on significantly (without breaking some sort of worst-case information theoretic barrier). Having a small dictionary is important for being able to learn the transformer in a data-efficient manner, and to avoid redundant tokens, which presents an attack surface.
>
> ### **[Q2] Model used in the experiments; Clarification of Fig. 2**
>
> In the paper, we used variants of the GPT-2 architecture as implemented in [13]. We shall add this citation to the paper. Our experiments considered variations over (i) the number of heads, (ii) the number of layers, and (iii) the embedding dimension and (iv) the dictionary size (excluding optimization hyperparameters). We shall include these in the description of figures. They are summarized in the Appendix (Table 3) due to a lack of space in the main paper.
>
> **Fig 2 clarification:** Fig. 2b uses $70 \times$ fewer parameters compared to 2a. This is an argument supporting that models with tokenization do not require as many parameters to quickly achieve the optimal test-loss. This is a less important point at the scale of models we considered, which is why we do not go into too much detail, but this also translates to an improvement in wall-clock time (cf. Fig. 4 in the paper)
>
> ### **[Q3] What are unigram models?**
>
> Unigram models are defined on lines 114-116 in the paper.  The specific model mentioned (probability of each token approximates the inverse frequency of character/token) lies in the class of unigram models we consider.
>
> ### **[Q4] Success of transformers beyond the unigram barrier?**
>
> Our reason for studying unigram models is not because this class of models are interesting in their own right (which one can argue for), but more so because we see that transformers empirically learn these kinds of models when trained on Markov data. This is the role of Fig. 3 (clarified in the common rebuttal). However, when the data processes grow to be more complicated than Markov models, we expect transformers to exhibit more complex behavior (beyond $n$-grams). Understanding the behavior of transformers with and without tokenization on more complex data processes is a rich direction for future work.
>
> On a separate note, character-level tokenization does appear to work to varying degrees in practice with transformers, and this is likely because real world data is a mixture of many different data types. For certain languages (such as those containing non-concatenative morphology), removing tokenization seems to help [14]. For arithmetic, character level tokenization is known to help models generalize better [2]. But from these results, it is not clear whether these are pitfalls of existing tokenizers, or of tokenization as a whole. On arithmetic, for instance, recent models like GPT4 and LLama3 appear to chunk numbers into the range 0-999 and split longer numbers into these chunks, instead of operating at the single digit/byte level, which is in support of the argument, “Good tokenizer $\ge$ no tokenizer”.

---

> > ### Comment · Reviewer_zrKY · 2024-08-10
> >
> > I thank the reviewers for their clarifications. I am convinced by their arguments, and I appreciate the comment where they address multiple concerns raised by different reviewers. I will keep my score.

---

### Official Review · Reviewer_veC3 · 2024-07-15

**Soundness:** 4
**Presentation:** 3
**Contribution:** 2
**Rating:** 6
**Confidence:** 3

**Summary:**

This paper offers theoretical insights into the importance of tokenization in language models. Tokenization is ostensibly the artifact that makes training LMs not an end-to-end procedure. This design choice introduces biases, as it is not optimized for exactly the same criterion as the full model. Yet training without a tokenization step almost always leads to worse language models. This paper attempts to provide reasons based in probability theory for this phenomenon. The authors first explore a toy setting, in which transformer models are tasked with predicting distributions from kth order Markov processes. They offer a theoretical explanation for why the error of models is capped at that of a unigram model and how tokenization alleviates this issue. They then show that tokenization schemes with certain properties can achieve the optimal cross-entropy loss. The work offers some basic experimental results confirming their insights.

**Strengths:**

* Tokenization is a core part NLP pipelines yet it still needs to be better understood from a theoretical perspective. The questions that this paper tries to answer are very relevant for both model interpretability and further development
* The theory is presented in an understandable manner and results for specific popular tokenization schemes are provided.

**Weaknesses:**

* The theory presented in this work is for a specific type of data-generating distribution (kth order Markov) and we can’t directly extrapolate these results to linguistic distributions, which do not necessarily follow such a distribution. There is minimal discussion about the relationship between kth order Markov and linguistic distributions, which leaves the reader questioning how relevant these results actually are.
* Ultimately, the results are limited; they essentially show an expected result (the existence of an optimal unigram language model as the dictionary size grows to infinity). While some intuition can be gained from these results, the theoretical implications are limited.
* There is minimal discussion of the empirical results and what conclusions can be drawn from them. Given how much of the theory is not directly applicable to real language modeling settings, it feels like such a discussion should be very important

**Questions:**

* In the kth-order Markov processes studied, are nth state distributions dependent only on the n-kth state? I may be misunderstanding the caption in figure 1
* The results are applicable to all language models, not just large ones. If anything, they are arguably more relevant for smaller language models. Perhaps consider changing the title
* How does the work differ from/build on Edelman et. al. 2024 and Makkuva et. al. 2024?

**Limitations:**

Limitations are not discussed in depth. The authors should address their limited experimental setting and the applicability of the results to linguistic distributions (which are not evidently k-order Markovian)

---

> ### Author Rebuttal · Authors · 2024-08-07
>
> We thank the reviewer for the feedback. Below we address the main questions and weaknesses pointed out. *(references cited in the common rebuttal)*
>
> ### **[W1] How well do $k$-th order Markov processes extrapolate to linguistic distributions?**
>
> There are two points to mention here,
>
> 1. $k$-th order Markov processes, while not perfect, capture many elements of linguisitic distributions - when $k$ grows larger, these models capture longer-range dependencies with the past. This class of models have had a rich history in language modeling and nearly every book on language modeling studies these processes. These models have been the study of many recent works, since they present a rich test-bed of interesting phenomena. A few recent works which have appeared in the literature since submission which study the interplay of transformers with these kinds of distributions are [8],[9].
> 2. This data-model, while simple, captures the ability of transformers to learn $n$-gram behavior. $n$-gram features appear prominently in many domains such as NMT [10], language modeling [11] and speech/music [12] to name a few.
>
> ### **[W2] Results show an expected result (asymptotically optimal unigram model)**
>
> The existence of an optimal unigram model as the dictionary size grows to infinity is not the main contribution of the paper. It is easy to show that there exists “some” tokenizer which satisfies this property. In this regard, two main contributions of our paper are,
>
> 1. From a theoretical point of view, we show that the LZW tokenizer in fact lies on the pareto frontier of test-loss vs. dictionary size, when the likelihood model is a unigram model. To a lesser extent, this is approximately true for the BPE tokenizer as well. This means that the tokenizer not only asymptotically works, but non-asymptotically, the size of the dictionary cannot be reduced significantly without hurting the test-loss.
> 2. The tokenizer which assigns all $r$-length tokens also satisfies the property that asymptotically there exists an optimal unigram model on it. However, this is an unnatural tokenizer. We show that the above properties are true for BPE and LZW: letting the number of tokens grow to $\infty$, it is a-priori not at all obvious that on the learnt dictionaries there exists a unigram model which achieves optimal test loss.
> In contrast, it is easy to come up with seemingly “natural” tokenizers but no unigram model trained on these tokenizers can asymptotically achieve the optimal test loss. In the response to [Q1] of Reviewer zrKY, we provide a surprising example of such a tokenizer.
>
> ### **[W3] Discussion of empirical results**
>
> Here is a quick summary of the empirical results in the paper, and takeaways.
>
> 1. **Fig. 2 and Fig. 4:** Transformers need to be exposed to much more data to learn $k$-th order Markov models without tokenization, compared to with tokenization. Likewise, they learn more quickly wrt wall-clock time compared to in the absence of tokenization (This is true even if we select the fastest untokenized model).
> **Takeaway:** Transformers generalize better with tokenization, both in terms of number of samples as well as number of optimization iterations.
> 2. **Fig. 3 conclusion:** Transformers approximately learn unigram models when trained on tokenized sequences generated from a Markov model.
> **Takeaway:** Transformers can achieve low loss by learning conceptually simple models when tokenization is present.
> 3. **Fig. 5 conclusion:** BPE, Unigram and Wordpiece tokenizers perform similarly on Wikitext-103 with unigram models. The decay of the loss as a function of the dictionary size appears to be captured by $\propto 1/\log(|\textsf{Dict}|)$, as predicted theoretically (Theorem 3.1).
> **Takeaway:** Scaling law for tokenization (as a function of the dictionary size).
>
> In this paper, we sought to build up a theoretical understanding of how tokenizers and transformers interact by studying their behavior on $k$-th order Markov chains. While these models are not perfect, a number of interesting empirical phenomenon can be inferred which concur with practice, even through the lens of these simple data models. More importantly, the questions and workflow considered in this paper (analyzing end-to-end loss on models learned by transformers) can be employed with more complex data processes, where transformers exhibit richer behavior. Separations between tokenizers may become more apparent here.
>
> ___
>
> ### **[Q1] $k$-th order processes considered**
>
> In the $k$-th order processes studied, the $n$-th state distribution only depends on the $(n-k)$-th state. However, our theory applies for all $k$-th order processes (as long as Assumption 3.2 is satisfied). In our experiments, we chose to compare against this restricted class of models to level the playing field between different values of $k$ (since the problem certainly becomes harder if we consider all Markov chains for increasing $k$). In the attached pdf, we plot the behavior of tokenizers on randomly chosen $k$-th order Markov processes for $k=1,2,3,4$. The gap between the tokenized and untokenized grows even more significantly as $k$ increases.
>
> ### **[Q3] Comparison with Edelman et al (2024) and Makkuva et al (2024)**
>
> These works show that transformers (in the absence of tokenization), when exposed to sufficiently much data, learn to represent $k$-th order Markov chains in-context for relatively small values of $k$. In practice, however, tokenization is almost always used. In our paper, we study the questions asked in these papers when transformers are trained with tokenization. Our key observations is that with tokenization, the model learns significantly more quickly, and is able to avoid going through multiple phases of learning when trained with tokenization. The general rhetoric surrounding tokenization has been negative, and our work shows that tokenization can make a big difference, even when learning simple models like $k$-th order Markov chains.

---

> > ### Comment · Reviewer_veC3 · 2024-08-11
> >
> > Thank you for the detailed response. The proposed changes in the main response would raise my score slightly. I adjust to reflect that.

---

### Official Review · Reviewer_88v6 · 2024-07-15

**Soundness:** 3
**Presentation:** 2
**Contribution:** 3
**Rating:** 7
**Confidence:** 3

**Summary:**

This paper investigates the learning dynamics of unigram language models on top of tokenised vs non-tokenised data, comparing these models’ expected cross-entropy to the distribution’s entropy. The paper performs this analysis while considering different data generating distributions (mainly focusing on relatively simple markov chains), and different tokenisation methods.

**Strengths:**

This paper tackles an interesting and timely topic: how tokenisation enables language modeling.

This paper provides an interesting theoretical analysis of the effect of tokenisation on unigram language modeling.

This paper also provides a couple of empirical analyses of how unigram models perform on real data.

The paper is relatively easy to follow, even though some of the mathematical results could be spelled out a bit more clearly to make it easier for a reader to follow them.

**Weaknesses:**

This paper’s framing, in my opinion, significantly over-claims its results:
* The title “Toward a Theory of Tokenization in LLMs” is very broad for the current contributions. A more appropriate title, in my opinion, would be “Analysing tokenisation’s effect on unigram distributions”, or something analogous to it. There is no “theory of tokenisation” being proposed here, but a theoretical analysis of how tokenisation affects a simple model’s cross-entropy.
* The abstract and introduction also significantly overclaim results, with statements such as “we study the end-to-end cross-entropy loss achieved by transformers with and without tokenization” while focusing on unigram cross-entropies. Transformers may serve as motivation to this work (as they initially learn unigram statistics), but are not in fact analysed here.

I think the paper would also be significantly more straightforward to read if the framing was fixed and it was clear from the start that the paper's analyses would focus on unigram models.

**Questions:**

My current score is mostly based on my current understand that this paper overclaims its results. I'm open to increasing my score if the authors either tone down the paper contributions' rhetoric, or make a convincing argument of why the current framing is appropriate.

> we study the end-to-end cross-entropy loss achieved by transformers with and without tokenization

I’d argue this paper does not actually study a transformer’s cross-entropy with and without tokenization, but a unigram model’s instead. Even if transformers learn unigram distributions early on (and in some tasks are never able to learn more than that), this is still a strong over-statement in my opinion.

> the models initially predict tokens according to a unigram model (in context unigrams), which delays learning the optimal solution. This phenomenon was also observed in Makkuva et al. (2024).


This phenomenon was previously shown by Chang et al., 2022; 2023.

> Line 115. Q(t1, t2, · · · , tj ) = Q#(j) Qji=1 Qtok(ti)

What does Q_{#}(j) represent?

> Figure 2a

What happens if models are trained for more iterations?

> Figure 3

I found this figure confusing. I don’t fully understand what is being presented here.

#### `References`

* Chang et al., 2022. Word Acquisition in Neural Language Models
* Chang et al., 2023. Characterizing Learning Curves During Language Model Pre-Training: Learning, Forgetting, and Stability

**Limitations:**

I think some important limitations are not sufficiently discussed in this paper. The most important of which is that the analysis focuses on unigram statistics, and transformers can clearly learn more than that. Expanding the limitations pointed out by Remark 3.3 in a dedicated limitations section could also be useful.

---

> ### Author Rebuttal · Authors · 2024-08-07
>
> We thank the reviewer for the constructive criticism. (*references cited in the common rebuttal*)
>
> **TLDR;** We are happy to see how to change the wording of the title + general rhetoric of the paper in a way which fit the contributions in the paper best. *Our proposed changes incorporating the reviewers’ points are discussed in the common rebuttal*. In doing so, we would like to add a few points to take into consideration,
>
> 1. **The tokenization objective studied in the paper (end-to-end cross-entropy loss), while seemingly obvious in hindsight, has not been the subject of any theoretical work on tokenization**.
>
>     There are several current theoretical formulations for tokenization. One viewpoint is the ability to compress the input sequence as well as possible [5],[6],[7]. Related work such as [4] considers tokenization as approximately optimizing some natural objective of the training dataset. There are two conceptual issues with these formulations,
>
>     +  These approaches study how tokenizers behave on the sequences they were trained on, which does not capture generalization.
>
>     +  These analyses look at tokenizers in isolation, rather than studying them in the context of how the end-to-end loss of the model changes.
>
>    The current theoretical discussion around tokenization is problematic for this reason: it is straightforward to compare tokenizer A with tokenizer B by evaluating them in practice, but there is no existing theoretical formulation which allows comparing different tokenizers with each other (say, by studying their behavior against the same loss/objective). For instance, the notion of “minimum number of merges” in [4] only makes sense for merging based tokenizers, and not for tokenizers like Unigram which don’t operate by merging tokens. The end-to-end cross-entropy loss is arguably the most natural objective which can allow comparisons between the behavior of the end-to-end model.
>
> 2. **Why do we not write the results as “an analysis of unigram models with tokenization”? Do transformers even learn unigram models?**
>
>    Section 2.1 and 3 of the paper are dedicated to understanding the behavior learnt by transformers when exposed to data generated from Markov processes. Prior work argues that transformers (without tokenization), when trained for many iterations, are indeed able to learn $k$-th order Markov data. In the context of these results it’s worth taking into account the fact that these models need to be trained for $10-50 \times$ the number of iterations to break past the unigram phase and learn even bigram models. See Fig. 1 of the reference [1] or Fig. 1a in the rebuttal pdf. In the presence of tokenization, Fig. 3 in our paper discusses the distributions learned in the presence of tokenization, which are observed to be close to a unigram model (see clarification in the common rebuttal).
>
> 3. **What happens when transformers don’t learn unigram models?**
>
>    This is a good question, and something that our work addresses from a theoretical point of view, but in a way that can be made more explicit (see proposed changes above). With sufficiently large dictionaries, the transformer only has to learn a unigram model. What happens when the dictionaries are not this large? The main implication from the theorems in our paper is that *tokenization always reduces the value of $n$ for which the transformer needs to learn $n$-gram behavior, so as to achieve near-optimal test loss*. Our current theorems instantiate this result when the dictionaries are large enough so it suffices to learn unigram behavior to perform well. When the dictionary sizes are smaller, transformers with tokenization are forced to learn $n$ -gram behavior for $n > 1$ to achieve low test loss. Our results argues that tokenization allows the model to get away with learning $n$-gram behavior for values of $n \ll k$, the true order of the underlying data distribution. With tokenization, transformers can avoid learning higher-order $n$-grams, which would have otherwise been learnt in its absence.
>
> 4. **What happens when transformers don’t learn $n$-gram models at all?**
>
>    $n$-gram models, while simple, do not capture the extent of the modeling power of transformers. Going beyond this requires looking at more complicated data generating processes. There are several ways to do so: the simplest extension compared to our work would be one where the transformer is trained on data generated from a mixture of many different Markov models. No single $n$-gram model captures the data well now, and it is interesting to see how transformers learn to identify what order the test data comes from and utilize the appropriate estimator. Even beyond Markov and mixture of Markov models, we may study these questions when data come from simple HMMs, where the role of tokenization may be very different.
>
>    We hope that our work initiates a study of tokenization when transformers are trained on even richer classes of data processes. We analyze the end-to-end loss by looking at the family of models $\mathcal{F}$ transformers empirically appear to represent and prove upper/lower bounds on the cross-entropy of models in $\mathcal{F}$. Ultimately, this framework can be applied to any data process, and provides a new lens for comparing tokenizers.
>
> ___
>
> ### **Additional questions/clarifications**
>
> 1. **What is $Q_{\\#} (j)$ in the unigram definition?** This is the distribution over the total number of tokens $j$. This needs to be present as otherwise the distribution over token sequences does not integrate to $1$ (i.e., is not a valid probability distribution).
> 2. **What happens when models are trained for longer in Fig. 2a:** When models are trained for more iterations, they eventually learn $k$-gram behavior. However, the number of iterations required is $10-50 \times$ more (see Fig. 1 in [1] or Fig. 1a in the rebuttal pdf)
> 3. **Figure 3 clarification:** Discussed in the common rebuttal.

---

> > ### Author Response · Authors · 2024-08-12
> > **Discussion period ending soon: call for response**
> >
> > Dear Reviewer 88v6,
> >
> > We sincerely appreciate the time you have taken to provide valuable feedback for our work. As we are getting closer to the end of the discussion period, could you let us know if our responses above have adequately addressed your concerns? We remain at your disposal for any further questions.
> >
> > If you agree that our responses to your reviews have addressed the concerns you listed, we kindly ask that you consider whether raising your score would more accurately reflect your updated evaluation of our paper. Thank you again for your time and thoughtful comments.
> >
> > Sincerely,&nbsp;
> > The Authors

---

> > ### Comment · Reviewer_88v6 · 2024-08-14
> > **Answer to Authors**
> >
> > I thank the authors for their detailed response and clarifications. I think the suggested changes listed by the authors will significantly improve it, and I have increased my score from 4 to 7.

---

### Author Rebuttal · Authors · 2024-08-07

## **Common rebuttal**

We thank all the reviewers for taking the time to go through our paper and suggest constructive criticism. Please find attached a pdf containing additional plots to aid in answering reviewers' questions. We begin with the suggested changes to the paper, and then address some common points raised by the reviewers.

### **Suggested changes to the paper**

1. We propose to change the title of the paper to:  “An Analysis of Tokenization: Transformers under Markov Data”
2. We will adjust the abstract + introduction of the paper to emphasize that we look at the class of models, $\mathcal{F}$, transformers are observed to express empirically, and then theoretically study the joint cross-entropy loss of natural tokenizers with models in $\mathcal{F}$. In the case of $k$-th order Markov data, the model class expressed by transformers, $\mathcal{F}$, appears to be the class of $n$-gram models. This is the case both with, and without tokenization (cf. Fig. 3 in the paper). We will also point out previous work studying this and related behavior such as Chang et al (2022, 2023).
3. We will adjust the description and discussion surrounding Fig. 3 to be more clear, and emphasize that the goal of this figure is to understand what is the behavior expressed by transformers trained on Markov data in the presence of tokenization. A clarification of this figure is provided below.
4. Since our theoretical discussion revolves around $n$-gram models trained with tokenization, we will make the discussion connecting $n$-gram models with transformers more clear. Currently this is spread across sections 2 and 3 of the paper.
5. We shall expand Remark 3.3 into a limitations section, in addition to some other limitations (such as: transformers can learn more than just $n$-gram behavior).
6. A discussion extending the current results to the case where transformers with tokenization learn $n$-gram behavior for $n \ge 2$ instead of just unigram models. This occurs when tokenizers are trained with very small dictionaries.

We are open to including other changes to the paper the reviewers see fit, in a way which better matches the contributions in the paper and makes it easier to read.

___

Below we address some common points raised by multiple reviewers.

### **Clarification of Fig. 3**

**Fig. 3 plots the next-token distributions learnt by the transformer** (dictionary size $=20$, sequence length $=500$) **at convergence**.

In this experiment, we sample a random sequence of length $2000$ from a Markov chain and feed it into the transformer (after tokenization, this results in a sequence of length $\approx 500$). We plot the next-token distribution predicted by the transformer at every single of the $500$ positions, generated by masking the input sequence. This plot stitches together $500$ next-token distributions together, each of which is a narrow column heatmap. Visually it appears that the transformer predicts symbols according to the same distribution most of the time (i.e., the plot looks approximately homogeneous along the $x$-axis). This is evidence that even though the transformer could learn behavior which depended on all the previous symbols, it sticks to outputting tokens independently most of the time, i.e., a unigram model.

From this point of view, the tables have turned - it is surprising that the transformer with tokenization performs so well even though it only learns unigram behavior, when this behavior is the root of problems in the absence of tokenization. Our work tries to formally address why this might be the case.

___

### **Summary**

We initiate the analytical study of the end-to-end loss of transformers with tokenization by looking at the class of models transformers appear to learn empirically, and then studying how the behavior changes with the addition of tokenization. We instantiate this framework for the case of Markov data generating processes, where transformers learn simple $n$-gram behavior, as observed in Chang et al. (2022,2023) in the context of real-world data and other more recent works (Makkuva et al). However, this framework itself can be instantiated with any kind of data process. Choosing more complex data processes will trade-off analytical tractability with the ability of transformers to express more complex behavior.

___

### **References**

[1] Edelman, Benjamin L., et al. "The evolution of statistical induction heads: In-context learning markov chains." arXiv:2402.11004.

[2] Golkar, Siavash, et al. "xVal: A continuous number encoding for large language models" arXiv:2310.02989

[3] Fujii, Takuro, et al. "How do different tokenizers perform on downstream tasks in scriptio continua languages?: A case study in Japanese." arXiv:2306.09572

[4] Zouhar, Vilém, et al. "A formal perspective on byte-pair encoding." arXiv:2306.16837

[5] Zouhar, Vilém, et al. "Tokenization and the noiseless channel." arXiv:2306.16842

[6] Gallé, Matthias. "Investigating the effectiveness of BPE: The power of shorter sequences." EMNLP-IJCNLP (2019)

[7] Goldman, Omer, et al. "Unpacking Tokenization: Evaluating Text Compression and its Correlation with Model Performance." arXiv preprint arXiv:2403.06265

[8] Svete, A., & Cotterell, R. "Transformers Can Represent $n$-gram Language Models" arXiv:2404.14994

[9] Nguyen, T. "Understanding Transformers via N-gram Statistics" arXiv:2407.12034

[10] Diao, Shizhe, et al. "Taming pre-trained language models with n-gram representations for low-resource domain adaptation" ACL 2021

[11] Irie, Kazuki, et al. "Language modeling with deep transformers" arXiv:1905.04226

[12] Tian, Jinhao, et al. "N-gram Unsupervised Compoundation and Feature Injection for Better Symbolic Music Understanding" AAAI 2024

[13] Pagliardini, M. GPT-2 modular codebase implementation. https://github.com/epfml/llm-baselines

[14] Clark, Jonathan H., et al. "Canine: Pre-training an efficient tokenization-free encoder for language representation" TACL (2022)

---

### Decision · Program_Chairs · 2024-09-25

**Decision:**

Accept (spotlight)

**Comment:**

This is an interesting paper on studying tokenization in LLMs under some assumptions on the underling data distribution (i.e. kth order markov processes). All reviewers support accepting the paper (contingent on the authors changing the title of the paper) and I think it will be valuable contribution to the community..